# DEFENDING AGAINST IMAGE CORRUPTIONS THROUGH ADVERSARIAL AUGMENTATIONS

**Dan A. Calian**[1]
dancalian@deepmind.com

**Florian Stimberg**[1]
stimberg@deepmind.com

**Olivia Wiles**[1]
oawiles@deepmind.com

**Sylvestre-Alvise Rebuffi**[1]
sylvestre@deepmind.com

**András György**[1]
agyorgy@deepmind.com

**Timothy Mann**[2]
mann.timothy@acm.org

**Sven Gowal**[1]
sgowal@deepmind.com

[1]DeepMind, [2]Meta

## ABSTRACT

Modern neural networks excel at image classification, yet they remain vulnerable to common image corruptions such as blur, speckle noise or fog. Recent methods that focus on this problem, such as AugMix and DeepAugment, introduce defenses that operate *in expectation* over a distribution of image corruptions. In contrast, the literature on $\ell_p$-norm bounded perturbations focuses on defenses against *worst-case* corruptions. In this work, we reconcile both approaches by proposing *AdversarialAugment*, a technique which optimizes the parameters of image-to-image models to generate adversarially corrupted augmented images. We theoretically motivate our method and give sufficient conditions for the consistency of its idealized version as well as that of DeepAugment. Classifiers trained using our method in conjunction with prior methods (AugMix & DeepAugment) improve upon the state-of-the-art on common image corruption benchmarks conducted in expectation on CIFAR-10-C and also improve worst-case performance against $\ell_p$-norm bounded perturbations on both CIFAR-10 and IMAGENET.

## 1 INTRODUCTION

By following a process known as Empirical Risk Minimization (ERM) (Vapnik, 1998), neural networks are trained to minimize the average error on a training set. ERM has enabled breakthroughs in a wide variety of fields and applications (Goodfellow et al., 2016; Krizhevsky et al., 2012; Hinton et al., 2012), ranging from ranking content on the web (Covington et al., 2016) to autonomous driving (Bojarski et al., 2016) via medical diagnostics (De Fauw et al., 2018). ERM is based on the principle that the data used during training is independently drawn from the same distribution as the one encountered during deployment. In practice, however, training and deployment data may differ and models can fail catastrophically. Such occurrence is commonplace as training data is often collected through a biased process that highlights confounding factors and spurious correlations (Torralba et al., 2011; Kuehlkamp et al., 2017), which can lead to undesirable consequences (e.g., http://gendershades.org).

As such, it has become increasingly important to ensure that deployed models are robust and generalize to various input corruptions. Unfortunately, even small corruptions can significantly affect the performance of existing classifiers. For example, Recht et al. (2019); Hendrycks et al. (2019) show that the accuracy of IMAGENET models is severely impacted by changes in the data collection process, while imperceptible deviations to the input, called adversarial perturbations, can cause neural networks to make incorrect predictions with high confidence (Carlini & Wagner, 2017a;b; Goodfellow et al., 2015; Kurakin et al., 2016; Szegedy et al., 2014). Methods to counteract such effects, which mainly consist of using random or adversarially-chosen data augmentations, struggle. Training against corrupted data only forces the memorization of such corruptions and, as a result, these models fail to generalize to new corruptions (Vasiljevic et al., 2016; Geirhos et al., 2018).

Recent work from Hendrycks et al. (2020b) (also known as AugMix) argues that basic pre-defined corruptions can be composed to improve the robustness of models to common corruptions. Another line of work, DeepAugment (Hendrycks et al., 2020a), corrupts images by passing them through

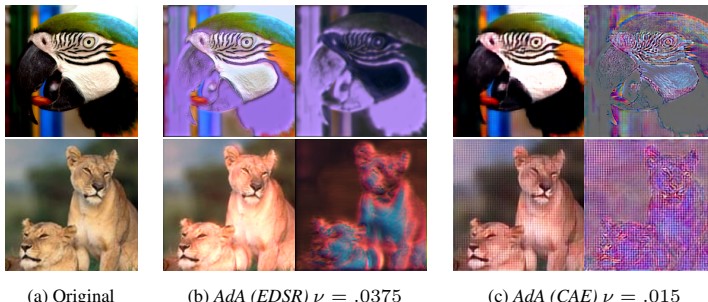

(a) Original          (b) *AdA (EDSR)* $\nu = .0375$          (c) *AdA (CAE)* $\nu = .015$

Figure 1: **Adversarial examples generated using our proposed method (*AdA*)**. Examples are shown from two different backbone architectures used with our method: *EDSR* in (b) and *CAE* in (c). Original images are shown in (a); the image pairs in (b) and (c) show the adversarial example produced by our method on the left and exaggerated differences on the right. In this case, adversarial examples found through either backbone show local and global color shifts, while examples found through *EDSR* preserve high-frequency details and ones found through *CAE* do not. Examples found through *CAE* also exhibit grid-like artifacts due to the transposed convolutions in the *CAE* decoder.

two specific image-to-image models while distorting the models' parameters and activations using an extensive range of manually defined heuristic operations. While both methods perform well on average on the common corruptions present in CIFAR-10-C and IMAGENET-C, they generalize poorly to the adversarial setting. Most recently, Laidlaw et al. (2021) proposed an adversarial training method based on bounding a neural perceptual distance (i.e., an approximation of the true perceptual distance), under the acronym of PAT for Perceptual Adversarial Training. Their method performs well against five diverse adversarial attacks, but, as it specifically addresses robustness to pixel-level attacks that directly manipulate image pixels, it performs worse than AugMix on common corruptions. In this work, we address this gap. We focus on training models that are robust to adversarially-chosen corruptions that preserve semantic content. We go beyond conventional *random data augmentation* schemes (exemplified by Hendrycks et al., 2020b;a) and *adversarial training* (exemplified by Madry et al., 2018; Gowal et al., 2019a; Laidlaw et al., 2021) by leveraging image-to-image models that can produce a wide range of semantics-preserving corruptions; in contrast to related works, our method does not require the manual creation of heuristic transformations. Our contributions are as follows:

- We formulate an adversarial training procedure, named *AdversarialAugment* (or *AdA* for short) which finds adversarial examples by optimizing over the weights of *any* pre-trained image-to-image model (i.e. over the weights of arbitrary autoencoders).

- We give sufficient conditions for the consistency of idealized versions of our method and Deep-Augment, and provide PAC-Bayesian performance guarantees, following Neyshabur et al. (2017). Our theoretical considerations highlight the potential advantages of *AdA* over previous work (DeepAugment), as well as the combination of the two. We also establish links to Invariant Risk Minimization (IRM) (Arjovsky et al., 2020), Adversarial Mixing (AdvMix) (Gowal et al., 2019a) and Perceptual Adversarial Training (Laidlaw et al., 2021).

- We improve upon the known state-of-the-art (at the time of initial submission)[1] on CIFAR-10-C by achieving a mean corruption error (mCE) of **7.83%** when using our method in conjunction with others (vs. 23.51% for Perceptual Adversarial Training (PAT), 10.90% for AugMix and 8.11% for DeepAugment). On IMAGENET we show that our method can leverage **4** pre-trained image-to-image models simultaneously (VQ-VAE of van den Oord et al., 2017, U-Net of Ronneberger et al., 2015, EDSR of Lim et al., 2017 & CAE of Theis et al., 2017) to yield the largest increase in robustness to common image corruptions, among all evaluated models.

- On $\ell_2$ and $\ell_\infty$ norm-bounded perturbations we significantly improve upon previous work (Deep-Augment & AugMix) using *AdA (EDSR)*, while slightly improving generalization performance on both IMAGENET-V2 and on CIFAR-10.1.

## 2  RELATED WORK

**Data augmentation.** Data augmentation has been shown to reduce the generalization error of standard (non-robust) training. For image classification tasks, random flips, rotations and crops are

---

[1]After our initial submission Diffenderfer et al. (2021) obtained slightly better mCE. See G.1 for more details.

commonly used (He et al., 2016a). More sophisticated techniques such as Cutout of DeVries & Taylor (2017) (which produces random occlusions), CutMix of Yun et al. (2019) (which replaces parts of an image with another) and mixup of Zhang et al. (2018a); Tokozume et al. (2018) (which linearly interpolates between two images) all demonstrate extremely compelling results. Guo et al. (2019) improved upon mixup by proposing an adaptive mixing policy. Works, such as AutoAugment (Cubuk et al., 2019) and the related RandAugment (Cubuk et al., 2020), learn augmentation policies from data directly. These methods are tuned to improve standard classification accuracy and have been shown to work well on CIFAR-10, CIFAR-100, SVHN and IMAGENET. However, these approaches do not necessarily generalize well to larger data shifts and perform poorly on benign corruptions such as blur or speckle noise (Taori et al., 2020).

**Robustness to synthetic and natural data shift.** Several works argue that training against corrupted data only forces the memorization of such corruptions and, as a result, models fail to generalize to new corruptions (Vasiljevic et al., 2016; Geirhos et al., 2018). This has not prevented Geirhos et al. (2019); Yin et al. (2019); Hendrycks et al. (2020b); Lopes et al. (2019); Hendrycks et al. (2020a) from demonstrating that some forms of data augmentation can improve the robustness of models on IMAGENET-C, despite not being directly trained on these common corruptions. Most works on the topic focus on training models that perform well in expectation. Unfortunately, these models remain vulnerable to more drastic adversarial shifts (Taori et al., 2020).

**Robustness to adversarial data shift.** Adversarial data shift has been extensively studied (Goodfellow et al., 2015; Kurakin et al., 2016; Szegedy et al., 2014; Moosavi-Dezfooli et al., 2019; Papernot et al., 2016; Madry et al., 2018). Most works focus on the robustness of classifiers to $\ell_p$-norm bounded perturbations. In particular, it is expected that a *robust* classifier should be invariant to small perturbations in the pixel space (as defined by the $\ell_p$-norm). Goodfellow et al. (2015), Huang et al. (2015) and Madry et al. (2018) laid down foundational principles to train robust networks, and recent works (Zhang et al., 2019; Qin et al., 2019; Rice et al., 2020; Wu et al., 2020; Gowal et al., 2020) continue to find novel approaches to enhance adversarial robustness. However, approaches focused on $\ell_p$-norm bounded perturbations often sacrifice accuracy on non-adversarial images (Raghunathan et al., 2019). Several works (Baluja & Fischer, 2017; Song et al., 2018; Xiao et al., 2018; Qiu et al., 2019; Wong & Kolter, 2021b; Laidlaw et al., 2021) go beyond these analytically defined perturbations and demonstrate that it is not only possible to maintain accuracy on non-adversarial images but also to reduce the effect of spurious correlations and reduce bias (Gowal et al., 2019a). Unfortunately, most aforementioned approaches perform poorly on CIFAR-10-C and IMAGENET-C.

## 3 DEFENSE AGAINST ADVERSARIAL CORRUPTIONS

In this section, we introduce *AdA*, our approach for training models robust to image corruptions through the use of adversarial augmentations while leveraging pre-trained autoencoders. In Appendix A we detail how our work relates to AugMix (Hendrycks et al., 2020b), DeepAugment (Hendrycks et al., 2020a), Invariant Risk Minimization (Arjovsky et al., 2020), Adversarial Mixing (Gowal et al., 2019a) and Perceptual Adversarial Training (Laidlaw et al., 2021).

**Corrupted adversarial risk.** We consider a model $f_\theta : \mathcal{X} \to \mathcal{Y}$ parametrized by $\theta$. Given a dataset $\mathcal{D} \subset \mathcal{X} \times \mathcal{Y}$ over pairs of examples $x$ and corresponding labels $y$, we would like to find the parameters $\theta$ which minimize the *corrupted adversarial risk*:

$$\mathbb{E}_{(x,y)\sim\mathcal{D}}\Big[ \max_{x'\in\mathcal{C}(x)} L(f_\theta(x'), y) \Big], \tag{1}$$

where $L$ is a suitable loss function, such as the 0-1 loss for classification, and $\mathcal{C} : \mathcal{X} \to 2^{\mathcal{X}}$ outputs a corruption set for a given example $x$. For example, in the case of an image $x$, a plausible corruption set $\mathcal{C}(x)$ could contain blurred, pixelized and noisy variants of $x$.

In other words, we seek the optimal parameters $\theta^*$ which minimize the corrupted adversarial risk so that $f_{\theta^*}$ is invariant to corruptions; that is, $f_{\theta^*}(x') = f_{\theta^*}(x)$ for all $x' \in \mathcal{C}(x)$. For example if $x$ is an image classified to be a horse by $f_{\theta^*}$, then this prediction should not be affected by the image being slightly corrupted by camera blur, Poisson noise or JPEG compression artifacts.

**AdversarialAugment (*AdA*).**   Our method, *AdA*, uses image-to-image models to generate adversarially corrupted images. At a high level, this is similar to how DeepAugment works: DeepAugment perturbs the parameters of two specific image-to-image models using heuristic operators, which are manually defined for each model. Our method, instead, is more general and optimizes directly over perturbations to the parameters of *any* pre-trained image-to-image model. We call these image-to-image models *corruption networks*. We experiment with four corruption networks: a vector-quantised variational autoencoder (*VQ-VAE*) (van den Oord et al., 2017); a convolutional U-Net (Ronneberger et al., 2015) trained for image completion (*U-Net*), a super-resolution model (*EDSR*) (Lim et al., 2017) and a compressive autoencoder (*CAE*) (Theis et al., 2017). The latter two models are used in DeepAugment as well. Additional details about the corruption networks are provided in Appendix E.6.

Formally, let $c_\phi : \mathcal{X} \to \mathcal{X}$ be a *corruption network* with parameters $\phi = \{\phi_i\}_{i=1}^{K}$, which acts, with perturbed parameters, upon clean examples by corrupting them. Here each $\phi_i$ corresponds to the vector of parameters in the $i$-th layer, and $K$ is the number of layers. Let $\delta = \{\delta_i\}_{i=1}^{K}$ be a weight perturbation set, so that a corrupted variant of $x$ can be generated by $c_{\{\phi_i+\delta_i\}_{i=1}^{K}}(x)$. With a slight abuse of notation, we shorten $c_{\{\phi_i+\delta_i\}_{i=1}^{K}}$ to $c_{\phi+\delta}$. Clearly, using unconstrained perturbations can result in exceedingly corrupted images which have lost all discriminative information and are not useful for training. For example, if $c_\phi$ is a multi-layer perceptron, trivially setting $\delta_i = -\phi_i$ would yield fully zero, uninformative outputs. Hence, we restrict the corruption sets by defining a maximum relative perturbation radius $\nu > 0$, and define the corruption set of *AdA* as $\mathcal{C}(x) = \{c_{\phi+\delta}(x) \mid \|\delta\|_{2,\phi} \le \nu\}$, where the norm $\|\cdot\|_{2,\phi}$ is defined as $\|\delta\|_{2,\phi} = \max_{i \in \{1,...,K\}} \|\delta_i\|_2 / \|\phi_i\|_2$.

**Finding adversarial corruptions.**   For a clean image $x$ with label $y$, a corrupted adversarial example within a bounded corruption distance $\nu$ is a corrupted image $x' = c_{\phi+\delta}(x)$ generated by the corruption network $c$ with bounded parameter offsets $\|\delta\|_{2,\phi} \le \nu$ which causes $f_\theta$ to misclassify $x$: $f_\theta(x') \ne y$. Similarly to Madry et al. (2018), we find an adversarial corruption by maximizing a surrogate loss $\tilde{L}$ to $L$, for example, the cross-entropy loss between the predicted logits of the corrupted image and its clean label. We optimize over the perturbation $\delta$ to $c$'s parameters $\phi$ by solving

$$\max_{\|\delta\|_{2,\phi} \le \nu} \tilde{L}(f_\theta(c_{\phi+\delta}(x)), y). \tag{2}$$

In practice, we solve this optimization problem (approximately) using projected gradient ascent to enforce that perturbations $\delta$ lie within the feasible set $\|\delta\|_{2,\phi} \le \nu$. Examples of corrupted images obtained by *AdA* are shown in Figure 1.

**Adversarial training.**   Given the model $f$ parameterized by $\theta$, we aim to minimize the corrupted adversarial risk from (1) by solving the following surrogate optimization problem (using the loss function $\tilde{L}$ instead of $L$):

$$\theta^* = \arg\min_\theta \mathbb{E}_{(x,y)\sim\mathcal{D}}\left[ \max_{\|\delta\|_{2,\phi} \le \nu} \tilde{L}(f_\theta(c_{\phi+\delta}(x)), y) \right]. \tag{3}$$

The full description of our algorithm is given in Appendix B.

**Meaningful corruptions.**   A crucial element of *AdA* is setting the perturbation radius $\nu$ to ensure that corruptions are varied enough to constitute a strong defense against common corruptions, while still being meaningful (i.e., without destroying semantics). We measure the extent of corruption induced by a given $\nu$ through the structural similarity index measure (SSIM) (Wang et al., 2004) between clean and corrupted images (details on how SSIM is computed are given in Appendix F.1). We plot the distributions of SSIM over various perturbation radii in Figure 2 for corrupted images produced by *AdA* using two backbones (*EDSR* and *CAE*) on CIFAR-10. We find that a relative perturbation radius of $\nu = .015$ yields enough variety in the corruptions for both *EDSR* and *CAE*. This is demonstrated for *EDSR* by having a large SSIM variance compared to, e.g. $\nu = 0.009375$, without destroying semantic meaning (retaining a high mean SSIM). We guard against unlikely but too severe corruptions (i.e. with too low SSIM) using an efficient approximate line-search procedure (details can be found in Appendix F). A similar approach for restricting the SSIM values of samples during adversarial training was used by Hameed (2020); Hameed & György (2021).

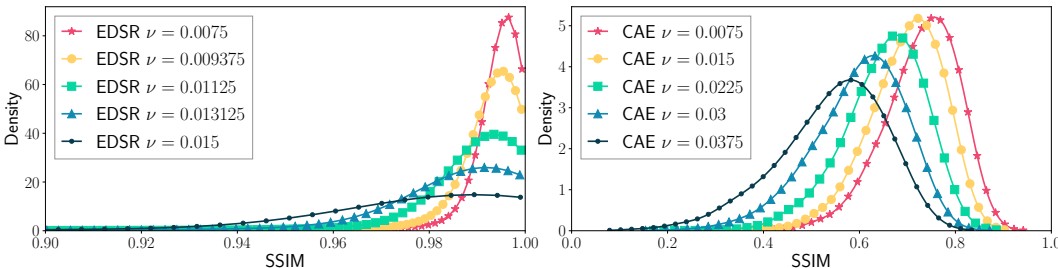

Figure 2: **Distribution of SSIM scores between clean and adversarial images found through *AdA*.** Densities are shown for two backbones (*EDSR & CAE*) backbones at five perturbation radii ($\nu$). The *AdA (EDSR)* SSIM distribution with a low perturbation radius ($\nu = 0.0075$) is highly concentrated around 0.99 yielding images very close to the clean inputs; increasing $\nu$ slightly, dissipates density rapidly. For *AdA (CAE)* increasing the perturbation radius shifts the density lower, yielding increasingly more corrupted images. Also note that the SSIM range with non-zero support of *AdA (CAE)* is much wider than of *AdA (EDSR)*.

## 4 THEORETICAL CONSIDERATIONS

In this section, we present conditions under which simplified versions of our approach (*AdA*) and previous work (DeepAugment), are consistent (i.e., as the data size grows, the expected error of the learned classifier over random corruptions converges to zero). The role of this section is two-fold: (1) to show that our algorithm is well-behaved (i.e., it converges); and (2) to introduce and to reason about sufficient assumptions for convergence.

In this section we assume that the classification problem we consider is binary (rather than multi-class) and that the loss function $L$ is the 0-1 loss. Thus, denoting the parameter space for $\theta$ and $\phi$ by $\Theta$ and $\Phi$, we assume that there exists a ground-truth binary classifier $f_{\theta^*}$ for some parameter $\theta^* \in \Theta$. We further assume that the clean input samples come from a distribution $\mu$ over $\mathcal{X}$, and the algorithm is given a set of $n$ labeled (clean) data points $\{(x_i, y_i)\}_{i=1}^n$ where $y_i = f_{\theta^*}(x_i)$.

To start with, we consider the case when the goal is to have good average performance on some future corruptions; as such, we assume that these corruptions can be described by an unknown distribution $\alpha$ of corruption parameters $\phi$ over $\Phi$. Then, for any $\theta \in \Theta$, the expected corrupted risk is defined as

$$R(f_\theta, \alpha) = \mathbb{E}_{x \sim \mu, \phi \sim \alpha} \left[ L([f_\theta \circ c_\phi](x), f_{\theta^*}(x)) \right] \quad , \tag{4}$$

which is the expected risk of $f_\theta$ composed with a random corruption function $c_\phi$, $\phi \sim \alpha$.

Since $\alpha$ is unknown, we cannot directly compute the expected corrupted risk. DeepAugment overcomes this problem by proposing a suitable replacement distribution $\beta$ and instead scores classification functions by approximating $R(f_\theta, \beta)$ (rather than $R(f_\theta, \alpha)$). In contrast, *AdA* searches over a set, which we denote by $\Phi_\beta \subseteq \Phi$ of corruptions to find the worst case, similarly to the adversarial training of Madry et al. (2018).

**DeepAugment.** The idealized version of DeepAugment (which neglects optimization issues and the use of a surrogate loss function) is defined as $\widehat{\theta}_{DA}^{(n)} = \arg\min_{\theta \in \Theta} \frac{1}{n} \sum_{i=1}^n L([f_\theta \circ c_{\phi^i}](x_i), y_i)$, where $\phi^i \sim \beta$ for $i = 1, 2, \ldots, n$. The following assumption provides a formal description of a suitable replacement distribution.

**Assumption 1. (Corruption coverage)** *There exists a known probability measure $\beta$ over $\Phi$ such that $\alpha$ is absolutely continuous with respect to $\beta$ (i.e., if $\alpha(W) > 0$ for some $W \subset \Phi$ then $\beta(W) > 0$) and for all $x$, $f_{\theta^*}(x) = f_{\theta^*}(c_\phi(x))$ for all $\phi \in \text{supp}(\beta)$, where $\text{supp}(\beta)$ is the support of the distribution $\beta$ (that is, $\Pr_{\phi \sim \beta}[f_{\theta^*}(x) = f_{\theta^*}(c_\phi(x))] = 1$).*

Assumption 1 says that while we do not know $\alpha$, we do know another distribution $\beta$ over corruption functions with support at least as broad as that of $\alpha$. Furthermore, any corruption function that belongs to the support of $\beta$ leaves the ground-truth label unchanged, which implies that the corrupted adversarial risk (1) of the optimal predictor $f_{\theta^*}$ is zero when the corruptions are restricted to $\text{supp}(\beta)$.

Given Assumption 1, the problem reduces to learning under covariate shift and, since $\theta^* \in \Theta$, empirical risk minimization yields a consistent solution to the original problem (Sugiyama et al., 2007, footnote 3), that is, the risk of $\widehat{\theta}_{DA}^{(n)}$ converges to the minimum of (4). Thus, the idealized version of DeepAugment is consistent.

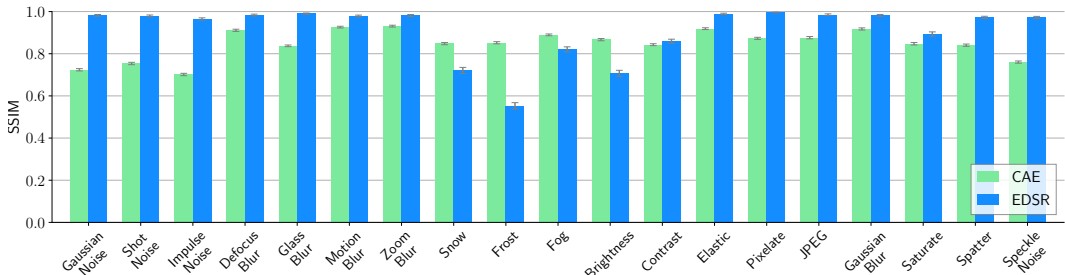

Figure 3: **Reconstructing CIFAR-10-C corruptions through two image-to-image models.** These bar plots show the extent to which two *AdA* backbones (*EDSR* & *CAE*) can be used to approximate the effects of the 15 corruptions present in CIFAR-10-C. Bars show mean (and 95% CI of the mean) SSIM (Wang et al., 2004) (higher is better) between pairs of corrupted images and their reconstructions (starting from the original images).

**AdversarialAugment.** An idealized version of *AdA* is defined as

$$\widehat{\theta}_{AdA}^{(n)} = \underset{\theta \in \Theta}{\arg\min} \frac{1}{n} \sum_{i=1}^{n} \operatorname{ess\,sup}_{\phi \sim \beta} L([f_\theta \circ c_\phi](x_i), y_i). \tag{5}$$

The essential supremum operation represents our ability to solve the difficult computational problem of finding the worst case corruption (neglecting corruptions of $\beta$-measure zero). Assuming that we can compute (5), the consistency of the idealized version of *AdA* (i.e., that the error of $\widehat{\theta}_{AdA}^{(n)}$ converges to the minimum of (1), which is 0 under Assumption 1 as discussed above) is guaranteed for predictor classes $\{f_\theta\}$ with bounded capacity measures, such as Rademacher complexity, in particular for neural networks with bounded weights, since $\theta^* \in \Theta$ (see, e.g., Anthony & Bartlett, 2009; Golowich et al., 2017). Furthermore, as the expected loss of the learned predictor converges to zero for the supremum loss (over the corruptions), so does $R(f_{\widehat{\theta}_{AdA}^{(n)}}, \beta)$, and also $R(f_{\widehat{\theta}_{AdA}^{(n)}}, \alpha)$ since $\operatorname{supp}(\alpha) \subseteq \operatorname{supp}(\beta)$ (based on Assumption 1).

**Discussion.** In Appendix C we relax this assumption to consider the case of inexact corruption coverage (Assumption 2). In Appendix D we also analyze these algorithms using the PAC-Bayesian view. In Figure 3 we explore how well Assumption 1 holds in practice: it is shown how well the 15 corruptions present in the CIFAR-10-C benchmark can be approximated by two corruption functions, *EDSR* and *CAE*. For each image pair (of a corrupted and clean image) in a random 640-image subset of CIFAR-10-C and CIFAR-10, we optimize the perturbation to the corruption network parameters that best transform the clean image into its corrupted counterpart by solving $\max_\delta \operatorname{SSIM}(c_{\phi+\delta}(x), x') - 10^{-5}\|\delta\|_2^2$, where $\delta$ is the perturbation to the corruption network's parameters, $x$ is the clean example and $x'$ is its corrupted counterpart; we also apply $\ell_2$ regularization with a constant weight (as shown above) to penalize aggressive perturbations. We use the Adam optimizer (Kingma & Ba, 2014) to take 50 ascent steps, with a learning rate of 0.001. Finally, we average the residual SSIM errors across all five severities for each corruption type. Note that both models can approximate most corruptions well, except for *Brightness* and *Snow*. Some corruption types (e.g. *Fog*, *Frost*, *Snow*) are better approximated by *CAE* ($0.84 \pm 0.16$ overall SSIM) while most are better approximated by *EDSR* ($0.91 \pm 0.26$ overall SSIM).

## 5 EMPIRICAL RESULTS

In this section we compare the performance of classifiers trained using our method (*AdA*) and competing state-of-the-art methods (AugMix of Hendrycks et al., 2020b, DeepAugment of Hendrycks et al., 2020a) on (1) robustness to common image corruptions (on CIFAR-10-C & IMAGENET-C); (2) robustness to $\ell_p$-norm bounded adversarial perturbations; and (3) generalization to distribution shifts on other variants of IMAGENET and CIFAR-10. For completeness, on CIFAR-10, we also compare with robust classifiers trained using four well-known adversarial training methods from the literature, including: Vanilla Adversarial Training (AT) (Madry et al., 2018), TRADES (Zhang et al., 2019), Adversarial Weight Perturbations (AWP) (Wu et al., 2020) as well Sharpness Aware Minimization (SAM) (Foret et al., 2021). Additional results are provided in Appendix G.

**Overview.** On CIFAR-10-C we set a new state-of-the-art (at time of initial submission) mCE of 7.83% by combining[2] *AdA (EDSR)* with DeepAugment and AugMix. On IMAGENET (downsampled

---

[2]Appendix E.5 details how methods are combined.

to $128 \times 128$) we demonstrate that our method can leverage 4 image-to-image models simultaneously to obtain the largest increases in mCE. Specifically, by combining *AdA (All)* with DeepAugment and AugMix we can obtain 62.90% mCE – which improves considerably upon the best model from the literature that we train (70.05% mCE, using nominal training with DeepAugment with AugMix). On both datasets, models trained with *AdA* gain non-trivial robustness to $\ell_p$-norm perturbations compared to all other models, while showing slightly better generalization to non-synthetic distribution shifts.

**Experimental setup and evaluation.** For CIFAR-10 we train pre-activation *ResNet50* (He et al., 2016b) models (as in Wong et al. (2020)) on the clean training set of CIFAR-10 (and evaluate on CIFAR-10-C and CIFAR-10.1); our models employ $3 \times 3$ kernels for the first convolutional layer, as in previous work (Hendrycks et al., 2020b). For IMAGENET we train standard *ResNet50* classifiers on the training set of IMAGENET with standard data augmentation but $128 \times 128$ re-scaled image crops (due to the increased computational requirements of adversarial training) and evaluate on IMAGENET-{C,R,v2}. We summarize performance on corrupted image datasets using the mean corruption error (mCE) introduced in Hendrycks & Dietterich (2019). mCE measures top-1 classifier error across 15 corruption types and 5 severities from IMAGENET-C and CIFAR-10-C. For IMAGENET only, the top-1 error for each corruption is weighted by the corresponding performance of a specific AlexNet classifier; see (Hendrycks & Dietterich, 2019). The mCE is then the mean of the 15 corruption errors. For measuring robustness to $\ell_p$-norm bounded perturbations, on CIFAR-10 we attack our models with one of the strongest available combinations of attacks: AutoAttack & MultiTargeted as done in Gowal et al. (2020); for IMAGENET we use a standard 100-step PGD attack with 10 restarts. Omitted details on the experimental setup and evaluation are provided in Appendix E.

**Common corruptions.** On CIFAR-10, models trained with *AdA* (coupled with AugMix) obtain very good performance against common image corruptions, as shown in Table 1 (left) and Table 8, excelling at *Digital* and *Weather* corruptions. Combining *AdA* with increasingly more complex methods results in monotonic improvements to mCE; i.e., coupling *AdA* with AugMix improves mCE from 15.47% to 9.40%; adding DeepAugment further pushes mCE to 7.83%.Compared to all adversarially trained baselines, we observe that *AdA (EDSR)* results in the most robustness to common image corruptions. Vanilla adversarial training (trained to defend against $\ell_2$ attacks) also produces classifiers which are highly resistant to common image corruptions (mCE of 17.42%).

On IMAGENET, in Table 2 (left) and Table 9 we see the same trend, where combining *AdA* with increasingly more methods results in similar monotonic improvements to mCE. The best method combination leverages all 4 image-to-image models jointly (*AdA (All)*), obtaining 62.90% mCE. This constitutes an improvement of more than 19% mCE over nominal training and 7.15% mCE over the best non-*AdA* method (DeepAugment + AugMix). These observations together indicate that the corruptions captured by *AdA*, by leveraging arbitrary image-to-image models, *complement* the corruptions generated by both DeepAugment and AugMix.

We observe that neither *VQ-VAE* nor *CAE* help improve robustness on CIFAR-10-C while they are both very useful for IMAGENET-C. For example, training with *AdA (VQ-VAE)* and *AdA (CAE)* results in 26.77% and 29.15% mCE on CIFAR-10-C respectively, while *AdA (All)* on IMAGENET obtains the best performance. We suspect this is the case as both *VQ-VAE* and *CAE* remove high-frequency information, blurring images (see Figure 1 (c)) which results in severe distortions for the small $32 \times 32$ images even at very small perturbation radii (see Figure 2).

**Adversarial perturbations.** For CIFAR-10, in the adversarial setting (Table 1, right) models trained with *AdA* using *any* backbone architecture perform best across all metrics. Models trained with *AdA* gain a limited form of robustness to $\ell_p$-norm perturbations despite not training *directly* to defend against this type of attack. Interestingly, combining *AdA* with AugMix actually results in a drop in robustness across *all* four $\ell_p$-norm settings for all backbones except *U-Net* (which increases in robustness). However, further combining with DeepAugment recovers the drop in most cases and increases robustness even more across the majority of $\ell_\infty$ settings. Unsurprisingly, the adversarially trained baselines (AT, TRADES & AWP) perform best on $\ell_2$- and $\ell_\infty$-norm bounded perturbations, as they are designed to defend against precisely these types of attacks. *AdA (EDSR)* obtains stronger robustness to $\ell_p$ attacks compared to SAM, while SAM obtains the best generalization to CIFAR-10.1.

On IMAGENET, as shown in Table 2 (right), the *AdA* variants that use *EDSR* obtain the highest robustness to $\ell_p$-norm adversarial perturbations; note that top-1 robust accuracy more than doubles

from *AdA (CAE)* to *AdA (EDSR)* across all evaluated $\ell_p$-norm settings. As for CIFAR-10, combining the best *AdA* variant with DeepAugment results in the best $\ell_p$ robustness performance.

By themselves, neither AugMix nor DeepAugment result in classifiers resistant to $\ell_2$ attacks for either dataset (top-1 robust accuracies are less than 4% across the board), but the classifiers have non-trivial resistance to $\ell_\infty$ attacks (albeit, much lower than *AdA* trained models). We believe this is because constraining the perturbations (in the $\ell_2$-norm) to weights and biases of the early layers of the corruption networks can have a similar effect to training adversarially with input perturbations (i.e., standard $\ell_p$-norm adversarial training). But note that *AdA* does not use any *input* perturbations.

**Generalization.** On CIFAR-10 (Table 1, center), the models trained with *AdA (U-Net)* and *AdA (EDSR)* alone or coupled with AugMix generalize better than the other two backbones to CIFAR-10.1, with the best variant obtaining 90.60% top-1 accuracy. Surprisingly, coupling *AdA* with DeepAugment can reduce robustness to the natural distribution shift captured by CIFAR-10.1. This trend applies to IMAGENET results as well (Table 2, center) when using *AdA (All)*, as combining it with AugMix, DeepAugment or both, results in lower accuracy on IMAGENET-V2 than when just using *AdA (All)*. These observations uncover an interesting trade-off where the combination of methods that should be used depends on the preference for robustness to image corruptions vs. to natural distribution shifts.

## 6 LIMITATIONS

In this section we describe the main limitations of our method: (1) the performance of *AdA* when used without additional data augmentation methods and (2) its computational complexity.

As can be seen from the empirical results (Tables 1 & 2), *AdA* performs best when used in conjunction with other data augmentations methods, i.e. with AugMix and DeepAugment. However, when *AdA* is used alone, without additional data augmentation methods, it does not result in increased corruption robustness (mCE) compared to either data augmentation method (except for one case, cf. rows 6 and 7 in Table 2). *AdA* also does not achieve better robustness to $\ell_p$-norm bounded perturbations when compared to adversarial training methods. However, this is expected, as the adversarially trained baselines (AT, TRADES and AWP) are designed to defend against precisely these types of attacks.

Our method is more computationally demanding than counterparts which utilize handcrafted heuristics. For example, AugMix specifies $K$ image operations, which are applied stochastically at the input (i.e. over images). DeepAugment requires applying heuristic stochastic operations to the weights and activations of an image-to-image model, so this requires a forward pass through the image-to-image model (which can be computed offline). But *AdA* must optimize over perturbations to the parameters of the image-to-image model; this requires several backpropagation steps and is similar to $\ell_p$-norm adversarial training (Madry et al., 2018) and related works. We describe in detail the computational and memory requirements of our method and contrast it with seven related works in Appendix B.1.

We leave further investigations into increasing the mCE performance of *AdA* when used alone, and into reducing its computational requirements, for future work.

## 7 CONCLUSION

We have shown that our method, *AdA*, can be used to defend against common image corruptions by training robust models, obtaining a new state-of-the-art (at the time of initial submission) mean corruption error on CIFAR-10-C. Our method leverages arbitrary pre-trained image-to-image models, optimizing over their weights to find adversarially corrupted images. Besides improved robustness to image corruptions, models trained with *AdA* substantially improve upon previous works' (Deep-Augment & AugMix) performance on $\ell_2$- and $\ell_\infty$-norm bounded perturbations, while also having slightly improved generalization to natural distribution shifts.

Our theoretical analysis provides sufficient conditions on the corruption-generating process to guarantee consistency of idealized versions of both our method, *AdA*, and previous work, DeepAugment. Our analysis also highlights potential advantages of our method compared to DeepAugment, as well as that of the combination of the two methods. We hope our method will inspire future work into theoretically-supported methods for defending against common and adversarial image corruptions.

Table 1: **CIFAR-10: Robustness to common corruptions, $\ell_p$ norm bounded perturbations and generalization performance.** Mean corruption error (mCE) summarizes robustness to common image corruptions from CIFAR-10-C; the corruption error for each group of corruptions (averaged across 5 severities and individual corruption types) is also shown. Generalization performance is measured by accuracy on CIFAR-10.1. Robustness to $\ell_p$ adversarial perturbations is summarized by the accuracy against $\ell_\infty$ and $\ell_2$ norm bounded perturbations.

| # | SETUP | MCE (↓) | CORRUPTION GROUP ERR. (↓) | | | | CLEAN ACC. (↑) | | $\ell_2$ ACC. (↑) | | $\ell_\infty$ ACC. (↑) | |
|---|---|---|---|---|---|---|---|---|---|---|---|---|
| | | | NOISE | BLUR | WEATHER | DIGITAL | CIFAR-10 | CIFAR-10.1 | $\epsilon = 0.5$ | $\epsilon = 1.0$ | $\epsilon = \frac{1}{255}$ | $\epsilon = \frac{2}{255}$ |
| *WITHOUT ADDITIONAL DATA AUGMENTATION:* | | | | | | | | | | | | |
| 1 | Nominal | 29.37 | 42.95 | 32.00 | 19.75 | 26.17 | 91.17 | 82.60 | 0.00 | 0.00 | 17.37 | 0.47 |
| 2 | AdA (U-Net) | 23.11 | 36.47 | 24.59 | 15.53 | 19.18 | 92.56 | 84.75 | 0.13 | 0.00 | 43.78 | 10.66 |
| 3 | AdA (VQ-VAE) | 26.77 | 26.12 | 25.38 | 28.09 | 27.34 | 78.30 | 64.65 | 5.87 | 0.40 | 41.42 | 18.42 |
| 4 | AdA (EDSR) | **15.47** | 26.83 | **14.94** | **10.80** | **12.14** | **93.37** | **86.55** | 9.28 | 0.09 | **72.41** | **41.13** |
| 5 | AdA (CAE) | 29.15 | 26.96 | 26.68 | 31.66 | 30.74 | 75.18 | 61.90 | **22.96** | **3.93** | 57.11 | 40.92 |
| 6 | AdA (All) | 18.49 | **20.86** | 19.37 | 16.98 | 17.35 | 88.49 | 78.60 | 9.76 | 0.29 | 62.30 | 33.41 |
| 7 | AT ($\ell_\infty$) | 23.64 | 20.56 | 20.29 | 25.90 | 27.04 | 86.08 | 74.20 | 51.66 | 16.64 | 82.32 | 78.20 |
| 8 | AT ($\ell_2$) | **17.42** | **13.93** | 15.20 | 19.34 | 20.33 | 91.02 | 80.40 | 67.06 | 30.73 | **85.98** | 79.77 |
| 9 | TRADES ($\ell_\infty$) | 24.72 | 21.31 | 21.49 | 27.26 | 27.99 | 84.79 | 71.00 | 54.55 | 19.51 | 81.33 | 77.59 |
| 10 | TRADES ($\ell_2$) | 18.08 | 14.07 | 15.64 | 20.46 | 21.15 | 89.96 | 78.60 | 68.16 | 37.01 | 85.07 | 79.12 |
| 11 | AWP ($\ell_\infty$) | 25.01 | 21.73 | 21.58 | 27.69 | 28.23 | 84.36 | 70.50 | 57.01 | 24.07 | 81.38 | 77.98 |
| 12 | AWP ($\ell_2$) | 18.79 | 14.92 | 15.87 | 21.54 | 21.86 | 89.20 | 77.40 | **71.79** | **44.83** | 85.18 | **80.21** |
| 13 | SAM | 24.59 | 49.44 | 24.74 | **12.80** | **17.59** | **96.17** | **90.35** | 0.01 | 0.00 | 48.10 | 6.86 |
| *WITH AUGMIX:* | | | | | | | | | | | | |
| 14 | Nominal | 12.26 | 21.11 | 10.56 | 7.60 | 11.98 | 96.09 | 89.90 | 0.13 | 0.00 | 42.26 | 7.17 |
| 15 | AdA (U-Net) | 12.02 | 20.56 | 10.34 | 8.05 | 11.25 | 95.74 | **90.60** | 1.04 | 0.00 | 54.90 | 16.08 |
| 16 | AdA (VQ-VAE) | 20.85 | 23.07 | 20.44 | 20.76 | 19.68 | 83.64 | 72.40 | 0.90 | 0.00 | 28.38 | 5.74 |
| 17 | AdA (EDSR) | **9.40** | 15.81 | 8.33 | **6.99** | 8.07 | 96.15 | 90.35 | **6.90** | 0.02 | **72.15** | 35.13 |
| 18 | AdA (CAE) | 20.20 | 20.15 | 19.40 | 21.30 | 19.94 | 84.25 | 72.85 | 2.30 | 0.00 | 42.28 | 15.22 |
| 19 | AdA (All) | 14.12 | 17.67 | 14.43 | 11.88 | 13.40 | 92.18 | 84.70 | 6.09 | **0.03** | 61.92 | 29.50 |
| *WITH DEEPAUGMENT:* | | | | | | | | | | | | |
| 20 | Nominal | **11.94** | **12.88** | 12.22 | **10.32** | 12.56 | **92.60** | 84.15 | 3.46 | 0.00 | 60.16 | 23.55 |
| 21 | AdA (U-Net) | 13.09 | 15.24 | 13.14 | 11.88 | 12.63 | 91.32 | 83.50 | 11.26 | 0.40 | 68.87 | 40.71 |
| 22 | AdA (VQ-VAE) | 26.35 | 30.35 | 24.30 | 27.08 | 24.65 | 77.79 | 63.25 | 0.03 | 0.01 | 1.60 | 0.33 |
| 23 | AdA (EDSR) | 12.37 | 14.24 | **12.16** | 11.63 | **11.91** | 91.17 | 82.20 | **26.31** | **2.87** | **76.63** | **56.33** |
| 24 | AdA (CAE) | 27.98 | 27.31 | 27.00 | 29.81 | 27.64 | 74.62 | 61.60 | 8.62 | 0.72 | 40.34 | 21.68 |
| 25 | AdA (All) | 21.70 | 22.01 | 22.20 | 21.92 | 20.75 | 82.17 | 68.95 | 11.05 | 0.95 | 58.21 | 33.07 |
| *WITH AUGMIX & DEEPAUGMENT:* | | | | | | | | | | | | |
| 26 | Nominal | 7.99 | **8.84** | 7.64 | **6.79** | 8.91 | **95.42** | **89.95** | 2.87 | 0.00 | 62.78 | 23.21 |
| 27 | AdA (U-Net) | 8.63 | 9.84 | 8.19 | 7.41 | 9.37 | 95.13 | 89.20 | 8.07 | 0.08 | 69.22 | 35.60 |
| 28 | AdA (VQ-VAE) | 25.17 | 26.94 | 24.05 | 26.41 | 23.71 | 77.36 | 65.40 | 4.62 | 0.32 | 38.81 | 15.47 |
| 29 | AdA (EDSR) | **7.83** | 9.26 | **7.61** | 7.27 | **7.55** | 94.92 | 87.35 | **18.63** | **0.99** | **77.72** | **49.95** |
| 30 | AdA (CAE) | 20.09 | 19.65 | 19.70 | 21.48 | 19.43 | 83.60 | 71.20 | 2.97 | 0.04 | 43.28 | 17.18 |
| 31 | AdA (All) | 11.72 | 12.49 | 11.77 | 11.38 | 11.43 | 91.53 | 82.95 | 11.60 | 0.78 | 62.38 | 34.82 |

Table 2: **IMAGENET: Robustness to common corruptions, $\ell_p$ norm bounded perturbations and generalization performance.** Mean corruption error (mCE) summarizes robustness to common image corruptions from IMAGENET-C; the corruption error for each major group of corruptions is also shown. We measure generalization to IMAGENET-V2 ("IN-v2") and IMAGENET-R ("IN-R"). Robustness to $\ell_p$ adversarial perturbations is summarized by the accuracy against $\ell_\infty$ and $\ell_2$ norm bounded perturbations.

| # | SETUP | MCE (↓) | CORRUPTION GROUP ERR. (↓) | | | | CLEAN ACC. (↑) | | | $\ell_2$ ACC. (↑) | | $\ell_\infty$ ACC. (↑) | |
|---|---|---|---|---|---|---|---|---|---|---|---|---|---|
| | | | NOISE | BLUR | WEATHER | DIGITAL | IN | IN-v2 | IN-R | $\epsilon = 0.5$ | $\epsilon = 1.0$ | $\epsilon = \frac{1}{255}$ | $\epsilon = \frac{2}{255}$ |
| *WITHOUT ADDITIONAL DATA AUGMENTATION:* | | | | | | | | | | | | | |
| 1 | Nominal | 82.40 | 76.01 | 71.43 | 53.05 | 62.23 | **74.88** | **62.97** | 18.04 | 15.40 | 1.82 | 0.23 | 0.01 |
| 2 | AdA (U-Net) | 83.51 | 85.25 | 78.13 | 49.44 | 56.07 | 70.59 | 59.15 | 21.72 | 15.58 | 3.06 | 0.97 | 0.10 |
| 3 | AdA (VQ-VAE) | 78.26 | **72.66** | **66.80** | 58.75 | 52.01 | 60.33 | 48.09 | 23.38 | 24.04 | 7.92 | 5.24 | **0.41** |
| 4 | AdA (EDSR) | 79.59 | 76.10 | 69.83 | 50.22 | 58.43 | 73.05 | 60.83 | 23.19 | **32.93** | **9.99** | **6.15** | 0.41 |
| 5 | AdA (CAE) | 86.44 | 91.06 | 70.32 | 61.30 | 57.55 | 56.67 | 46.32 | 18.55 | 13.76 | 3.05 | 1.03 | 0.08 |
| 6 | AdA (All) | **75.03** | 78.57 | 68.49 | **47.08** | **48.61** | 73.20 | 61.30 | **24.43** | 23.97 | 6.42 | 2.97 | 0.20 |
| *WITH AUGMIX:* | | | | | | | | | | | | | |
| 7 | Nominal | 77.12 | 72.52 | 66.30 | 48.63 | 57.91 | 74.25 | **62.19** | 19.22 | 13.18 | 1.74 | 0.23 | 0.01 |
| 8 | AdA (U-Net) | 77.87 | 76.58 | 71.52 | 46.58 | 54.28 | 71.20 | 59.58 | 23.37 | 15.77 | 3.00 | 0.78 | 0.06 |
| 9 | AdA (VQ-VAE) | 73.41 | 72.73 | 64.74 | 49.49 | **48.31** | 65.12 | 53.52 | 22.22 | 10.51 | 1.75 | 0.28 | 0.01 |
| 10 | AdA (EDSR) | 73.59 | 69.06 | 65.84 | 47.89 | 52.06 | **74.31** | 61.58 | 23.65 | **34.29** | **12.21** | **7.32** | **0.50** |
| 11 | AdA (CAE) | 80.85 | 88.40 | **64.56** | 57.54 | 52.38 | 60.68 | 49.28 | 21.17 | 16.95 | 3.73 | 1.14 | 0.09 |
| 12 | AdA (All) | **72.27** | **68.28** | 65.05 | **46.55** | 50.21 | 71.68 | 59.43 | **24.25** | 20.83 | 4.74 | 1.68 | 0.11 |
| *WITH DEEPAUGMENT:* | | | | | | | | | | | | | |
| 13 | Nominal | 73.04 | 52.49 | 64.67 | 47.68 | 61.26 | 72.25 | 60.50 | 21.67 | 18.16 | 3.61 | 0.71 | 0.02 |
| 14 | AdA (U-Net) | 75.03 | 56.97 | 70.91 | 46.54 | 57.79 | 68.27 | 56.60 | 25.28 | 20.28 | 4.52 | 1.94 | 0.10 |
| 15 | AdA (VQ-VAE) | 69.15 | 57.39 | 62.89 | 48.74 | 48.43 | 66.01 | 54.55 | 25.11 | 12.52 | 2.12 | 0.38 | 0.03 |
| 16 | AdA (EDSR) | 65.62 | 48.88 | **62.63** | 45.94 | **47.37** | **73.60** | **61.65** | 25.83 | **42.67** | **16.87** | **12.83** | **1.16** |
| 17 | AdA (CAE) | 77.55 | 66.90 | 67.90 | 48.26 | 59.59 | 66.16 | 54.90 | 22.80 | 11.08 | 1.82 | 0.24 | 0.02 |
| 18 | AdA (All) | **65.54** | **47.78** | 62.81 | **45.11** | 47.80 | 70.61 | 59.06 | **27.18** | 31.08 | 9.57 | 5.79 | 0.32 |
| *WITH AUGMIX & DEEPAUGMENT:* | | | | | | | | | | | | | |
| 19 | Nominal | 70.05 | 50.65 | 61.15 | **44.67** | 59.13 | 71.57 | 60.11 | 22.32 | 16.07 | 2.44 | 0.35 | 0.01 |
| 20 | AdA (U-Net) | 71.64 | 55.41 | 66.16 | 44.81 | 55.38 | 68.59 | 57.04 | 25.93 | 19.44 | 4.20 | 1.58 | 0.06 |
| 21 | AdA (VQ-VAE) | 69.02 | 54.25 | 61.25 | 51.01 | 48.96 | 60.28 | 49.26 | 24.76 | 14.88 | 2.90 | 0.88 | 0.03 |
| 22 | AdA (EDSR) | 64.31 | 48.19 | **58.36** | 44.74 | 48.19 | **72.03** | **60.20** | 25.83 | **34.82** | **10.85** | **6.67** | **0.36** |
| 23 | AdA (CAE) | 67.89 | 55.70 | 59.53 | 49.41 | 48.09 | 62.29 | 51.18 | 23.12 | 12.91 | 2.15 | 0.43 | 0.03 |
| 24 | AdA (All) | **62.90** | 48.45 | 58.51 | 44.88 | **44.34** | 69.26 | 57.57 | **27.83** | 27.10 | 7.41 | 3.89 | 0.15 |

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

## A    RELATIONSHIPS TO RELATED WORK

### A.1    RELATIONSHIP TO DEEPAUGMENT & AUGMIX

AugMix is a data augmentation method which stochastically composes standard image operations which affect color (e.g., posterize, equalize, etc.) and geometry (e.g., shear, rotate, translate) (Hendrycks et al., 2020b). Classifiers trained with AugMix excel at robustness to image corruptions, and AugMix (when used with a Jensen-Shannon divergence consistency loss) sets the known state-of-the-art on CIFAR-10-C of 10.90%, prior to our result herein.

DeepAugment is a data augmentation technique introduced by Hendrycks et al. (2020a). This method creates novel image corruptions by passing each clean image through two specific pre-defined image-to-image models (*EDSR* and *CAE*) while distorting the network's internal parameters as well as

intermediate activations. An extensive range of manually defined heuristic operations (9 for *CAE* and 17 for *EDSR*) are stochastically applied to the internal parameters, including: transposing, negating, zero-ing, scaling and even convolving (with random convolutional filters) parameters randomly. Activations are similarly distorted using pre-defined operations; for example, random dimensions are permuted or scaled by random Rademacher or binary matrices. Despite this complexity, classifiers trained with DeepAugment, in combination with AugMix, set the known state-of-the-art mCE on IMAGENET-C of 53.6%.

Any method which relies on image-to-image models (e.g., *AdA* or DeepAugment) will inherit the limitations of the specific image-to-image models themselves. In this paper we focus on local image corruptions, but one might want to train classifiers robust to global image transformations – and it is known that most image-to-image models are not well suited for capturing global image transformations (Richardson & Weiss, 2021). For modeling global image transformations, one could, for example, consider using an image-to-image model consisting of a single spatial transformer layer (Jaderberg et al., 2015). Applied at the input layer, the spatial transformer differentiably parameterizes a global image transformation. Leveraging such an image-to-image model would allow *AdA* to find adversarial examples by optimizing over the global image transformation. But, using such a model with DeepAugment would first require manually defining heuristic weight transformations for this specific image-to-image model.

Similarly to both AugMix and DeepAugment, *AdA* is also "specification-agnostic", i.e., all three methods can be used to train classifiers to be robust to corruptions which are not known in advance (e.g., CIFAR-10-C, IMAGENET-C). Both DeepAugment and *AdA* operate by perturbing the parameters of image-to-image models. DeepAugment constrains parameter perturbations to follow pre-specified user-defined directions (i.e., defined heuristically); *AdA* bounds perturbation magnitudes automatically and prevents severe distortions (by approximately constraining the minimum SSIM deviation of the corrupted examples). To be able to define heuristic operations, DeepAugment relies on in-depth knowledge of the internals of the image-to-image models it employs; our method, *AdA*, does not. Instead, our method requires a global *scalar* threshold $\nu$ to specify the amount of allowed *relative* parameter perturbation. AugMix requires having access to a palette of pre-defined useful input transformations; our method only requires access to a pre-trained autoencoder.

We believe the generic framework proposed in our paper, which (at a minimum) requires an auto-encoder and a scalar perturbation radius, is applicable to other domains beyond images. We leave further investigations to future work.

## A.2 RELATIONSHIP TO INVARIANT RISK MINIMIZATION

Invariant Risk Minimization (IRM) proposed by Arjovsky et al. (2020) considers the case where there are multiple datasets $D_e = \{x_i, y_i\}_{i=1}^n$ drawn from different training environments $e \in \mathcal{E}$. The motivation behind IRM is to minimize the worst-case risk

$$\max_{e \in \mathcal{E}} \mathbb{E}_{(x,y) \in D_e} \left[ L(f_\theta(x), y) \right]. \tag{6}$$

In our work, the environments are defined by the different corruptions $x'$ resulting from adversarially choosing the parameter offsets $\delta$ of $\phi$. Given a dataset $\{x_i, y_i\}_{i=1}^n$, we can rewrite the *corrupted adversarial risk* shown in (1) as (6) by setting the environment set $\mathcal{E}$ to

$$\mathcal{E} = \{\{c_{\phi+\delta}(x_i), y_i\}_{i=1}^n | \|\delta\|_{2,\phi} \le \nu\}. \tag{7}$$

This effectively creates an ensemble of datasets for all possible values of $\delta$ for all examples. The crucial difference between IRM and *AdA* is in the formulation of the risk. In general, we expect *AdA* to be more risk-averse than IRM, as it considers individual examples to be independent from each other.

## A.3 RELATIONSHIP TO ADVERSARIAL MIXING

Gowal et al. (2019a) formulate a similar adversarial setup where image perturbations are generated by optimizing a subset of latents corresponding to pre-trained generative models. In our work, we can consider the parameters of our image-to-image models to be latents and could formulate Adversarial Mixing (AdvMix) in the *AdA* framework. Unlike AdvMix, we do not need to rely on a known partitioning of the latents (i.e., disentangled latents), but do need to restrict the feasible set of parameter offsets $\delta$.

### A.4 RELATIONSHIP TO PERCEPTUAL ADVERSARIAL TRAINING

Perceptual Adversarial Training (PAT) (Laidlaw et al., 2021) finds adversarial examples by optimizing pixel-space perturbations, similar to works on $\ell_p$ norm robustness, e.g. (Madry et al., 2018). The perturbed images are constrained to not exceed a maximum distance from the corresponding original (clean) images measured in the LPIPS metric (Zhang et al., 2018b). Their setup requires a complex machinery to project perturbed images back to the feasible set of images (within a fixed perceptual distance). *AdA*, by construction, uses a well-defined perturbation set and projecting corrupted network's parameters is a trivial operation. This is only possible with our method because perturbations are defined on weights and biases rather than on input pixels.

## B ALGORITHM DETAILS & COMPUTATIONAL AND SPACE COMPLEXITY

Algorithm 1 contains the algorithm listing for our proposed method. We illustrate our algorithm using SGD for clarity. In practice, we batch training examples together and use the Adam optimizer (Kingma & Ba, 2014) to update the classifier's parameters $\theta$. We still compute adversarial examples for each training sample *individually* using projected FGSM (Goodfellow et al., 2015) steps.

---

**Algorithm 1** *AdA*, our proposed method.

---

1: **Inputs:** training dataset $D$; classifier's ($f_\theta$) initial parameters $\theta^{(0)}$; corruption network $c_\phi$; corruption network's pretrained parameters $\phi$; relative perturbation radius over the corruption network's parameters $\nu$; number of layers of the corruption network $K$; learning rate $\eta_f$ and number of gradient descent steps $N$ for the outer optimization; learning rate $\eta_c$ and number of projected gradient ascent steps $M$ for the inner optimization.

2: **for** $t = 1 \ldots N$ **do**          ▷ Outer optimization over $\theta$.

3:      $(x, y) \sim D$

4:      **for** $i = 1 \ldots K$ **do**          ▷ Initialize $\delta$ perturbation.

5:          $r \sim \mathrm{U}(0, \nu \|\phi_i\|_2)$

6:          $\delta_i^{(0)}$ = uniformly random vector of the same shape as $\phi_i$ with length equal to $r$

7:      **for** $j = 1 \ldots M$ **do**          ▷ Inner optimization over $\delta$ using PGD.

8:          $\delta^{(j)} = \delta^{(j-1)} + \eta_c \, \mathrm{sign}[\nabla_\delta \tilde{L}(f_\theta(c_{\phi+\delta}(x)), y)]\Big|_{\delta = \delta^{(j-1)}}$      ▷ FGSM step.

9:          **for** $i = 1 \ldots K$ **do**          ▷ Project $\delta_i$ to lie in $\nu$-length $\ell_2$-ball around $\phi_i$.

10:             **if** $\|\delta_i^{(j)}\|_2 > \nu \|\phi_i\|_2$ **then**

11:                $\delta_i^{(j)} = \frac{\delta_i^{(j)}}{\|\delta_i^{(j)}\|_2} \cdot \nu \|\phi_i\|_2$

12:      $x' = c_{\phi+\delta^{(M)}}(x)$          ▷ The adversarial example $x'$.

13:      $x' = \mathrm{SSIMLineSearch}(x, x')$          ▷ Approx. SSIM line-search: App. F.1.

14:      $\theta^{(t)} = \theta^{(t-1)} - \eta_f \, \nabla_\theta \tilde{L}(f_\theta(x'), y)\Big|_{\theta = \theta^{(t-1)}}$          ▷ Update classifier parameters.

15: **Return** optimized classifier parameters $\theta^{(N)}$

---

### B.1 COMPUTATIONAL COMPLEXITY AND MEMORY REQUIREMENTS

Table 3 lists the computational and memory requirements of our method and of related works. We primarily compare to similar methods which perform iterative optimization to find adversarial examples: Vanilla Adversarial Training (AT) (Madry et al., 2018), TRADES (Zhang et al., 2019), Adversarial Weight Perturbations (AWP) (Wu et al., 2020) and Perceptual Adversarial Training (PAT) (Laidlaw et al., 2021). For completeness, we also compare to related methods which do not perform adversarial optimization: Sharpness Aware Minimization (SAM) Foret et al. (2021), DeepAugment (Hendrycks et al., 2020a) and AugMix (Hendrycks et al., 2020b). We characterize the number of passes (forward and backward, jointly) through the main classifier network ($f_\theta$) and, where applicable, through an auxiliary neural network ($c_\phi$) separately. For methods using adversarial optimization we also describe the cost of the projection steps used for keeping perturbations within

the feasible set. Taken together, these costs represent the computational complexity of performing one training step with each method.

Note that for our method, the auxiliary network corresponds to the corruption network (i.e. the image-to-image model); for PAT it corresponds to the network used to compute the LPIPS (Zhang et al., 2018b) metric. We characterize the PAT variant which uses the Lagrangian Perceptual Attack in the externally-bounded case and the default bisection method for projecting images into the LPIPS-ball.

Table 3: **Computational complexity and memory requirements for one training step of our method and related works.** $M$ represents the number of inner/adversarial optimizer steps (i.e. PGD steps in *AdA* or AT); $|x|$ is the dimensionality of the input; $|\theta|$ and $|\phi|$ are the number of parameters of the main classifier and of the auxiliary network respectively. The number of adversarial weight perturbations in AWP or SAM is denoted by $W$; this is typically set to 1 (Wu et al., 2020; Foret et al., 2021). For PAT, we also refer to the number of bisection iterations as $N$ and to the number of steps used for updating the Lagrange multiplier as $S$; N is set to 10 and S to 5 by default (Laidlaw et al., 2021, Appendix A.3).

| | FWD. AND BWD. PASSES THROUGH | | |
|---|---|---|---|
| SETUP | CLASSIFIER ($f_\theta$) | AUX. NET ($c_\phi$) | ADV. PROJECTION |
| METHODS WHICH PERFORM ADVERSARIAL OPTIMIZATION: | | | |
| *AdA* (this work) | $\mathcal{O}(M)$ | $\mathcal{O}(M)$ | $\mathcal{O}(M \cdot |\phi| + |x|)$ |
| PAT (Laidlaw et al., 2021) | $\mathcal{O}(M \cdot S)$ | $\mathcal{O}(M \cdot S)$ | $\mathcal{O}(N)$ |
| AT (Madry et al., 2018) | $\mathcal{O}(M)$ | - | $\mathcal{O}(M \cdot |x|)$ |
| TRADES (Zhang et al., 2019) | $\mathcal{O}(M)$ | - | $\mathcal{O}(M \cdot |x|)$ |
| AWP (Wu et al., 2020) | $\mathcal{O}(M + W)$ | - | $\mathcal{O}(M \cdot |x| + W \cdot |\theta|)$ |
| METHODS WHICH DO NOT PERFORM ADVERSARIAL OPTIMIZATION: | | | |
| SAM (Foret et al., 2021) | $\mathcal{O}(W)$ | - | - |
| DeepAugment (Hendrycks et al., 2020a) | $\mathcal{O}(1)$ | $\mathcal{O}(1)$ | - |
| AugMix (Hendrycks et al., 2020b) | $\mathcal{O}(1)$ | - | - |

**Computational complexity.** For each input example, *AdA* randomly initializes weight perturbations in $\mathcal{O}(|\phi|)$ time. It then adversarially optimizes the perturbation using M PGD steps. These steps amount to M forward and M backward steps through the main classifier ($f_\theta$) and the corruption network ($c_\phi$). At the end of each iteration, the weight perturbation is projected back onto the feasible set by layer-wise re-scaling in $\mathcal{O}(|\phi|)$ time. After the PGD iterations, an additional forward pass is done through the corruption network, using the optimized perturbation, to augment the input example. Finally, the SSIM safe-guarding step (see Appendix F) takes time proportional to $\mathcal{O}(|x|)$.

Similar to *AdA*, PAT performs multiple forward and backward passes through both the main classifier and an auxiliary network. PAT searches for a suitable Lagrange multiplier using $S$ steps, and for each candidate multiplier it performs an $M$-step inner optimization. This results in (at most) $S$ times more forward and backward passes through the two networks when compared to *AdA*. The projection step of PAT is considerably more expensive, requiring $N+1$ forward passes through the auxiliary network; however, it is applied only once at the end of the optimization, rather than at every iteration as in *AdA*.

AT, TRADES and AWP perform $M$ PGD steps, with forward and backward passes done only through the main classifier network – they do not use an auxiliary network. Hence, these methods have reduced computational complexity compared to PAT and *AdA*.

DeepAugment augments each data sample by passing it through an auxiliary network while stochastically perturbing the network's weights and intermediate activations (using a pre-defined set of operations); this amounts to one (albeit modified) forward pass through the auxiliary network for each input example. AugMix stochastically samples and composes standard (lightweight) image operations; it does not pass images through any auxiliary network. AugMix applies at most $3 \cdot K$ image operations for each input image; K is set to 3 by default (Hendrycks et al., 2020b, Section 3). Methods which use adversarial training, AT, TRADES, PAT as well as *AdA*, have higher computa-

tional complexity than both DeepAugment or AugMix, as they specifically require multiple iterations ($M$) of gradient ascent to construct adversarial examples.

**Memory requirements.** Compared to AT, TRADES & PAT which operate on input perturbations, *AdA* has higher memory requirements as it operates on weight perturbations ($\phi$) instead. Similar to AT, TRADES and PAT, *AdA* also operates on each input example independently. For a mini-batch of inputs, naively, this would require storing the corresponding weight perturbations for each input at the same time in memory. This could amount to considerable memory consumption, as the storage requirements grow linearly with the batch size and the number of parameters in the corruption network. Instead, we partition each mini-batch of inputs into multiple smaller "nano-batches", and find adversarial examples one "nano-batch" at a time. In practice, on each accelerator (GPU or TPU), we use 8 examples per "nano-batch" for CIFAR-10 and 1 for IMAGENET. This allows us to trade-off performance (i.e. parallelization) with memory consumption.

In contrast to AWP, which computes perturbations to the weights of the main classifier ($\theta$) at the mini-batch level, *AdA* computes perturbations to the weights of the corruption network ($\phi$) for each input example individually.

DeepAugment samples weight perturbations whose storage grows linearly with the number of parameters in the auxiliary network ($\mathcal{O}(|\phi|)$), matching *AdA*'s output storage requirements. But note that all methods which use adversarial optimization (*AdA* included), must also store intermediate activations during each backward pass.

For each input sample AugMix updates an input-sized array using storage proportional to $\mathcal{O}(|x|)$, having the same storage requirements as AT or PAT.

Our method must keep the weights of the image-to-image models in memory. This does not incur a large cost as all models that we use are relatively small: the *U-Net* model has 58627 parameters (0.23MB); the *VQ-VAE* model has 762053 parameters (3.05MB); the *EDSR* model has 1369859 parameters (5.48MB) and the *CAE* model has 2241859 parameters (8.97MB).

## C ADDITIONAL THEORETICAL CONSIDERATIONS ON THE CORRUPTION COVERAGE

As in Section 4, we consider binary classification problems with $L$ being the 0-1 loss.

**Inexact corruption coverage.** Assumption 1 on corruption coverage, introduced in Section 4, can be unreasonable in practical settings: for example, if the perturbations induced by elements of $\Phi$ can be unbounded (e.g., for simple Gaussian perturbations), the assumption that the labels do not change with any of the perturbations are usually violated (e.g., for sure for Gaussian perturbations). On the other hand, it may be reasonable to assume that there exists a subset of the perturbations, $\Phi_1 \subset \Phi$ such that all perturbations in $\Phi_1$ keep the label unchanged for any input $x \in \mathcal{X}$. Even this assumption becomes unrealistic if input points can be arbitrarily close to the decision boundary, in which case even for arbitrarily small perturbations we can find inputs for which the label changes. However, it is reasonable to assume that this does not happen if we only consider input points sufficiently far from the boundary (which is only meaningful if the probability of getting too close to the boundary is small, say at most some $\gamma > 0$). We formalize these considerations in the next assumption, where we assume that there exists a subset of input points $\mathcal{X}_\gamma \subset \mathcal{X}$ with $\mu(\mathcal{X}_\gamma) \geq 1 - \gamma$ and a subset of perturbations $\Phi_1$ such that Assumption 1 holds on $\mathcal{X}_\gamma$ and $\Phi_1$:

**Assumption 2. (Inexact corruption coverage)** *Let* $\gamma \in [0, 1)$. *Let* $\Phi_1 \subset \Phi$ *and* $\mathcal{X}_\gamma \subset \mathcal{X}$ *with* $\mu(\mathcal{X}_\gamma) \geq 1 - \gamma$ *such that (i) no perturbation in* $\Phi_1$ *changes the label of any* $x \in \mathcal{X}_\gamma$, *that is,* $f_{\theta^*}(x) = f_{\theta^*}(c_\phi(x))$ *for any* $\phi \in \Phi_1$; *(ii) there exists a distribution* $\beta$ *supported on* $\Phi_1$ *such that* $\alpha$ *is absolutely continuous with respect to* $\beta$ *on* $\Phi_1$.

Assuming we have access to a distribution $\beta$ satisfying the above assumption and $\mathcal{X}_\gamma$, we can consider the performance of the idealized AdA rule (5) for $x_1, \ldots, x_n \in \mathcal{X}_\gamma$. Again by the result of Sugiyama et al. (2007), under Assumption 2, the resulting robust error restricted to $\mathcal{X}_\gamma$ and $\Phi_1$ converges, as $n \to \infty$, to the minimum restricted robust error

$$\mathbb{E}_{x \sim \mu|_{\mathcal{X}_\gamma}, \phi \sim \alpha|_{\Phi_1}}[L(f_\theta \circ c_\phi(x), f_{\theta^*}(x))],$$

where $\mu|_{\mathcal{X}_\gamma}$ and $\alpha|_{\Phi_1}$ denote the restrictions of $\mu$ and $\alpha$ to $\mathcal{X}_\gamma$ and $\Phi_1$, respectively. Since $\theta^* \in \Theta$ and by the assumption, no corruption in $\Phi_1$ changes the label, this minimum is in fact 0. Then, since we assumed that $L$ is the 0-1 loss, the error of the classifier when the input is not in $\mathcal{X}_\gamma$ or the perturbation is not in $\Phi_1$, can be bounded by 1. Hence, the robust error of the learned classifier $f_{\widehat{\theta}_{AdA}}$ (obtained for $n = \infty$, i.e., in the limit of infinite data) can be bounded as

$$\mathbb{E}_{x\sim\mu,\phi\sim\alpha}[L(f_{\widehat{\theta}_{AdA}} \circ c_\phi(x), f_{\theta^*}(x))]$$
$$\leq \mathbb{E}_{x\sim\mu|_{\mathcal{X}_\gamma},\phi\sim\alpha|_{\Phi_1}} L(f_{\widehat{\theta}_{AdA}} \circ c_\phi(x), f_{\theta^*}(x)) + \gamma + 1 - \alpha(\Phi_1)$$
$$= \mathbb{E}_{x\sim\mu|_{\mathcal{X}_\gamma},\phi\sim\alpha|_{\Phi_1}} L(f_{\theta^*} \circ c_\phi(x), f_{\theta^*}(x)) + \gamma + 1 - \alpha(\Phi_1)$$
$$= \gamma + 1 - \alpha(\Phi_1)$$

Note that in the assumption there is an interplay between $\mathcal{X}_\gamma$ and $\Phi_1$ (the conditions on $\Phi_1$ only apply to inputs from $\mathcal{X}_\gamma$, and as $\mathcal{X}_\gamma$ decreases, $\Phi_1$ can increase). As one should always choose $\mathcal{X}_\gamma$ so that $\Phi_1$ be the largest given $\gamma$, to get the best bound, one can optimize $\gamma$ to minimize $\gamma + 1 - \alpha(\Phi_1)$. If the probability of the inputs close to the decision boundary ($\gamma$) and the probability of perturbations changing the label for some input ($1 - \alpha(\Phi_1)$) are small enough, the resulting bound also becomes small.

## D   PAC-BAYESIAN ANALYSIS

We can also reason about the idealized *AdA* and DeepAugment algorithms (from Section 4 in the main manuscript) using the PAC-Bayesian view. If random perturbations $\epsilon$ are introduced to the parameter $\theta$ of the classifier $f_\theta$, the following bound holds (see, e.g., Neyshabur et al., 2017):[3] Given a prior distribution $P$ over $\Theta$, which is independent of the training data, for any $\eta \in (0, 1)$, with probability at least $1 - \eta$,

$$\mathbb{E}_\epsilon[R(f_\theta, \alpha)] \leq \mathbb{E}_{\epsilon,\phi\sim\alpha}[\widehat{R}(f_\theta)] + 4\sqrt{\frac{\mathrm{KL}(\theta + \epsilon \| P) + \log\frac{2n}{\eta}}{n}}, \tag{8}$$

where $\widehat{R}(f_\theta) = \frac{1}{n}\sum_{i=1}^n L([f_\theta \circ c_\phi](x_i), y_i)$ and $\mathrm{KL}(\theta + \epsilon \| P)$ is the KL divergence between the parameter distribution $\theta + \epsilon$ (given $\theta$) and $P$.[4] Defining $P$ and $\epsilon$ to have spherically invariant normal distributions with variance $\sigma^2$ in every direction, the KL term becomes $\frac{\|\theta\|_2^2}{2\sigma^2}$, and so the second term goes to zero as the number of samples increases (and $\theta$ does not grow fast). An idealistic choice (recommended, e.g., by Neyshabur et al., 2017) is to choose the perturbation of each parameter to be proportional to the parameter value itself, by setting the standard deviation of $\epsilon_j$ (the $j$th coordinate of $\epsilon$) and the corresponding coordinate of $P$ as $\sigma_j = \sigma|\theta_j| + b$, making the KL term equal to $\sum_j \frac{\theta_j^2}{2\sigma_j^2}$. Note, however, that since $f_\theta$ depends on the training data, this choice makes the bound in (8) invalid, hence, in our experiment we choose $\epsilon_i$ to be proportional to the average norm of the weights in each layer when trained on a different (but similar) dataset. Note that minimizing the first term on the right hand side of (8) is not straightforward during training, since we have no access to corruptions sampled from $\alpha$. DeepAugment tries to remedy this situation by using samples from $\beta$; however, the effectiveness of this depends on how well $\beta$ approximates $\alpha$, more directly on the importance sampling ratios $\alpha(W)/\beta(W)$ for $W \subset \Phi$ with $\alpha(W) > 0$ (see Assumption 1). On the other hand, *AdA* minimizes the worst-case loss over $\mathrm{supp}(\beta)$, which dominates the worst-case loss over $\mathrm{supp}(\alpha)$ under Assumption 1, which minimizes the expected loss over $\alpha$, which is the first term. Thus, *AdA* minimizes an upper bound on the first term, while DeepAugment only minimizes a proxy.[5]

We have assumed that $\theta^* \in \Theta$ implying that the classifier is over-parameterized. This is a common assumption in the literature which often holds in practice (Zhang et al., 2016).

---

[3]Experimental results presented in Table 4 show that if the variance of $\epsilon$ is small enough, the performance of the classifier only changes very slightly.

[4]To obtain this bound, one can think of the randomized classifier as the compound classifier $f_\theta \circ c_\phi$ having parameters both $\theta$ and $\phi$, and the prior on $\phi$ is $\alpha$, which then cancels from the KL term.

[5]Note that this is a calibrated proxy in the sense that the minimum of both the proxy and the original target is assumed at the same value $\theta^*$.

### D.1 Performance under stochastic parameter perturbations

In Table 4, we show that the performance of an *AdA*-trained classifier gradually degrades under increasingly larger perturbations to its parameters, providing experimental support for the PAC-Bayes analysis.

Let $w$ denote a block of parameters (e.g., the set of convolutional filters of a layer) and $w_j$ be the $j$-th coordinate of the parameter block; then, $\epsilon_j$ is the stochastic perturbation we add to the individual scalar parameter $w_j$. We draw the perturbation $\epsilon_j$ from a 0-mean Gaussian with standard deviation proportional to the $\ell_\infty$ norm (i.e., the maximum absolute value) of the enclosing parameter block: $\epsilon_j \sim \mathcal{N}(0, \eta \|w\|_\infty)$, where $\eta$ denotes the standard deviation scaling factor. We vary $\eta$ from 0.00 (0% noise; i.e., nominally evaluated model) to 0.10 (10% noise) in increments of 0.02 (2% noise at a time), as shown in the first column of Table 4. We sample stochastic parameter perturbations (for all of $f_\theta$'s parameters) 50 times per dataset example and average the model's predicted probabilities (i.e., we average the softmax predictions and then compare the averaged prediction with the ground-truth label).

Table 4: **Robustness to common image corruptions under stochastic parameter perturbations.** The table shows top-1 clean error, mCE and individual corruption error (averaged over all severities) for increasingly larger stochastic parameter perturbations to the *AdA* + AugMix classifier trained on Cifar-10.

| Noise (%) | Clean E | mCE | Noise | | | Blur | | | | Weather | | | | Digital | | | |
|---|---|---|---|---|---|---|---|---|---|---|---|---|---|---|---|---|---|
| | | | Gauss | Shot | Impulse | Defocus | Glass | Motion | Zoom | Snow | Frost | Fog | Bright | Contrast | Elastic | Pixel | JPEG |
| 0% (default) | **3.74%** | **8.80%** | **16**.4 | **12**.8 | **14**.4 | **5**.7 | **10**.9 | **7**.5 | **6**.3 | 7.7 | 6.7 | 6.6 | 3.9 | **4**.7 | 8.1 | 11.9 | 8.4 |
| 2% | 3.76% | 8.81% | 16.5 | 12.8 | 14.5 | 5.7 | 10.9 | 7.6 | 6.4 | **7**.7 | **6**.7 | **6**.6 | **3**.9 | 4.8 | **8**.1 | 11.8 | **8**.3 |
| 4% | 3.75% | 9.21% | 17.4 | 13.4 | 15.2 | 6.2 | 11.4 | 8.2 | 6.9 | 7.9 | 6.9 | 6.8 | 4.0 | 5.6 | 8.2 | **11**.7 | 8.3 |
| 6% | 4.28% | 10.49% | 19.5 | 15.1 | 17.2 | 7.3 | 12.8 | 9.8 | 8.5 | 8.6 | 7.7 | 7.8 | 4.4 | 8.5 | 9.0 | 12.4 | 8.9 |
| 8% | 6.01% | 14.56% | 24.5 | 19.4 | 22.9 | 11.2 | 17.9 | 14.7 | 12.6 | 11.6 | 10.1 | 11.3 | 6.4 | 15.8 | 12.3 | 16.2 | 11.4 |
| 10% | 17.75% | 31.34% | 40.7 | 35.5 | 44.5 | 26.9 | 40.9 | 34.5 | 30.2 | 29.0 | 22.2 | 24.9 | 19.2 | 34.2 | 28.2 | 32.8 | 26.2 |

We observe a gradual degradation in mCE and clean error when increasing the expected magnitude of parameter perturbations. Performance on mCE is maintained up to and including a noise level of 2%, and on clean error up to a noise level of 4% respectively.

## E Experimental setup details

### E.1 Training and evaluation details

For Cifar-10 we train pre-activation *ResNet50* (He et al., 2016b) models (as in Wong et al. (2020)); as in previous work (Hendrycks et al., 2020b) our models use $3 \times 3$ kernels for the first convolutional layer. We use a standard data augmentation consisting of padding by 4 pixels, randomly cropping back to $32 \times 32$ and randomly flipping left-to-right.

We train all robust classifiers ourselves using the same training strategy and architecture as for our own method (as described above); we use a perturbation radius of 0.5 for $\ell_2$ robust training and of $8/255$ for $\ell_\infty$ robust training. For both $\ell_2$ and $\ell_\infty$ robust training, we sweep over AWP step sizes of 0.005, 0.001, 0.0005 and 0.0001; we further evaluate the two models which obtained the best robust train accuracy (for $\ell_2$ this is the model which uses an AWP step size of 0.001, and for $\ell_\infty$ it is the model which uses one of 0.0005). Similarly, for SAM we train five models sweeping over the step size (0.2, 0.15, 0.1, 0.05, 0.01) and further evaluate the one obtaining the best robust train accuracy (which uses a step size of 0.05).

For ImageNet we use a standard *ResNet50* architecture for parity with previous works. We use a standard data augmentation, consisting of random left-to-right flips and random crops. Due to the increased computational requirements of training models with *AdA* (due to the adversarial training formulation) we resize each random crop (of size $224 \times 224$) to $128 \times 128$ using bilinear interpolation. We perform standard evaluation by using the central image crop resized to $224 \times 224$ on ImageNet, even though we train models on $128 \times 128$ crops.

All methods are implemented in the same codebase and use the same training strategy (in terms of standard data augmentation, learning rate schedule, optimizers, etc.).

We constrain capacity and use the *ResNet50* architecture for all our models for parity with previous works that tackle robustness to image corruptions (like Hendrycks et al., 2020a; Lee et al., 2020; Rusak et al., 2020), but note that larger models can achieve better mCE: e.g., Hendrycks et al. (2020a) train a very large model (RESNEXT-101 $32 \times 8d$; Xie et al., 2016) with AugMix and DeepAugment to obtain 44.5% mCE on IMAGENET-C.

## E.2  EVALUATING ROBUSTNESS AGAINST $\ell_p$-NORM BOUNDED PERTURBATIONS

For CIFAR-10, we follow the evaluation protocol of Gowal et al. (2020) and evaluate their robustness to $\ell_p$-norm bounded perturbations using a combination of two of the strongest adversarial attacks: AutoAttack (Croce & Hein, 2020) and MultiTargeted (Gowal et al., 2019b). Specifically, we run AutoPGD on the cross-entropy loss with 5 restarts and 100 steps; we also run AutoPGD on the difference of logits ratio loss with 5 restarts and 100 steps; finally, we run the MultiTargeted attack on the margin loss with 10 restarts and 200 steps. For IMAGENET we use a standard 100-step PGD attack with 10 restarts.

## E.3  OUTER MINIMIZATION

We minimize the corrupted adversarial risk by optimizing the classifier's parameters using stochastic gradient descent with Nesterov momentum (Polyak, 1964; Nesterov, 1983). For CIFAR-10 we train for 300 epochs with a batch size of 1024 and use a global weight decay of $10^{-4}$. For IMAGENET we train for 90 epochs with a batch size of 4096 and use a global weight decay of $5 \cdot 10^{-4}$. We use a cosine learning rate schedule (Loshchilov & Hutter, 2017), without restarts, with 5 warm-up epochs, with an initial learning rate of 0.1 which is decayed to 0 at the end of training. We scale all learning rates using the linear scaling rule of Goyal et al. (2017), i.e., effective $\text{LR} = \max(\text{LR} \times \text{batch size}/256, \text{LR})$. In Algorithm 1 the effective learning rate of the outer optimization is denoted by $\eta_f$.

## E.4  INNER MAXIMIZATION

Corrupted adversarial examples are obtained by maximizing the cross-entropy between the classifier's predictions on the corrupted inputs (by passing them through the corruption network) and their labels. We initialize the perturbations to the corruption network parameters randomly within the feasible region. We optimize the perturbations using iterated fast-gradient sign method (FGSM) steps (Goodfellow et al., 2015; Kurakin et al., 2016)[6]. We project the optimization iterates to stay within the feasible region. We use a step size equal to $1/4$ of the median perturbation radius over all parameter blocks (for each backbone network individually). In Algorithm 1 this step size is denoted by $\eta_c$. For CIFAR-10 we use 10 steps; for IMAGENET we use only 3 steps (due to the increased computational requirements) but we increase the step size by a factor of $10/3$ to compensate for the reduction in steps. When *AdA* is specified to use multiple backbones simultaneously (e.g., "*AdA (All)*" which uses four image-to-image models) each backbone is used for finding adversarial examples for an equal proportion of examples in each batch.

## E.5  COMBINING DATA AUGMENTATION METHODS

We view the process of combining data augmentation methods as a data pipeline, where the output from each stage is fed as input to the next stage. We first draw random samples either from the clean training dataset or from the DeepAugment-processed training set if DeepAugment is used, as in Hendrycks et al. (2020a). Then we apply standard data augmentation (random pad and crop for CIFAR-10, random left-right flip and resized crop for IMAGENET). When *AdA* is used, we apply it now in the pipeline, followed by the SSIM line-search procedure. When AugMix is used, we apply it as the final step in the data pipeline.

---

[6]We also experimented with Adam, SGD and normalized gradient ascent but we obtained the best results using FGSM.

### E.6 CORRUPTION NETWORKS

We train a separate *VQ-VAE* and *U-Net* model on the training set of each of CIFAR-10 and IMAGENET (for a total of four models). We train *VQ-VAE* models using the standard VQ-VAE loss (van den Oord et al., 2017, Eq. 3) with 128 (base) convolutional filters, a 2-layer residual stack (with 32 convolutional filters); we use a vector quantisation embedding dimension of 512, a commitment cost of 0.25 and use exponential moving averages to update the embedding vectors.

We train *U-Net* models on an image completion task; i.e. we zero-out 2-35% random pixels of each input image and train the *U-Net* to fill-in the deleted pixels. We use an image reconstruction loss which is the sum of the mean absolute pixel differences and the mean squared pixel differences (with the latter being scaled by $0.5 \cdot 0.1$). The U-Net architecture we use has a two-layer encoder (with 16 and 32 filters respectively) and a three-layer decoder (with 64, 32 and 16 filters respectively).

We train all four image-to-image models for 90 epochs, with a batch size of 1024 for IMAGENET and of 16384 for CIFAR-10. We use the Adam optimizer (Kingma & Ba, 2014) with an initial learning rate of 0.0003 for IMAGENET and 0.003 for CIFAR-10, which we decay throughout training using the linear scaling rule of Goyal et al. (2017).

For both *EDSR* and *CAE* we use the same model architectures and pre-trained weights as the ones used by DeepAugment (which are available online).

## F APPROXIMATE SSIM LINE-SEARCH PROCEDURE

The adversarial examples produced by *AdA* can sometimes become too severely corrupted (i.e., left-tail of densities in Figure 2 of the main manuscript). We guard against these unlikely events by using an efficient, approximate line-search procedure. We set a maximum threshold, denoted by $t$, on the SSIM distance between the clean example and the *AdA* adversarial example.

Denote by $x_\gamma$ the linear combination of the clean example, $x$, and the corrupted example output by *AdA*, $\hat{x}$:

$$x_\gamma = (1 - \gamma) \, x + \gamma \, \hat{x}.$$

When the deviation in SSIM between the clean and the corrupted example is greater than the threshold ($\mathrm{SSIM}(x, \hat{x}) > t$), we find a scalar $\gamma^* \in [0, 1]$ such that the deviation between the corrected example, $x_{\gamma}^*$, and the clean example, $x$, is $t$: $\mathrm{SSIM}(x, x_{\gamma}^*) = t$. We take 9 equally-spaced $\gamma$ values in $[0, 1]$ and evaluate the SSIM distance between the clean example and $x_\gamma$ for each considered $\gamma$. We fit a quadratic polynomial in $\gamma$ to all the pairs of $\gamma$ and the threshold-shifted SSIM deviation from the clean example $\mathrm{SSIM}(x, x_\gamma) - t$. We then find the roots of this polynominal, clip them to $[0, 1]$, and take the $\gamma$ closest to 1 as the desired $\gamma^*$. This corresponds to returning the most corrupted variant of $x$ along the ray between $x$ and $\hat{x}$ which obeys the SSIM threshold. The procedure is very efficient on accelerators (GPUs, TPUs) as it requires no iteration. It is approximate, however, because the quadratic polynomial can underfit.

In practice, we use a maximum SSIM threshold ($t$) of 0.3 for CIFAR-10 experiments and one of 0.7 for IMAGENET experiments.

### F.1 COMPUTING THE SSIM DISTANCE

For computing the SSIM distance we follow the original paper (Wang et al., 2004) and use an isotropic Gaussian weighting function of $11 \times 11$ with a 1.5 standard deviation (normalized), and regularization parameters $K_1 = 0.01$, $K_2 = 0.03$.

## G ADDITIONAL EXPERIMENTS AND COMPARISONS

### G.1 COMPARISON TO PREVIOUS WORK ON CIFAR-10

Perceptual Adversarial Training (PAT) (Laidlaw et al., 2021) proposes an adversarial training method based on bounding a neural perceptual distance (LPIPS). Appendix G (Table 10) of the PAT article shows the performance of various models on common image corruptions. However, performance

is summarized using *relative* mCE, whereas we use *absolute* mCE throughout. The authors kindly provided us[7] with the raw corruption errors of their models at each severity and we reproduce their results in (the top half of) Table 5. We observe that PAT has overall lower robustness to common image corruptions (best variant obtains 23.54% mCE) than AugMix (10.90% mCE) and than our best *AdA*-trained model (7.83% mCE).

PAT however, performs very well against other adversarial attacks[8], including $\ell_p$-norm bounded perturbations. The best PAT model obtains 28.7% robust accuracy against $\ell_\infty$ attacks ($\epsilon = 8/255$) and 33.3% on $\ell_2$ attacks ($\epsilon = 1$) while our best *AdA*-variant obtains less robust accuracy in each case (0.99% against $\ell_2$ attacks and 13.88% against $\ell_\infty$ attacks with $\epsilon = 4/255$). This difference in performance against $\ell_p$-norm attacks is not surprising, as PAT addresses robustness to pixel-level attacks (i.e., it manipulates image pixels directly); whereas *AdA* applies adversarial perturbations to the corruption function parameters (and not to the image pixels directly).

In similar spirit to PAT, Kireev et al. (2021) introduce an efficient relaxation of adversarial training with LPIPS as the distance metric. Their best model, with a smaller architecture, RESNET18, obtains 11.47% mCE on CIFAR-10-C. The authors of Kireev et al. (2021) also show that models trained adversarially against $\ell_p$-norm bounded perturbations can act as a strong baseline for robustness to common image corruptions.

The strongest known[9] adversarially trained model against $\ell_p$-norm bounded perturbations on common image corruptions is that of Gowal et al. (2020) which obtains 12.32% mCE (training against $\ell_2$-norm bounded perturbations with $\epsilon = 0.5$ while using extra-data).

In Wong & Kolter (2021a), the authors first train generative models to represent image corruptions by feeding them a subset of the common image corruptions (from CIFAR-10-C). In a second stage, they train classifiers using samples (adversarial or random) coming from these pre-trained generative models. Despite having this additional knowledge of the test set corruptions, the robust classifiers they train only achieve between 9.5% and 9.7 mCE%.

At the time of our initial submission, to the best of our knowledge, our best performing model (*AdA (EDSR)* coupled with AugMix and DeepAugment) was more robust to common image corruptions on CIFAR-10 than all previous methods, obtaining a new state-of-the-art mCE of 7.83%.

After our initial submission, Diffenderfer et al. (2021) obtained a lower mCE of 7.22% through an adaptive test-time ensemble of six compressed models (where each model had been pruned to 95% sparsity). Diffenderfer et al. (2021) also show that carefully compressing (i.e., pruning and optionally quantizing) neural networks can yield sparser models with similar clean (test) accuracy and similar or better robustness to corruptions.

Table 5: **Performance of Perceptual Adversarial Training on common image corruptions.** The table lists the performance of *ResNet50* models trained using Perceptual Adversarial Training by the original authors of Laidlaw et al. (2021) on common image corruptions and two of our *AdA*-trained models. The table shows clean error, mean corruption error on CIFAR-10-C and individual corruption errors for each corruption type (averaged across all severities). "PAT-self" denotes the case where the same model is used for classification as well as for computing the LPIPS distance, while "PAT-AlexNet" denotes the case where the LPIPS distance is computed using a pre-trained CIFAR-10 AlexNet (Krizhevsky et al., 2012) classifier.

| SETUP | CLEAN E | mCE | NOISE | | | BLUR | | | | WEATHER | | | | DIGITAL | | | |
|---|---|---|---|---|---|---|---|---|---|---|---|---|---|---|---|---|---|
| | | | GAUSS | SHOT | IMPULSE | DEFOCUS | GLASS | MOTION | ZOOM | SNOW | FROST | FOG | BRIGHT | CONTRAST | ELASTIC | PIXEL | JPEG |
| PAT MODELS (LAIDLAW ET AL., 2021) | | | | | | | | | | | | | | | | | |
| Nominal Training | 5.20% | 25.80% | 54.0 | 42.3 | 38.8 | 16.5 | 50.0 | 21.9 | 21.1 | 18.3 | 24.0 | 10.5 | 6.1 | 16.0 | 17.6 | 28.2 | 20.7 |
| Adversarial Training $\ell_\infty$ | 13.20% | 20.71% | 18.3 | 17.0 | 22.5 | 16.9 | 19.8 | 20.4 | 17.5 | 17.0 | 18.2 | 32.9 | 13.7 | 47.9 | 18.1 | 15.0 | 15.3 |
| Adversarial Training $\ell_2$ | 15.00% | 21.83% | 18.4 | 17.5 | 21.4 | 18.3 | 20.2 | 21.0 | 18.7 | 19.2 | 20.5 | 35.9 | 16.0 | 47.1 | 20.0 | 16.6 | 16.5 |
| PAT-self | 17.60% | 23.54% | 22.5 | 21.1 | 25.7 | 20.0 | 22.5 | 22.2 | 20.3 | 23.7 | 23.6 | 33.5 | 19.8 | 38.8 | 21.7 | 18.7 | 19.1 |
| PAT-AlexNet | 28.40% | 34.25% | 33.2 | 31.9 | 36.3 | 30.9 | 34.3 | 33.0 | 32.0 | 33.9 | 35.0 | 43.7 | 30.5 | 46.5 | 32.6 | 29.9 | 29.8 |
| SELECTION OF AdA MODELS (OURS) | | | | | | | | | | | | | | | | | |
| *AdA (EDSR)* | 6.63% | 15.47% | 27.3 | 21.5 | 31.7 | 11.8 | 20.0 | 13.9 | 14.1 | 12.9 | 11.5 | 11.8 | 7.0 | 14.9 | 12.7 | 9.4 | 11.5 |
| *AdA (EDSR)* + DeepAugment + AugMix | 5.07% | 7.83% | 8.8 | 7.8 | 11.2 | 5.9 | 10.7 | 7.3 | 6.5 | 8.5 | 6.7 | 8.7 | 5.2 | 6.2 | 8.5 | 7.7 | 7.8 |

---

[7]Personal communication.

[8]See Table 2 of Laidlaw et al. (2021) for full details.

[9]See the ROBUSTBENCH (Croce et al., 2020) leaderboard: https://robustbench.github.io.

## G.2 NUMBER OF INNER OPTIMIZATION STEPS

We show the effect of changing the number of inner optimization steps (i.e. the number of PGD steps for finding adversarial examples) for *AdA (EDSR)* on CIFAR-10 in Table 6. We sweep a number of PGD steps from 0 to 10 where 10 is the default for CIFAR-10. As described in Appendix E.4, we compensate for the decrease in number of PGD steps (from 10) by proportionally scaling up the step size (i.e. effective $\eta_c = \frac{\eta_c \cdot 10}{\text{num\_steps}}$). Zero (0) PGD steps correspond to only performing the random initialization of perturbations to the corruption network's parameters (i.e. lines 5-6 in Algorithm 1). We observe that performance on common image corruptions in aggregate (mCE) and for individual groups, in general, increases with the number of inner optimization PGD steps. We note that adversarial training appears to be *necessary* for training the most robust models, and randomly sampling the parameter perturbations is not sufficient.

Table 6: **Effect of number of inner optimization steps on robustness to common image corruptions (CIFAR-10-C).** The table shows the performance of the *AdA (EDSR)* model while varying the number of inner optimization steps ($M$). The table shows the clean top-1 error, the mean corruption error on CIFAR-10-C and on individual corruptions (averaged across all severities).

| | | | NOISE | | | BLUR | | | | WEATHER | | | | DIGITAL | | | |
|---|---|---|---|---|---|---|---|---|---|---|---|---|---|---|---|---|---|
| NUM. PGD STEPS | CLEAN E | MCE | GAUSS | SHOT | IMPULSE | DEFOCUS | GLASS | MOTION | ZOOM | SNOW | FROST | FOG | BRIGHT | CONTRAST | ELASTIC | PIXEL | JPEG |
| 0 (i.e. only random init.) | 8.99 | 28.35 | 43.4 | 34.1 | 35.3 | 22.6 | 49.9 | 30.0 | 29.1 | 22.9 | 24.6 | 16.3 | 10.5 | 28.2 | 23.6 | 31.0 | 23.6 |
| 2 | 7.87 | 17.91 | 21.5 | 18.3 | 28.5 | 18.8 | 27.5 | 17.7 | 23.7 | 16.0 | 15.4 | 11.3 | 8.5 | 12.6 | 18.6 | 14.0 | 16.4 |
| 4 | 6.99 | **14.91** | 26.2 | 21.0 | 25.8 | 12.0 | 18.6 | 12.9 | 14.7 | 12.8 | 11.7 | 11.6 | 7.4 | 13.7 | 13.4 | 9.8 | 12.0 |
| 6 | 6.83 | 15.94 | 30.2 | 23.7 | 29.8 | 12.1 | 19.2 | 13.7 | 15.4 | 13.0 | 12.2 | 12.8 | 7.2 | 15.1 | 13.3 | 9.5 | 11.9 |
| 8 | 6.80 | 15.66 | 28.0 | 22.4 | 30.5 | 12.6 | 18.4 | 14.3 | 15.0 | 12.7 | 11.9 | 12.2 | 7.3 | 15.1 | 13.0 | 9.7 | 12.0 |
| 10 | **6.63** | 15.47 | 27.3 | 21.5 | 31.7 | 11.8 | 20.0 | 13.9 | 14.1 | 12.9 | 11.5 | 11.8 | 7.0 | 14.9 | 12.7 | 9.4 | 11.5 |

## G.3 PERTURBATION RADIUS

We show the effect of changing the corruption network parameters perturbation radius in *AdA* on CIFAR-10 in Table 7. We perform a sweep on the perturbation radius by scaling the radius ($\nu = 0.015$) of the best performing model, *AdA (EDSR)* + DeepAugment + AugMix (from Table 1 in the main manuscript), by $\{0.5, 0.75, 1.0, 1.25, 1.5\}$. We observe that the robustness performance varies minimally across a small range of perturbation radii.

Table 7: **Effect of perturbation radius on robustness to common image corruptions (CIFAR-10-C).** The table shows the performance of the best *AdA*-combination from Table 1 from the main manuscript (*AdA* (*EDSR*) + DeepAugment + AugMix) while varying the perturbation radius ($\nu$). The table shows mean corruption error on CIFAR-10-C and individual corruption errors for each corruption type (averaged across all severities).

| | | NOISE | | | BLUR | | | | WEATHER | | | | DIGITAL | | | |
|---|---|---|---|---|---|---|---|---|---|---|---|---|---|---|---|---|
| PERTURBATION RADIUS | MCE | GAUSS | SHOT | IMPULSE | DEFOCUS | GLASS | MOTION | ZOOM | SNOW | FROST | FOG | BRIGHT | CONTRAST | ELASTIC | PIXEL | JPEG |
| $\nu = 0.0225$ | 8.10% | 9.4 | 8.1 | 11.5 | 6.2 | 11.4 | 7.5 | 6.6 | 8.8 | 6.9 | **8.5** | 5.5 | 6.4 | 8.8 | 7.8 | 8.1 |
| $\nu = 0.01875$ | 8.06% | 8.8 | 7.9 | 11.5 | 6.1 | 11.2 | 7.6 | 6.5 | 9.0 | 7.0 | 8.7 | 5.5 | 6.5 | 8.8 | 7.6 | 8.1 |
| $\nu = 0.015$ (default) | **7.83%** | 8.8 | 7.8 | 11.2 | **5.9** | **10.7** | 7.3 | 6.5 | **8.5** | **6.7** | 8.7 | **5.2** | **6.2** | **8.5** | 7.7 | **7.8** |
| $\nu = 0.01125$ | 8.41% | 9.1 | 8.3 | 12.1 | 6.5 | 11.3 | 8.0 | 6.8 | 9.2 | 7.2 | 9.2 | 5.8 | 6.8 | 9.2 | 8.1 | 8.5 |
| $\nu = 0.0075$ | 7.99% | 8.9 | 8.0 | **11.0** | 6.2 | 10.7 | 7.7 | **6.2** | 9.0 | 6.9 | 8.6 | 5.5 | 6.9 | 8.7 | **7.6** | 8.1 |

## G.4 PERFORMANCE ON IMAGE CORRUPTIONS THROUGH TRAINING

We visualize the performance of *AdA* trained models (best *AdA*-combination from Tables 1 and 2 from the main manuscript) during training in Figure 4. Due to adversarial training, we expect the performance on each of the *-C corruptions to improve as training progresses, and this is indeed what we observe. On both datasets, the *AdA*-trained classifiers perform consistently best on *Brightness*, especially at the beginning of training. On IMAGENET performance increases more slowly on the *Blur*-type corruptions than on all others.

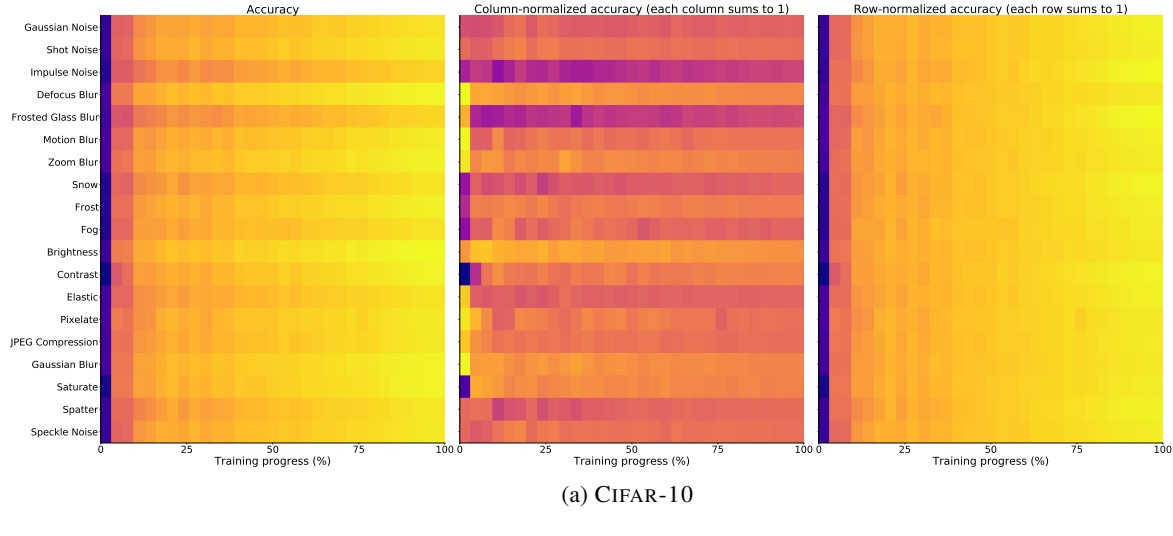

(a) CIFAR-10

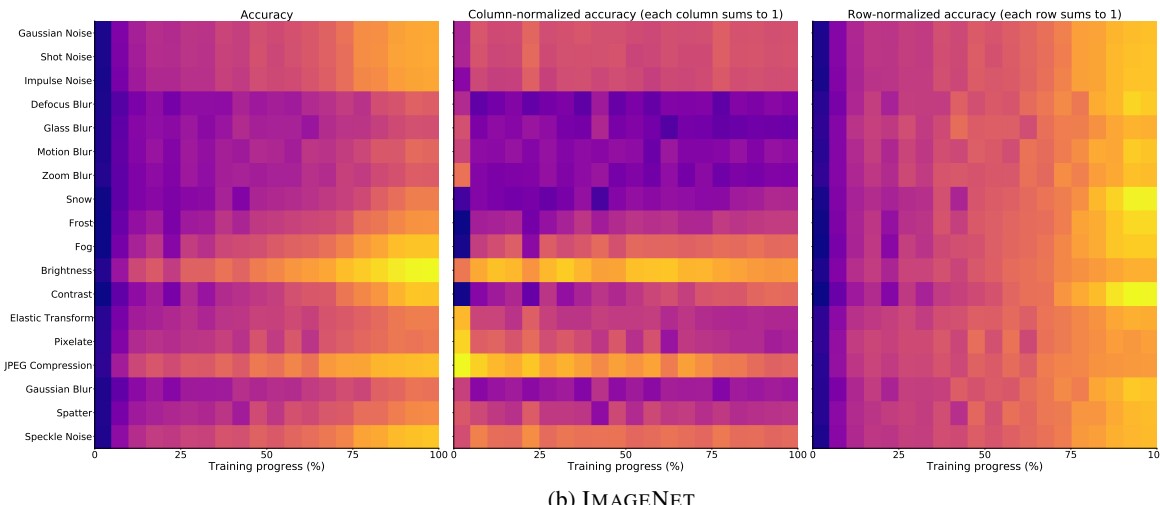

(b) IMAGENET

Figure 4: **Performance on image corruptions through training.** These plots visualize the performance of the best *AdA* combination on each of the common and extra *-C corruptions as training progresses. Each individual rectangle plots top-1 accuracy. Brighter is better. The accuracies are visualized raw (plots to the left), normalized over the columns (middle plots) or over the rows (plots to the right). Normalizing over the columns visualizes which corruption's performance is best at that point in training. Normalizing over the rows visualizes at which stage the classifier performs best on a given corruption.

## G.5 EXTENDED RESULTS ON INDIVIDUAL IMAGE CORRUPTIONS

We provide supplemental details to Tables 1 and 2 (from the main manuscript) on individual corruption types in Tables 8 and 9, respectively.

Table 8: **CIFAR-10: Extended results on robustness to common image corruptions.** The table shows mean corruption error on CIFAR-10-C and individual corruption errors for each corruption type (averaged across all 5 severities).

| SETUP | mCE | NOISE | | | BLUR | | | | WEATHER | | | | DIGITAL | | | |
|---|---|---|---|---|---|---|---|---|---|---|---|---|---|---|---|---|
| | | GAUSS | SHOT | IMPULSE | DEFOCUS | GLASS | MOTION | ZOOM | SNOW | FROST | FOG | BRIGHT | CONTRAST | ELASTIC | PIXEL | JPEG |
| *WITHOUT ADDITIONAL DATA AUGMENTATION:* | | | | | | | | | | | | | | | | |
| Nominal | 29.37 | $49_8$ | $39_5$ | $39_5$ | $20_1$ | $53_2$ | $27_8$ | $26_8$ | $24_1$ | $27_8$ | $17_0$ | $10_1$ | $28_6$ | $21_2$ | $31_9$ | $23_0$ |
| *AdA (U-Net)* | 23.11 | $40_4$ | $33_1$ | $35_9$ | $16_8$ | $41_5$ | $18_9$ | $21_2$ | $19_8$ | $20_9$ | $13_5$ | $7_9$ | $17_2$ | $17_7$ | $23_6$ | $18_2$ |
| *AdA (VQ-VAE)* | 26.77 | $25_1$ | $24_3$ | $28_9$ | $23_8$ | $26_8$ | $25_9$ | $25_1$ | $27_7$ | $24_9$ | $37_1$ | $22_6$ | $37_1$ | $26_7$ | $22_8$ | $22_6$ |
| *AdA (EDSR)* | **15.47** | $27_3$ | $21_5$ | $31_7$ | **$11_8$** | **$20_0$** | **$13_9$** | **$14_1$** | **$12_9$** | **$11_5$** | **$11_8$** | **$7_0$** | **$14_9$** | **$12_7$** | **$9_4$** | **$11_5$** |
| *AdA (CAE)* | 29.15 | $26_5$ | $26_3$ | $28_2$ | $25_2$ | $28_0$ | $27_1$ | $26_4$ | $31_3$ | $28_5$ | $40_9$ | $25_9$ | $43_0$ | $29_1$ | $25_5$ | $25_4$ |
| *AdA (All)* | 18.49 | **$20_6$** | **$18_6$** | **$23_4$** | $16_3$ | $23_7$ | $18_6$ | $18_9$ | $18_6$ | $17_4$ | $19_7$ | $12_2$ | $20_8$ | $18_2$ | $15_1$ | $15_4$ |
| *AT ($\ell_\infty$)* | 23.64 | $18_6$ | $17_5$ | $25_6$ | $18_6$ | $20_2$ | $22_8$ | $19_4$ | $20_8$ | $24_9$ | $40_1$ | $17_7$ | $56_2$ | $20_0$ | $15_8$ | $16_2$ |
| *AT ($\ell_2$)* | **17.42** | $12_8$ | $11_7$ | $17_2$ | $13_5$ | $16_0$ | $17_2$ | $14_2$ | $15_7$ | $16_4$ | $33_9$ | $11_3$ | $44_8$ | $14_8$ | $10_8$ | $10_8$ |
| *TRADES ($\ell_\infty$)* | 24.72 | $19_8$ | $18_7$ | $25_4$ | $20_0$ | $21_5$ | $23_6$ | $20_8$ | $22_2$ | $26_5$ | $40_9$ | $19_4$ | $56_2$ | $21_2$ | $17_2$ | $17_4$ |
| *TRADES ($\ell_2$)* | 18.08 | $13_0$ | $12_3$ | $16_9$ | $14_2$ | $16_1$ | $17_6$ | $14_7$ | $16_9$ | $17_9$ | $34_4$ | $12_7$ | $45_3$ | $15_6$ | $11_9$ | $11_7$ |
| *AWP ($\ell_\infty$)* | 25.01 | $20_4$ | $19_2$ | $25_7$ | $20_1$ | $21_7$ | $23_5$ | $21_0$ | $22_6$ | $27_6$ | $40_4$ | $20_2$ | $56_1$ | $21_4$ | $17_7$ | $17_8$ |
| *AWP ($\ell_2$)* | 18.79 | $14_1$ | $13_1$ | $17_6$ | $14_5$ | $16_4$ | $17_5$ | $15_2$ | $17_8$ | $19_3$ | $35_3$ | $13_7$ | $46_7$ | $15_9$ | $12_5$ | $12_4$ |
| *SAM* | 24.59 | $58_5$ | $43_8$ | $46_0$ | $14_7$ | $46_8$ | $19_4$ | $18_0$ | **$15_2$** | $19_8$ | **$11_0$** | **$5_2$** | **$20_5$** | **$13_0$** | $20_5$ | $16_4$ |
| *WITH AUGMIX:* | | | | | | | | | | | | | | | | |
| Nominal | 12.26 | $27_3$ | $20_2$ | **$15_9$** | $7_7$ | $14_4$ | $10_7$ | $9_4$ | $9_2$ | $9_4$ | **$7_2$** | $4_5$ | $6_0$ | $9_8$ | $20_6$ | $11_6$ |
| *AdA (U-Net)* | 12.02 | $24_6$ | $18_8$ | $18_2$ | $7_2$ | $16_3$ | $9_1$ | $8_7$ | $10_1$ | $9_9$ | $7_8$ | $4_4$ | $5_3$ | $10_0$ | $19_4$ | $10_3$ |
| *AdA (VQ-VAE)* | 20.85 | $22_4$ | $21_2$ | $25_6$ | $19_0$ | $24_3$ | $19_4$ | $19_1$ | $22_0$ | $20_0$ | $24_4$ | $16_7$ | $20_2$ | $20_3$ | $19_9$ | $18_3$ |
| *AdA (EDSR)* | **9.40** | **$17_3$** | **$13_7$** | $16_5$ | **$6_2$** | **$11_9$** | **$8_1$** | **$7_1$** | **$8_5$** | **$7_6$** | $7_6$ | **$4_2$** | **$5_3$** | **$8_7$** | **$10_3$** | **$8_0$** |
| *AdA (CAE)* | 20.20 | $19_3$ | $18_5$ | $22_6$ | $17_1$ | $22_9$ | $19_6$ | $18_0$ | $22_6$ | $20_0$ | $26_0$ | $16_6$ | $21_4$ | $20_5$ | $20_1$ | $17_8$ |
| *AdA (All)* | 14.12 | $18_8$ | $16_3$ | $17_9$ | $12_8$ | $17_0$ | $13_8$ | $14_1$ | $13_8$ | $12_5$ | $12_9$ | $8_4$ | $10_6$ | $13_6$ | $16_6$ | $12_8$ |
| *WITH DEEPAUGMENT:* | | | | | | | | | | | | | | | | |
| Nominal | **11.94** | $12_5$ | $11_1$ | $15_0$ | $8_6$ | $18_7$ | $11_7$ | $9_9$ | $12_7$ | $10_5$ | $10_1$ | $7_9$ | $10_9$ | **$13_0$** | $14_3$ | $12_1$ |
| *AdA (U-Net)* | 13.09 | $13_7$ | $12_6$ | $19_5$ | $10_0$ | $19_4$ | $12_1$ | $11_1$ | $14_4$ | $11_2$ | $13_0$ | $8_9$ | **$9_8$** | $14_1$ | $13_8$ | $12_8$ |
| *AdA (VQ-VAE)* | 26.35 | $29_8$ | $28_4$ | $32_9$ | $22_7$ | $26_6$ | $24_7$ | $23_3$ | $27_8$ | $25_2$ | $32_6$ | $22_7$ | $26_5$ | $26_1$ | $23_1$ | $23_0$ |
| *AdA (EDSR)* | 12.37 | $13_0$ | $12_0$ | $17_7$ | $10_0$ | **$16_1$** | $11_8$ | $10_7$ | $13_3$ | $10_9$ | $13_2$ | $9_1$ | $11_4$ | $13_6$ | **$10_8$** | $11_9$ |
| *AdA (CAE)* | 27.98 | $26_9$ | $26_6$ | $28_4$ | $25_8$ | $28_4$ | $27_1$ | $26_6$ | $30_3$ | $27_4$ | $36_3$ | $25_2$ | $29_5$ | $29_2$ | $25_8$ | $26_1$ |
| *AdA (All)* | 21.70 | $21_5$ | $20_6$ | $23_9$ | $20_1$ | $24_9$ | $22_4$ | $21_4$ | $23_4$ | $20_4$ | $25_7$ | $18_1$ | $20_1$ | $23_1$ | $19_9$ | $19_9$ |
| *WITH AUGMIX & DEEPAUGMENT:* | | | | | | | | | | | | | | | | |
| Nominal | 7.99 | $8_9$ | $7_5$ | $10_0$ | $5_4$ | $11_4$ | $7_7$ | **$6_1$** | $8_5$ | $6_5$ | $7_2$ | $4_9$ | $6_3$ | $8_2$ | $12_3$ | $8_8$ |
| *AdA (U-Net)* | 8.63 | $9_4$ | $8_2$ | $11_9$ | $5_9$ | $12_6$ | $7_7$ | $6_6$ | $9_2$ | $6_8$ | $8_6$ | $5_0$ | **$5_9$** | $9_1$ | $14_0$ | $8_5$ |
| *AdA (VQ-VAE)* | 25.17 | $26_1$ | $25_5$ | $29_2$ | $22_3$ | $27_6$ | $23_8$ | $22_5$ | $27_1$ | $25_4$ | $30_1$ | $22_9$ | $23_6$ | $25_0$ | $23_0$ | $23_2$ |
| *AdA (EDSR)* | **7.83** | **$8_8$** | $7_8$ | $11_2$ | $5_9$ | **$10_7$** | $7_3$ | $6_5$ | $8_5$ | $6_7$ | $8_7$ | $5_2$ | $6_2$ | $8_5$ | **$7_7$** | **$7_8$** |
| *AdA (CAE)* | 20.09 | $19_2$ | $18_6$ | $21_2$ | $17_3$ | $23_1$ | $20_4$ | $18_1$ | $22_6$ | $20_1$ | $26_4$ | $16_8$ | $19_8$ | $20_4$ | $19_5$ | $18_0$ |
| *AdA (All)* | 11.72 | $12_0$ | $11_1$ | $14_3$ | $10_2$ | $14_7$ | $11_5$ | $10_6$ | $13_0$ | $10_5$ | $13_7$ | $8_3$ | $9_6$ | $12_2$ | $12_2$ | $11_7$ |

Table 9: **IMAGENET (128×128): Extended results on robustness to common image corruptions.** The table shows mean corruption error on IMAGENET-C and individual corruption errors for each corruption type (averaged across all 5 severities).

| SETUP | MCE | NOISE | | | BLUR | | | | WEATHER | | | | DIGITAL | | | |
|---|---|---|---|---|---|---|---|---|---|---|---|---|---|---|---|---|
| | | GAUSS | SHOT | IMPULSE | DEFOCUS | GLASS | MOTION | ZOOM | SNOW | FROST | FOG | BRIGHT | CONTRAST | ELASTIC | PIXEL | JPEG |
| **WITHOUT ADDITIONAL DATA AUGMENTATION:** | | | | | | | | | | | | | | | | |
| Nominal | 82.40 | **73.2** | 74.8 | 80.1 | 71.4 | 78.5 | 70.4 | **65.4** | 66.7 | 63.6 | 46.1 | 35.8 | 63.3 | 57.4 | 72.4 | 55.9 |
| AdA (U-Net) | 83.51 | 81.2 | 81.7 | 92.8 | 79.0 | 86.2 | 73.0 | 74.4 | 61.3 | 60.7 | **41.6** | 34.2 | **48.4** | 65.9 | 51.3 | 58.7 |
| AdA (VQ-VAE) | 78.26 | 73.7 | **71.9** | **72.4** | 71.5 | **66.1** | **62.6** | 66.9 | 67.0 | 63.8 | 60.1 | 44.3 | 64.7 | **54.7** | 44.9 | **43.6** |
| AdA (EDSR) | 79.59 | 74.9 | 75.4 | 78.0 | 70.1 | 75.0 | 67.9 | 66.3 | 62.9 | 60.0 | 44.4 | 33.6 | 56.7 | 57.4 | 64.8 | 54.9 |
| AdA (CAE) | 86.44 | 89.6 | 89.2 | 94.4 | 75.8 | 70.3 | 68.3 | 66.8 | 75.5 | 68.3 | 56.4 | 45.1 | 65.3 | 55.1 | 62.1 | 47.8 |
| AdA (All) | **75.03** | 76.3 | 76.4 | 83.1 | **67.8** | 72.9 | 65.8 | 67.5 | **59.0** | **55.7** | 41.9 | **31.7** | 48.5 | 57.4 | **37.8** | 50.8 |
| **WITH AUGMIX:** | | | | | | | | | | | | | | | | |
| Nominal | 77.12 | 70.4 | 71.6 | 75.6 | 67.2 | 75.1 | 64.0 | **59.0** | 60.9 | 61.1 | 37.9 | 34.7 | 46.9 | 57.5 | 71.4 | 55.9 |
| AdA (U-Net) | 77.87 | 71.2 | 72.1 | 86.5 | 74.0 | 81.4 | 66.1 | 64.6 | **57.1** | 58.1 | **37.5** | 32.9 | 41.4 | 65.4 | 51.0 | 59.3 |
| AdA (VQ-VAE) | 73.41 | 75.1 | 73.7 | **69.3** | 68.3 | 69.5 | 61.1 | 60.0 | 61.8 | 56.3 | 41.7 | 38.1 | 44.0 | 56.6 | **44.5** | 48.1 |
| AdA (EDSR) | 73.59 | 67.3 | 68.2 | 71.7 | **65.6** | 73.8 | 62.8 | 61.1 | 61.3 | 58.7 | 39.6 | **31.9** | 43.2 | 55.6 | 60.8 | 48.5 |
| AdA (CAE) | 80.85 | 87.4 | 86.6 | 91.2 | 66.7 | **65.7** | 63.0 | 63.0 | 71.3 | 66.0 | 50.6 | 42.2 | 52.2 | **53.5** | 59.1 | **44.6** |
| AdA (All) | **72.27** | **66.9** | **66.6** | 71.4 | 67.7 | 71.5 | **60.4** | 60.6 | 58.2 | **55.3** | 39.7 | 33.0 | **40.5** | 58.8 | 46.4 | 55.2 |
| **WITH DEEPAUGMENT:** | | | | | | | | | | | | | | | | |
| Nominal | 73.04 | 52.1 | 52.1 | 53.3 | 60.5 | 69.0 | 65.9 | **63.3** | 57.7 | 55.4 | 43.8 | 33.8 | 58.2 | 56.2 | 66.8 | 63.8 |
| AdA (U-Net) | 75.03 | 54.6 | 56.3 | 60.0 | 67.5 | 77.5 | 66.7 | 72.0 | 55.9 | 54.8 | **40.7** | 34.8 | **42.1** | 62.6 | 47.0 | 79.5 |
| AdA (VQ-VAE) | 69.15 | 56.7 | 58.7 | 56.8 | **57.8** | **63.5** | 61.6 | 68.7 | 57.2 | 54.4 | 46.2 | 37.2 | 49.8 | 57.4 | 37.3 | **49.3** |
| AdA (EDSR) | 65.62 | 48.2 | 49.7 | 48.8 | 58.4 | 66.3 | 62.3 | 63.4 | 56.8 | 53.4 | 42.6 | **31.0** | 50.9 | **53.3** | **33.7** | 51.7 |
| AdA (CAE) | 77.55 | 64.2 | 66.9 | 69.6 | 64.3 | 73.8 | 66.9 | 66.7 | 56.5 | 55.2 | 44.6 | 36.7 | 50.2 | 61.5 | 39.3 | 87.3 |
| AdA (All) | **65.54** | **47.1** | **48.3** | **47.9** | 59.2 | 66.4 | **60.4** | 65.2 | **54.2** | **51.3** | 42.5 | 32.5 | 44.2 | 55.4 | 37.4 | 54.2 |
| **WITH AUGMIX & DEEPAUGMENT:** | | | | | | | | | | | | | | | | |
| Nominal | 70.05 | 50.5 | 50.0 | 51.5 | 58.0 | 67.8 | 60.1 | 58.7 | 54.5 | 53.3 | **37.5** | 33.4 | 43.6 | 56.8 | 64.2 | 72.0 |
| AdA (U-Net) | 71.64 | 53.9 | 54.5 | 57.8 | 62.3 | 73.3 | 63.1 | 65.9 | 53.7 | 53.2 | 38.1 | 34.2 | **38.0** | 61.8 | 45.8 | 76.0 |
| AdA (VQ-VAE) | 69.02 | 54.2 | 54.6 | 54.0 | 56.8 | 62.9 | 59.2 | 66.2 | 58.3 | 56.0 | 47.7 | 42.0 | 47.6 | 59.5 | 42.0 | 46.7 |
| AdA (EDSR) | 64.31 | **47.6** | **48.1** | 48.5 | **55.3** | 64.5 | 56.8 | **56.8** | 55.0 | 52.0 | 39.8 | **32.1** | 41.2 | **54.3** | **35.1** | 62.2 |
| AdA (CAE) | 67.89 | 55.8 | 56.1 | 55.2 | 55.6 | **62.1** | 59.4 | 60.9 | 57.2 | 55.5 | 45.8 | 39.1 | 45.7 | 59.7 | 38.5 | 48.5 |
| AdA (All) | **62.90** | 48.1 | 48.4 | 48.9 | 55.4 | 63.7 | **54.7** | 60.2 | **53.3** | **51.1** | 41.0 | 34.1 | 39.6 | 56.1 | 36.3 | **45.4** |

