# OpenReview forum: "Defending Against Image Corruptions Through Adversarial Augmentations"
_ICLR.cc/2022/Conference — ICLR 2022 Poster_

### Official Review · Reviewer_31sh · 2021-10-25

**Correctness:** 3
**Technical Novelty And Significance:** 3
**Empirical Novelty And Significance:** 3
**Recommendation:** 6
**Confidence:** 4

**Main Review:**

Strength:
* the proposed Adversarial Augment (AdA) is a novel contribution that comes with the benefit of being a general purpose approach. That is: in contrast to other augmentation operations, it could be applied on any type of data if there exist a suitable pretrained corruption network for the type of data
 * Strong empirical results, establishing a new state-of-the-art for CIFAR-10-C when combined with other augmentations. Moreover, strong results are also shown for ImageNet-C
* paper is well written and experiments are well structured and presented (Table 1 and Table 2)

Weaknesses:
 * Related work on sharpness-aware minimization (SAM; Foret et al. "Sharpness-aware Minimization for Efficiently Improving Generalization", ICLR 2021) and adversarial weight perturbation (AWP, Wu et al., "Adversarial Weight Perturbation Helps Robust Generalization", NeurIPS 2020) should be discussed. The main difference seems to be that the proposed method utilizes an auxiliary image-to-image model while SAM and AWP generate adversarial weight perturbations directly in parameter space of the classifier. Since the setting of SAM/AWP is more generic (no requirement of pretrained vorruption network), it would be important to clearly show an advantage of AdA over these works.
 * Assumption 1 (Corruption coverage), which is the basis for the theoretical considerations, is not plausible. The authors show in Figure 3 that the first part of the assumption ("$\beta$ has support as least as broad as $\alpha$") is approximately fulfilled. However, the second part ("any corruption function sampled from $\beta$ leaves the ground-truth label unchanged") seems highly unlikely: because of the first part of the assumption, the same property would have to hold for $\alpha$. Prior work has established that images corresponding to different classes can have small $\ell_p$ distance. The support of corruptions such as Gaussian noise-severity 5 on an image thus very likely covers parts of the input space that belong to other classes (even though sampling such Gaussian noise is highly unlikely).
* since the general method is not restricted to image classifiers or even image-based tasks, including results on other types of data would strengthen the paper.
* discussing the computational and memory overhead more transparently would be desirable. As I see it, the additional inner maximization should increase runtime by a factor of M (M=10 on CIFAR10). Moreover, the additional corruption network needs to fit in memory, increasing the required memory. This should be part of a discussion of pros and cons of the methods.
* AdA is by design a deterministic procedure that should generate the same augmented version of the input for a given model when applied repeatedly. A desirable property of data augmentation is diversity, that is: the same input is augmented differently in every application of the augmentation operator. Could the authors comment on why AdA is defined as a deterministic procedure? (I acknowledge that due to randomness in optimization and changing model parameters, the input augmentation during training is not static but still, diversity is more an accidental byproduct and not an intrinsic property of AdA).

Additional comments:
 * in the current form, all layers of the corruption network are perturbed. Would it make sense to perturb only a subset of the layers - for instance the first layers that presumable focus more on appearance than semantics?
 * shouldn't $\eta_c$ in line 8 of Algorithm 1 depend on the layer index i - because it needs to depend on the {2, \phi} norm ?  Also, shouldn't Algorithm 1 use the {2, \phi} norm?
* Having error bars in Figure 3 that go beyond 1 does not make sense since SSIM cannot be larger than 1. This indicates that a different way of plotting the empirical distribution of SSIM scores would be better suited for the data.


**Summary Of The Paper:**

The main contribution of the paper is the proposed AdversarialAugment (AdA) method. AdA generates augmented versions of an input by passing the input through an corruption network (such as a pretrained image-to-image model) while adding a worst-case perturbation to the _weights_ of this pretrained network. The paper thus aims at using worst-case perturbations for increasing average case out-of-distribution generalization such as common corruption robustness. Tuning the weight perturbation radius of the corruption network is done based on controlling the SSIM of the augmented to the clean input.

The paper provides theoretical considerations that state assumptions under which AdA is well-behaved (converges) and how it is related to prior work such as DeepAugment.

The paper presents an extensive evaluation on common corruption benchmarks (CIFAR10-C and ImageNet-C), domain shift (ImageNet-R), resampled test sets (CIFAR-10.1 and ImageNet-v2), and worst-case robustness against $\ell_p$ perturbations.


**Summary Of The Review:**

The main strengths of the paper are its strong empirical results and that the proposed AdA is a generic approach that could be extended to other types of data. However, the strong empirical results come at the cost of highly increased computation at train time and it remains unclear if appropriately tuned SAM couldn't provide similar benefits. Moreover, the general purpose nature of the approach is not demonstrated because experiments are restricted to image classification. Finally, the theoretical considerations are based on an impractical assumption.
In summary, I consider the paper in its current form below the acceptance threshold (albeit marginally so). Showing the potential of AdA on other types of data _and_ comparing to SAM/AWP could bring it above the threshold.

### Update after rebuttal ###
The authors have convincingly addressed my concerns regarding a comparison to SAM/AWP, discussing the computational and memory overhead more transparently, and several further minor points. I am raising my score to 6: marginally accept.
However, I would still like to emphasize that the "Theoretical Considerations" in their current form are grounded on unrealistic assumptions and disconnected from the rest of the paper. The authors describe in the discussion how the theoretical considerations could be revised to be grounded on more realistic assumptions. It is difficult to judge the correctness of this line of thought based on a discussion in the review forum, without a proper revision of the paper. I highly encourage the authors to update the "Theoretical Considerations" of the main paper.
In summary, I lean towards acceptance because the paper is strong on the empirical side and provides a compelling approach - but this is in spite of (not because of) the theoretical considerations.

---

> ### Author Response · Authors · 2021-11-19
> **Rebuttal to Reviewer 31sh (part 1)**
>
> We thank the reviewer for their thorough and insightful review. We have split our reply into multiple comments.
>
> > Related work on sharpness-aware minimization [...] should be discussed. [...]  it would be important to clearly show an advantage of AdA over these works.
>
> We have run new experiments to compare our method with Vanilla Adversarial Training (AT) (Madry et al., 2018), TRADES (Zhang et al., 2019), Adversarial Weight Perturbations (AWP) (Wu et al., 2020) and Sharpness Aware Minimization (SAM) (Foret et al., 2021). Please find the results in the table below. We have also updated our paper to include these new results in Appendix F.
>
> | Setup | mCE | Noise err. | Blur err. | Weather err. | Digital err. | Cifar-10 acc. | Cifar-10.1 acc. | $\ell_2$ acc. $\epsilon=0.5$ | $\ell_2$ acc. $\epsilon=1.0$ | $\ell_\infty$ acc. $\epsilon=\frac{1}{255}$ | $\ell_\infty$ acc. $\epsilon=\frac{2}{255}$ |
> |:-----------------------|:----------|:----------------|:---------------|:------------------|:------------------|:------------------|:--------------------|:--------------------------------|:--------------------------------|:------------------------------------------------|:-------------------------------------------------|
> | AT ($\ell_\infty$) | 23.64 | 20.56 | 20.29 | 25.90 | 27.04 | 86.08 | 74.20 | 51.66 | 16.64 | 82.32 | 78.20 |
> | AT ($\ell_2$) | 17.42 | **13.93** | 15.20 | 19.34 | 20.33 | 91.02 | 80.40 | 67.06 | 30.73 | **85.98** | 79.77 |
> | TRADES ($\ell_\infty$) | 24.72 | 21.31 | 21.49 | 27.26 | 27.99 | 84.79 | 71.00 | 54.55 | 19.51 | 81.33 | 77.59 |
> | TRADES ($\ell_2$) | 18.08 | 14.07 | 15.64 | 20.46 | 21.15 | 89.96 | 78.60 | 68.16 | 37.01 | 85.07 | 79.12 |
> | AWP ($\ell_\infty$) | 25.01 | 21.73 | 21.58 | 27.69 | 28.23 | 84.36 | 70.50 | 57.01 | 24.07 | 81.38 | 77.98 |
> | AWP ($\ell_2$) | 18.79 | 14.92 | 15.87 | 21.54 | 21.86 | 89.20 | 77.40 | **71.79** | **44.83** | 85.18 | **80.21** |
> | SAM | 24.59 | 49.44 | 24.74 | 12.80 | 17.59 | **96.17** | **90.35** | 0.01 | 0.00 | 48.10 | 6.86 |
> | AdA (EDSR) | **15.47** | 26.83 | **14.94** | **10.80** | **12.14** | 93.37 | 86.55 | 9.28 | 0.09 | 72.41 | 41.13 |
>
>
> From our new experiments we observe that our method (AdA - EDSR) results in the most robustness to common image corruptions, as measured by mCE, compared to all adversarially trained baselines. Vanilla adversarial training (trained to defend against $\ell_2$ attacks) also produces classifiers which are highly resistant to common image corruptions (mCE of 17.42%). Unsurprisingly, the adversarially trained baselines (AT, TRADES and AWP) perform best on $\ell_2$ and $\ell_\infty$ norm bounded perturbations, as they are designed to defend against precisely these types of attacks. Our method obtains greater robustness to $\ell_p$ attacks compared to SAM, while SAM obtains the best generalization to CIFAR-10.1.

---

> ### Author Response · Authors · 2021-11-19
> **Rebuttal to Reviewer 31sh (part 2)**
>
> > Assumption 1 (Corruption coverage), which is the basis for the theoretical considerations, is not plausible. The authors show in Figure 3 that the first part of the assumption ("β has support as least as broad as α") is approximately fulfilled. However, the second part ("any corruption function sampled from β leaves the ground-truth label unchanged") seems highly unlikely: because of the first part of the assumption, the same property would have to hold for α. Prior work has established that images corresponding to different classes can have small ℓp distance. The support of corruptions such as Gaussian noise-severity 5 on an image thus very likely covers parts of the input space that belong to other classes (even though sampling such Gaussian noise is highly unlikely).
>
> Indeed, if all corruptions sampled from $\beta$ leave the ground-truth label unchanged, this must also hold for $\alpha$. Of course, it is true that this might be violated in practice (e.g., for unbounded perturbations, such as Gaussian noise). In the analysis we consider the idealized setting where both conditions in Assumption 1 hold, and this analysis provides insights on how the methods work. Additionally, we can easily get results for the case when Assumption 1 does not hold. If the perturbations can change the label, we can split the analysis into two: restricting $\alpha$ to the corruptions which do not change the label, and considering the remaining perturbations separately. Our existing analysis applies for the first part, while the additional error caused by the second part (corruptions which change the labels) can be bounded by the expected error conditioned on the corruption changing the label, and which can be bounded by $P(\mathrm{corruption\ changes\ the\ label})\cdot \max L$. Furthermore, in this case the error of the optimal robust classifier (the Bayes error) would be non-zero; in fact, if the support of $\alpha$ (and $\beta$) can be split in two disjoint sets $W_1$ and $W_2$ such that for any input, no corruption in $W_1$ will change the label, and all corruptions in $W_2$ will, and $\alpha(W_2)$ is small enough, then for the zero-one loss this term is exactly the Bayes error, and we obtain that the robust test error of the methods converge to the optimum. Finally, if the support of $\beta$ is smaller than that of $\alpha$, we can have a similar additional term in our error bound, which is the probability of selecting a corruption (according to $\alpha$) which is not in the support of $\beta$.

---

> > ### Comment · Reviewer_31sh · 2021-11-22
> > **Additional questions**
> >
> > I would like to thank the authors for the extensive response to my review. Most of my concerns have been addressed very well. I am replying to this part since I have some remaining questions regarding the theoretical considerations. Generally, I rated "Correctness: 2: Several of the paper’s claims are incorrect or not well-supported." because I found the claims of the theoretical considerations not well supported (not necessarily incorrect). But I am willing to revise this rating.
> >
> > The remaining questions I have are:
> >  * in the term P(corruption changes the label) * max L, over what domain is the maximization? Specifically, is max L bounded?
> >  * The union of W_1 and W_2  is not the entire support for \alpha, or? There will be corruptions that will change the label on some input but not on all.
> >  * Can the authors summarize the main take-away from the theoretical considerations? Specifically, does it connect to the empirical results in any direct way?

---

> > > ### Author Response · Authors · 2021-11-23
> > > **Reply to Reviewer 31sh**
> > >
> > > Thank you for your reply.
> > >
> > > > in the term P(corruption changes the label) * max L, over what domain is the maximization? Specifically, is max L bounded?
> > >
> > > The domain of the maximization is over corruptions sampled from $\alpha$. $L$ is the 0-1 loss so $\max L$ is indeed bounded.
> > >
> > > > The union of W_1 and W_2 is not the entire support for \alpha, or? There will be corruptions that will change the label on some input but not on all.
> > >
> > > We can consider $W_2$ to be the set of all corruptions which change the label on some input; if $\alpha(W_2)$ remains small enough then the above analysis applies -- and the union of $W_1$ and $W_2$ is the entire support for $\alpha$.
> > >
> > > If $\alpha(W_2)$ is not small enough above (e.g., if the margin is $0$, arbitrarily small perturbations can belong to $W_2$), a more refined analysis can be done, where we choose $W_2$ to be the set of all corruptions which change the labels on an input set of probability $1-\gamma$ (still the union of $W_1$ and $W_2$ form the entire support). This would result in an additional $\gamma$ term in the error bound, which can be balanced together with the $\alpha(W_2)$ term to get the tightest error bound.
> > >
> > > > Can the authors summarize the main take-away from the theoretical considerations? Specifically, does it connect to the empirical results in any direct way?
> > >
> > > The theoretical considerations validate that, when certain technical assumptions about the corruption sets hold, minimizing the robust error is sensible -- so (neglecting optimization issues) AdA converges to the optimum.
> > >
> > > > Most of my concerns have been addressed very well [...]
> > >
> > > > [...] I am willing to revise this rating.
> > >
> > > In light of our replies, we would appreciate it if the reviewer will be revising their rating.

---

> ### Author Response · Authors · 2021-11-19
> **Rebuttal to Reviewer 31sh (part 3)**
>
> > since the general method is not restricted to image classifiers or even image-based tasks, including results on other types of data would strengthen the paper.
>
> We agree; however, this would be a large undertaking, and targeting any new modality could easily generate enough results to constitute an entire new paper. As such, we have toned down our paper to state: "We believe the generic framework proposed in our paper, which (at a minimum) requires an auto-encoder and a scalar perturbation radius, is applicable to other domains beyond images. We leave further investigations to future work."
>
> > discussing the computational and memory overhead more transparently would be desirable.
>
> We agree. Our method (AdA) is indeed more computationally demanding than counterparts which utilize handcrafted heuristics. For example, AugMix specifies K image operations, which are applied stochastically at the input (i.e. over images). DeepAugment requires applying heuristic stochastic operations to the weights and activations of an image-to-image model - so this requires a forward pass through the image-to-image model (which can be computed offline). AdA must optimize over perturbations to the parameters of the image-to-image model; this requires several backpropagation steps and is similar to Lp-norm adversarial training (Madry et al., 2018) and related works.
>
> We summarize the computational requirements of our method (AdA), AugMix, DeepAugment, Perceptual Adversarial Training (PAT) (Laidlaw et al., 2021), Lp norm adversarial training (AT), TRADES (Zhang et al., 2019), Adversarial Weight Perturbations (AWP) (Wu et al., 2020) and Sharpness Aware Minimization (SAM) (Foret et al., 2021) in the table below:
>
> | Setup | Passes through classifier ($f_\theta$) | Passes through aux. net ($c_\phi$) | Adv. Projection |
> |:----------------|:-------------------------------------------|:---------------------------------------|:----------------------------------------------|
> | AdA (this work) | $\mathcal{O}(M)$ | $\mathcal{O}(M)$ | $\mathcal{O}(M \cdot \|\phi\| + \|x\|)$ |
> | PAT | $\mathcal{O}(M \cdot S)$ | $\mathcal{O}(M \cdot S)$ | $\mathcal{O}(N)$ |
> | AT | $\mathcal{O}(M)$ | - | $\mathcal{O}(M \cdot \|x\|)$ |
> | TRADES | $\mathcal{O}(M)$ | - | $\mathcal{O}(M \cdot \|x\|)$ |
> | AWP | $\mathcal{O}(M + W)$ | - | $\mathcal{O}(M\cdot\|x\| + W\cdot\|\theta\|)$ |
> | SAM | $\mathcal{O}(W)$ | - | - |
> | DeepAugment | $\mathcal{O}(1)$ | $\mathcal{O}(1)$ | - |
> | AugMix | $\mathcal{O}(1)$ | - | - |
>
> In this table $M$ represents the number of inner/adversarial optimizer steps (i.e. PGD steps in AdA or AT); $|x|$ is the dimensionality of the input; $|\theta|$ and $|\phi|$ are the number of parameters of the main classifier and of the auxiliary network respectively. The number of adversarial weight perturbations in AWP and SAM is denoted by $W$ (W is typically 1). For PAT, we also refer to the number of bisection iterations as $N$ and to the number of steps used for updating the Lagrange multiplier as $S$.
>
> We characterize the number of passes (forward and backward, jointly) through the main classifier network ($f_\theta$) and, for AdA and PAT, through an auxiliary neural network ($c_\phi$) separately. For methods using adversarial optimization we also describe the cost of the projection steps used for keeping perturbations within the feasible set. Taken together, these costs represent the computational complexity of performing one training step with each method.
>
> We have also added a new section in the paper (Appendix B.1) which includes an in-depth analysis of the computational and memory requirements of our method and a comparison with those of seven related works.
>
>
> > Moreover, the additional corruption network needs to fit in memory, increasing the required memory.
>
> Indeed, our method must store the weights of the image-to-image models in memory. However,  note that the models that we use are relatively small: UNet has 58627 parameters (0.23MB); VQ-VAE has 762053 parameters (3.05MB); EDSR has 1369859 parameters (5.48MB) and CAE has 2241859 parameters (8.97MB).

---

> ### Author Response · Authors · 2021-11-19
> **Rebuttal to Reviewer 31sh (part 4)**
>
> > AdA is by design a deterministic procedure that should generate the same augmented version of the input for a given model when applied repeatedly. A desirable property of data augmentation is diversity, that is: the same input is augmented differently in every application of the augmentation operator. Could the authors comment on why AdA is defined as a deterministic procedure? [...]
>
> AdA's goal is to train a classifier robust to image corruptions. Even though AdA can be viewed as a data augmentation method, it is more closely related to Lp adversarial training methods. For example, AdA also formulates the training objective as a robust optimization problem. Throughout the paper we show that solving this problem (approximately) yields classifiers with improved robustness to image corruptions.
>
> Note that, in theory, diversity is not required for training a robust model: the only requirement regarding samples is to find the worst-case one in each corruption set and enforce that the classifier labels it correctly.
> In practice, improving the training data diversity (by leveraging generated samples [1] or carefully tuned data augmentation [2]) has recently been found to improve Lp norm adversarial robustness. We leave investigating mechanisms for improving the diversity of samples within the AdA framework for future work.
>
> [1] Gowal et al., Improving Robustness using Generated Data, NeurIPS 2021
>
> [2] Rebuffi et al., Data Augmentation Can Improve Robustness, NeurIPS 2021
>
>
>
> > in the current form, all layers of the corruption network are perturbed. Would it make sense to perturb only a subset of the layers - for instance the first layers that presumable focus more on appearance than semantics?
>
> In general, we agree. However, as long as the underlying classifier capacity is large enough, covering more corruptions should not hurt the final accuracy. We have run some experiments that fix the last n-layers of the EDSR model early on, but we did not observe a noticeable improvement in performance.
>
> > shouldn't ηc in line 8 of Algorithm 1 depend on the layer index i - because it needs to depend on the {2, \phi} norm ? Also, shouldn't Algorithm 1 use the {2, \phi} norm?
>
> ηc is the scalar learning rate for the adversarial optimization procedure (i.e. the inner optimization). This is constant across outer and inner optimizer iterations and layers.
>
> Intermediary adversarial perturbations ($\delta$'s) are rescaled appropriately to obey the {2, \phi} norm in lines 9-11 in Algorithm 1.
>
>
> > Having error bars in Figure 3 that go beyond 1 does not make sense since SSIM cannot be larger than 1. [...]
>
> Thank you for the observation. In the original figure each error bar represented the standard deviation around the mean SSIM. We have now updated the figure so that each error bar represents the 95% confidence interval around the mean instead.
>
>
> > Correctness: 2: Several of the paper’s claims are incorrect or not well-supported.
>
> We are unsure why the rating of 2 was given with respect to correctness. The updated version of the manuscript and our replies above should address all reviewers' concerns. We would appreciate it if other incorrect claims can be specified as a response.

---

> ### Author Response · Authors · 2021-11-29
> **Response regarding theoretical considerations to Reviewer 31sh**
>
> We thank the reviewer for updating their recommendation after the rebuttal. We appreciate that the reviewer has remaining concerns regarding the theoretical considerations section.
>
> As suggested by the reviewer, we have formalized our previous reply to relax our initial corruption coverage assumption; we include this formalization below and have updated our paper to include this (but are prevented by the system from uploading a new revision now).
>
> We hope our updated theoretical considerations serve to alleviate, at least in part, the reviewer's concerns.
>
>
> ### Inexact corruption coverage.
>
> Assumption 1 can be unreasonable in practical settings: for example, if the perturbations induced by elements of $\\Phi$ can be unbounded (e.g., for simple Gaussian perturbations), the assumption that the labels do not change with any of the perturbations are usually violated (e.g., for sure for Gaussian perturbations). On the other hand, it may be reasonable to assume that there exists a subset of the perturbations, $\\Phi_1 \\subset \\Phi$ such that all perturbations in $\\Phi_1$ keep the label unchanged for any input $x \\in \\mathcal{X}$. Even this assumption becomes unrealistic if input points can be arbitrarily close to the decision boundary, in which case even for arbitrarily small perturbations we can find inputs for which the label changes. However, it is reasonable to assume that this does not happen if we only consider input points sufficiently far from the boundary (which is only meaningful if the probability of getting too close to the boundary is small, say at most some $\\gamma>0$). We formalize these considerations in the next assumption, where we assume that there exists a subset of input points $\\mathcal{X}_\\gamma \\subset \\mathcal{X}$ with  $\\mu(\\mathcal{X}_\\gamma) \\ge 1-\\gamma$ and a subset of perturbations $\\Phi_1$ such that Assumption 1 holds on $\\mathcal{X}_\\gamma$ and $\\Phi_1$:
>
> **Assumption 2. (Inexact corruption coverage)**
> Let $\\gamma \\in [0,1)$. Let $\\Phi_1 \\subset \\Phi$ and $\\mathcal{X}_\\gamma \\subset \\mathcal{X}$ with $\\mu(\\mathcal{X}_\\gamma) \\ge 1-\\gamma$ such that (i) no perturbation in $\\Phi_1$ changes the label of any $x \\in \\mathcal{X}_\\gamma$, that is, $f\_{\\theta*}(x)=f\_{\\theta*}(c\_\\phi(x))$ for any $\\phi \\in \\Phi_1$; (ii) there exists a distribution $\\beta$ supported on $\\Phi_1$ such that $\\alpha$ is absolutely continuous with respect to $\\beta$ on $\\Phi_1$.
>
>
> Assuming we have access to a sampling distribution $\\beta$ satisfying the above assumption and $\\mathcal{X}\_\\gamma$, we can consider the performance of the idealized AdA rule (Eq. 5) for $x\_1, \\ldots,x_n \\in \\mathcal{X}\_\\gamma$. Again by the result of Sugiyama et al. (2007), under Assumption 2, the resulting robust error restricted to $\\mathcal{X}\_\\gamma$ and $\\Phi\_1$ converges, as  $n\\to\\infty$, to the minimum restricted robust error
>
> $$
> E_{x \\sim \\mu|\_{\\mathcal{X}\_\\gamma}, \\phi \\sim \\alpha|\_{\\Phi\_1}}
> [L(f\_\\theta \\circ c\_\\phi(x),f\_{\\theta^*}(x))],
> $$
>
> where $\\mu|\_{\\mathcal{X}\_\\gamma}$ and $\\alpha|\_{\\Phi_1}$ denote the restrictions of $\\mu$ and $\\alpha$ to $\\mathcal{X}\_\\gamma$ and $\\Phi\_1$, respectively. Since $\\theta^* \\in \\Theta$ and by the assumption, no corruption in $\\Phi\_1$ changes the label, this minimum is in fact 0.
>
> Then, assuming the loss function $L$ takes values in $[0,L\_{\\max}]$ (where $L\_{\\max}=1$ for the zero-one loss), the error of the classifier when the input is not in $\\mathcal{X}\_\\gamma$ or the perturbation is not in $\\Phi\_1$, can be bounded by $L\_{\\max}$. Hence, the robust error of the learned classifier $f_{\\widehat{\\theta}\_{AdA}}$ (obtained for $n=\\infty$, i.e., in the limit of infinite data) can be bounded as
>
> $$
> E\_{x \\sim \\mu, \\phi \\sim \\alpha [L(f\_{\\widehat{\\theta}\_{AdA}} \\circ c\_\\phi(x),f\_{\\theta^*}(x))]}
>  \\le E\_{x \\sim \\mu|\_{\\mathcal{X}\_\\gamma}, \\phi \\sim \\alpha|\_{\\Phi\_1}}
> L(f\_{\\widehat{\\theta}\_{AdA}} \\circ c\_\\phi(x),f\_{\\theta^*}(x)) + (\\gamma + 1-\\alpha(\\Phi\_1)) L\_{\\max}
>  = E\_{x \\sim \\mu|\_{\\mathcal{X}\_\\gamma}, \\phi \\sim \\alpha|\_{\\Phi_1}}
> L(f\_{\\theta^*} \\circ c\_\\phi(x),f\_{\\theta^*}(x)) + (\\gamma + 1-\\alpha(\\Phi\_1)) L\_{\\max}
> = (\\gamma + 1- \\alpha(\\Phi\_1)) L\_{\\max}
> $$
>
> Note that in the assumption there is an interplay between $\mathcal{X}\_\gamma$ and $\Phi\_1$ (the conditions on $\Phi\_1$ only apply to inputs from $\mathcal{X}\_\gamma$, and as $\mathcal{X}\_\gamma$ decreases, $\Phi\_1$ can increase). As one should always choose $\mathcal{X}\_\gamma$ so that $\Phi\_1$ be the largest given $\gamma$,  to get the best bound, one can optimize $\gamma$ to minimize $\gamma + 1-\alpha(\Phi\_1)$. If the probability of the inputs close to the decision boundary ($\gamma$) and the probability of perturbations changing the label for some input ($1-\alpha(\Phi\_1)$) are small enough, the resulting bound also becomes small.

---

### Official Review · Reviewer_AUQw · 2021-11-02

**Correctness:** 4
**Technical Novelty And Significance:** 3
**Empirical Novelty And Significance:** 3
**Recommendation:** 6
**Confidence:** 3

**Main Review:**

strengths:
===========
- empirical evaluations are thorough and detailed.

- formalisation and theoretical analysis are provided, connecting the proposed method with previous work

- combining defense against perturbation with adversarial training, providing a data augmentation method that can provide resilience against both at the same time, without sacrificing generalisation.

weaknesses:
===========
- Main results (Table1,2) suggest that none of the single Ada methods can consistently outperform DeepAugment or AugMix, in any of the corruptions, clean accuracy, or adversarial attacks. In fact, none of the 4 proposed variants (EDSR,CAE,UNET, VQVAE) can by their own outperform DeepAugment or AugMix, on any of the corruptions, generalisation, or adversarial robustness (compare any of the Ada methods, with nominal row in AugMix and DeepAugment). This is a very important drawback, which is  hard to justify, that only by combinations with other techniques, the proposed method becomes effective.

- Although one of the aspects of this work is building adversarial robustness into models, no thorough comparisons against SOTA adversarial training methods (such as [1,2,3]) are provided. Table4 in appendix F only shows perturbation and clean performance, and not adversarial robust accuracy. This is the second drawback of the current paper, that does not provide the full comparison to the two edge of the related work. Ideally, we would like a method that can improve both, or at least perform on-par, or with a reasonable trade-off.

- The proposed approach seems to be significantly more computationally demanding than its counterparts AugMix and DeepAugment. I would like to see a comparison in terms of computational requirements.


Additional questions
====================

- In Appx E, what is the value of max threshold t? and how is it calculated?

- In the sample guarding mechanism, how many of the augmented samples are guarded? are there any statistics? and also on the linear interpolation, some statistics on what values of lambda have been used would be insightful, to demonstrate that actually the augmented samples have been largely influential, and not ignored by the guard or interpolation.

- Do all methods (Ada, DeepAugment, AugMix) use the same training strategy (lr schedule, resizing on ImageNet, etc)?

- In Appx C, it is mentioned that for ImageNet and CIFAR10 standard data augmentation has been used. Is this the basic setting? E.g, all methods used and evaluated in this paper, use the "standard" augmentations, in addition to the mentioned augmentations (e.g, DeemAugment, AugMix, Ada)?

- In Alg1, it seems there is no randomness choosing between training on real samples and augmented samples. Were models always trained on augmented samples (e.g, no original training was used)?

- In Appx A: One of the benefits of Ada was mentioned to be its generality to other tasks and data modalities such as Audio and text. Though there is no provided result that can support this claim. I recommend to rephrase or strengthen the claim by additional results.

- In section 4, discussions, where did 10^-5 ||\delta|| come from?


- are there any evaluation metrics on the generative models used? How good do they perform in the task they have been trained on?


- Adversarial Mixing [4] is an augmentation method that similar to Ada uses a generative model, as well as an adversarial loss, to create augmented samples. In this regard, it is the closes method to Ada. I expect to see how it compares to Ada, at least on some of the experiments.

refs:
=====
[1] Zhang, Hongyang, et al. "Theoretically principled trade-off between robustness and accuracy." International Conference on Machine Learning. PMLR, 2019.

[2] Gowal, Sven, et al. "Uncovering the limits of adversarial training against norm-bounded adversarial examples." arXiv preprint arXiv:2010.03593 (2020).

[3] Kireev, Klim, Maksym Andriushchenko, and Nicolas Flammarion. "On the effectiveness of adversarial training against common corruptions." arXiv preprint arXiv:2103.02325 (2021).

[4] Gowal, Sven, et al. "Achieving robustness in the wild via adversarial mixing with disentangled representations." Proceedings of the IEEE/CVF Conference on Computer Vision and Pattern Recognition. 2020.


**Summary Of The Paper:**

This paper provides a data augmentation method that augments samples by perturbing parameters of a generative model. The perturbations are found by an adversarial loss, and are constrained based on a perceptual similarity distance to guard from the outliers.
In addition to thorough empirical evaluations, this paper provides formalization of their method and a closely related one (deep augment), adding theoretical insights , and provide interesting convergence properties and PAC-Bayesian analysis.


**Summary Of The Review:**

The proposed method builds on recent related work, and its design is justified. Provides insightful empirical and theopretical results, and unifying some related work.
However:
- Empirical results does not support the claims (considering single methods applied)
- Is not compared to adversarial training methods on adversarial robustness AND perturbation robustness.
- Some relevant approaches have been missed in comparisons, as discussed above.


## after rebuttal:
During the rebuttal, the authors responded to all my concerns, and strengthened their empirical evaluations with additional results. Also the claims of the paper are now aligned with the reported results, and the limitations of the proposed approach is clarified and sufficiently discussed.
I find this paper a good reference for research on adversarial and perturbation robustness method that benchmarks a wide range of methods from both areas, with a method that although has its limitation, a strong baseline for the future work in this direction. I therefore increase my score.

---

> ### Author Response · Authors · 2021-11-19
> **Rebuttal to Reviewer AUQw (part 1)**
>
> We thank the reviewer for their thorough and insightful review. We have split our reply into multiple comments.
>
> > Main results (Table1,2) suggest that none of the single Ada methods can consistently outperform DeepAugment or AugMix, in any of the corruptions, clean accuracy, or adversarial attacks. In fact, none of the 4 proposed variants (EDSR,CAE,UNET, VQVAE) can by their own outperform DeepAugment or AugMix, on any of the corruptions, generalisation, or adversarial robustness (compare any of the Ada methods, with nominal row in AugMix and DeepAugment).
>
> There might be a misunderstanding with regards to the result tables. Our individual AdA models do perform better than AugMix and DeepAugment in the majority of cases on the metrics mentioned in your statement.
>
> On CIFAR-10 (Table 1), our single model, AdA (EDSR; row 4), performs better than AugMix (row 7) or DeepAugment (row 13) on 11 out of 20 cases (on the metrics mentioned in your statement).
>
> On ImageNet (Table 2), our single model, AdA (EDSR; row 4) performs better than AugMix (row 7) or DeepAugment (row 13) on 13 out of 22 cases (on the metrics mentioned in your statement). Furthermore, AdA (Full; row 6), which uses more image-to-image models performs better on 16 out of 22 cases.
>
> Also note that our single model (row 4 in both Tables 1 and 2) performs better on all L-2 and L-inf attacks than both AugMix and DeepAugment.
>
> We have added indices to each row in Tables 1 and 2 to facilitate comparisons.
>
>
> Indeed, as mentioned in the review, our method is complementary with additional data augmentation methods, and increases in performance do result from combining multiple methods. But, we believe the greatest benefit of our method is its generality and ease-of-use, as it does not rely on handcrafted image processing operations (AugMix) or heuristic weights and activations transformations (DeepAugment). Just to highlight the difficulty of building such heuristics, here is one of the 25 random weight transformations done by DeepAugment:
>
> ```
> i = np.random.choice(np.arange(1,13,4))
> z = torch.zeros_like(weights['body.'+str(i)+'.body.0.weight'])
> for j in range(z.size(0)):
>     shift_x, shift_y = np.random.randint(3, size=(2,))
>     z[:,j,shift_x,shift_y] = np.random.choice([1.,-1.])
> weights['body.'+str(i)+'.body.0.weight'] = conv2d(weights['body.'+str(i)+'.body.0.weight'], z, padding=1)
> ```
> As one can see, these are not only far from trivial but also do not generalize to other image-to-image models to the contrary of our method.

---

> > ### Comment · Reviewer_AUQw · 2021-11-22
> > **Response to authors**
> >
> > I thank the authors for their response.
> > Regarding the misunderstanding, I clarify what I meant in the quoted paragraph, and provide more details.
> >
> > In comparisons of AdA-EDSR when applied alone, with image corruption benchmarks (AugMix, DeepAugment):
> >
> > in Table 1 (Cifar10):
> >
> > for corruptions (mCE): both AugMix and DeepAug are better than EDSR.
> >
> > For accuracy:
> > AugMix is better than EDSR, in both CIFAR10 and CIFAR10.1.
> >
> > For adversarial robustness:
> > EDSR is clearly better.
> >
> >
> > In Table 2:
> > for corruptions (mCE):
> > both AugMix and DeepAug are better than EDSR.
> >
> >
> > For accuracy:
> > AugMix is better than EDSR in 2/3 of the cases, and DeepAugment in 1/3.
> >
> > For adversarial robustness:
> > EDSR is clearly better.
> >
> >
> > In summary, in terms of clean and corrupted accuracy,  AdA is consistently outperformed by AugMix on 2 datasets.
> > In terms of adversarial robustness in comparisons with image corruption benchmarks, AdA outperforms both baselines (AugMix, DeepAugment).
> >
> > I hope it clarifies my previous comments.

---

> > > ### Author Response · Authors · 2021-11-23
> > > **Reply to AUQw**
> > >
> > > We thank the reviewer for the clarifications.
> > >
> > > > in Table 1 (Cifar10):
> > >
> > > Regarding Table 1, we believe the reviewer's comments appear to be inadvertently missing some comparisons where AdA performs better than the two baselines.
> > >
> > > We would like to add:
> > > - For corruptions (mCE): both AugMix and DeepAug are better than AdA (EDSR). AdA (EDSR) is better on the Digital corruption error group than DeepAugment.
> > > - For accuracy: AugMix is better than AdA (EDSR), in both CIFAR10 and CIFAR10.1. But, AdA (EDSR) is better than DeepAugment, in both CIFAR10 and CIFAR10.1.
> > > - For adversarial robustness: AdA (EDSR) is best.
> > >
> > > > In Table 2:
> > >
> > > For Table 2 (ImageNet), the best-performing and relevant AdA variant is "AdA (All)" (as described in section 5 in the paragraph on common corruptions).
> > >
> > > In our previous response to the reviewer's comment ("Rebuttal to Reviewer AUQw (part 1)") we accidentally mentioned "AdA (EDSR)" but nevertheless proceeded to describe the performance of the actually relevant method "AdA (All)". We apologize for this confusion. "AdA (All)" is also the method we refer to in the main manuscript when describing the best AdA variant on ImageNet.
> > >
> > > We would like to specify:
> > > - For corruptions (mCE): DeepAugment is better than AdA (All). But, AdA (All) is better than AugMix. AdA (All) is best on both the Weather corruption group and the Digital corruption group.
> > > - For accuracy: On ImageNet-R AdA (All) is best. On both ImageNet and ImageNet-v2 AugMix is better than AdA (All); but, AdA (All) is better than DeepAugment.
> > > - For adversarial robustness: AdA (All) is best.
> > >
> > > > In summary, in terms of clean and corrupted accuracy, AdA is consistently outperformed by AugMix on 2 datasets.
> > >
> > > From the fine-grained analysis above we can see that this is not the case.
> > >
> > > We wanted to also note that using our method in combination with AugMix and DeepAugment results in a new state-of-the-art mCE on CIFAR-10. Given the difficulty of making progress in this direction, we believe this result is significant in itself -- even when requiring a combination of methods.
> > >
> > > [Edited to correct the typo: "AdA (Full)" to "AdA (All)"]

---

> ### Author Response · Authors · 2021-11-19
> **Rebuttal to Reviewer AUQw (part 2)**
>
> > Although one of the aspects of this work is building adversarial robustness into models, no thorough comparisons against SOTA adversarial training methods (such as [1,2,3]) are provided.
>
> We have run new experiments to compare our method with Vanilla Adversarial Training (AT) (Madry et al., 2018), TRADES (Zhang et al., 2019), Adversarial Weight Perturbations (AWP) (Wu et al., 2020) and Sharpness Aware Minimization (SAM) (Foret et al., 2021). Please find the results in the table below. We have also updated our paper to include these new results in Appendix F.
>
> | Setup | mCE | Noise err. | Blur err. | Weather err. | Digital err. | Cifar-10 acc. | Cifar-10.1 acc. | $\ell_2$ acc. $\epsilon=0.5$ | $\ell_2$ acc. $\epsilon=1.0$ | $\ell_\infty$ acc. $\epsilon=\frac{1}{255}$ | $\ell_\infty$ acc. $\epsilon=\frac{2}{255}$ |
> |:-----------------------|:----------|:----------------|:---------------|:------------------|:------------------|:------------------|:--------------------|:--------------------------------|:--------------------------------|:------------------------------------------------|:-------------------------------------------------|
> | AT ($\ell_\infty$) | 23.64 | 20.56 | 20.29 | 25.90 | 27.04 | 86.08 | 74.20 | 51.66 | 16.64 | 82.32 | 78.20 |
> | AT ($\ell_2$) | 17.42 | **13.93** | 15.20 | 19.34 | 20.33 | 91.02 | 80.40 | 67.06 | 30.73 | **85.98** | 79.77 |
> | TRADES ($\ell_\infty$) | 24.72 | 21.31 | 21.49 | 27.26 | 27.99 | 84.79 | 71.00 | 54.55 | 19.51 | 81.33 | 77.59 |
> | TRADES ($\ell_2$) | 18.08 | 14.07 | 15.64 | 20.46 | 21.15 | 89.96 | 78.60 | 68.16 | 37.01 | 85.07 | 79.12 |
> | AWP ($\ell_\infty$) | 25.01 | 21.73 | 21.58 | 27.69 | 28.23 | 84.36 | 70.50 | 57.01 | 24.07 | 81.38 | 77.98 |
> | AWP ($\ell_2$) | 18.79 | 14.92 | 15.87 | 21.54 | 21.86 | 89.20 | 77.40 | **71.79** | **44.83** | 85.18 | **80.21** |
> | SAM | 24.59 | 49.44 | 24.74 | 12.80 | 17.59 | **96.17** | **90.35** | 0.01 | 0.00 | 48.10 | 6.86 |
> | AdA (EDSR) | **15.47** | 26.83 | **14.94** | **10.80** | **12.14** | 93.37 | 86.55 | 9.28 | 0.09 | 72.41 | 41.13 |
>
>
> From our new experiments we observe that our method (AdA - EDSR) results in the most robustness to common image corruptions, as measured by mCE, compared to all adversarially trained baselines. Vanilla adversarial training (trained to defend against $\ell_2$ attacks) also produces classifiers which are highly resistant to common image corruptions (mCE of 17.42%). Unsurprisingly, the adversarially trained baselines (AT, TRADES and AWP) perform best on $\ell_2$ and $\ell_\infty$ norm bounded perturbations, as they are designed to defend against precisely these types of attacks. Our method obtains greater robustness to $\ell_p$ attacks compared to SAM, while SAM obtains the best generalization to CIFAR-10.1.
>
> Note that in our original paper (in Appendix F) we also compared our method's performance with the performance of the strongest known adversarially trained model against Lp norm attacks on common image corruptions (Gowal et al., 2020). This model was trained against $\ell_2$ norm bounded perturbations with $\epsilon=0.5$ with extra-data and obtains 12.32\% mCE; our best model obtains 7.83\% mCE.
>
>
>
>
> >Table4 in appendix F only shows perturbation and clean performance, and not adversarial robust accuracy.
>
> Regarding comparisons to Perceptual Adversarial Training (PAT) indeed Table 7 (originally 4) in Appendix F does not show adversarial robust accuracy. We received the performances on common image corruptions from the PAT authors but we do not have access to the original models to evaluate their robustness to Lp attacks at the same perturbation radii that we used throughout our paper.
>
> However, as we stated in the original paper in the second paragraph of Appendix F, PAT results in classifiers which have considerably higher adversarial robustness than classifiers trained using our method. Specifically, the best PAT model obtains 28.7\% robust accuracy against $\ell_\infty$ attacks ($\epsilon=8/255$) and 33.3\% on $\ell_2$ attacks ($\epsilon=1$) while our best AdA variant obtains less robust accuracy in each case: 0.99\% against $\ell_2$ $\epsilon=1$ attacks and 13.88\% against $\ell_\infty$ $\epsilon=4/255$ attacks. Note that for $\ell_\infty$ attacks our method obtains less robust accuracy at a smaller perturbation radius than PAT does at a higher perturbation radius.
>
> This difference in performance against Lp-norm attacks is not surprising, as PAT addresses robustness to pixel-level attacks (i.e., it manipulates image pixels directly); whereas our method applies adversarial perturbations to the corruption function parameters (and not to the image pixels directly).

---

> > ### Comment · Reviewer_AUQw · 2021-11-22
> > **Response to authors**
> >
> > Thank you for your detailed response.
> >
> > In results presented now in Table5, it can be seen that AdA does not perform better than Adversarial training methods on both clean and adversarial accuracy.
> > Considering the results in Table 1,2; and in summary, AdA is better in adversarial robustness than methods designed for corruption robustness, and is better in corruption robustness than methods designed for adversarial robustness, and Ada as a method that is targeting both corruption robustness and adversarial robustness, it is not breaking the Pareto-front.
> >
> > Achieving different kinds of robustness without sacrificing much of generalisation is a hard task and therefore, is of importance that this front is explored.
> > However, the failures discussed above are not communicated clearly, which may be misleading. It needs to be clarified that, AdA fails at outperforming the current methods designed specifically for a kind of robustness. I recommend Table 5, &, and results above, be added to Table1 for a broader view. At its current form, I find it a bit misleading that comparisons to adversarial training methods are only provided at the end of the appendix. A section for discussing the limits of this work as provided above is strongly encouraged!

---

> > > ### Author Response · Authors · 2021-11-23
> > > **Reply to AUQw**
> > >
> > > > I recommend Table 5, &, and results above, be added to Table1 for a broader view.
> > >
> > > > [...] comparisons to adversarial training methods are only provided at the end of the appendix
> > >
> > > Thank you for the suggestion. We have incorporated Table 5 into Table 1 from the main manuscript. We have incorporated Table 6 (which provides additional information on individual corruption errors) into Table 8 in appendix (for space reasons). We have incorporated the results above on adversarial training methods into the empirical results section of the main manuscript.
> > >
> > > > A section for discussing the limits of this work as provided above is strongly encouraged!
> > >
> > > Thank you for the suggestion. We have added a new section (Section 6) on limitations of our method (including on its performance when applied by itself) just before the conclusion section.

---

> ### Author Response · Authors · 2021-11-19
> **Rebuttal to Reviewer AUQw (part 3)**
>
> > The proposed approach seems to be significantly more computationally demanding than its counterparts AugMix and DeepAugment. I would like to see a comparison in terms of computational requirements.
>
> Our method (AdA) is indeed more computationally demanding than counterparts which utilize handcrafted heuristics. For example, AugMix specifies K image operations, which are applied stochastically at the input (i.e. over images). DeepAugment requires applying heuristic stochastic operations to the weights and activations of an image-to-image model - so this requires a forward pass through the image-to-image model (which can be computed offline). AdA must optimize over perturbations to the parameters of the image-to-image model; this requires several backpropagation steps and is similar to Lp-norm adversarial training (Madry et al., 2018) and related works.
>
> We summarize the computational requirements of our method (AdA), AugMix, DeepAugment, Perceptual Adversarial Training (PAT) (Laidlaw et al., 2021), Lp norm adversarial training (AT), TRADES (Zhang et al., 2019), Adversarial Weight Perturbations (AWP) (Wu et al., 2020) and Sharpness Aware Minimization (SAM) (Foret et al., 2021) in the table below:
>
>
> | Setup | Passes through classifier ($f_\theta$) | Passes through aux. net ($c_\phi$) | Adv. Projection |
> |:----------------|:-------------------------------------------|:---------------------------------------|:----------------------------------------------|
> | AdA (this work) | $\mathcal{O}(M)$ | $\mathcal{O}(M)$ | $\mathcal{O}(M \cdot \|\phi\| + \|x\|)$ |
> | PAT | $\mathcal{O}(M \cdot S)$ | $\mathcal{O}(M \cdot S)$ | $\mathcal{O}(N)$ |
> | AT | $\mathcal{O}(M)$ | - | $\mathcal{O}(M \cdot \|x\|)$ |
> | TRADES | $\mathcal{O}(M)$ | - | $\mathcal{O}(M \cdot \|x\|)$ |
> | AWP | $\mathcal{O}(M + W)$ | - | $\mathcal{O}(M\cdot\|x\| + W\cdot\|\theta\|)$ |
> | SAM | $\mathcal{O}(W)$ | - | - |
> | DeepAugment | $\mathcal{O}(1)$ | $\mathcal{O}(1)$ | - |
> | AugMix | $\mathcal{O}(1)$ | - | - |
>
> In this table $M$ represents the number of inner/adversarial optimizer steps (i.e. PGD steps in AdA or AT); $|x|$ is the dimensionality of the input; $|\theta|$ and $|\phi|$ are the number of parameters of the main classifier and of the auxiliary network respectively. The number of adversarial weight perturbations in AWP and SAM is denoted by $W$ (W is typically 1). For PAT, we also refer to the number of bisection iterations as $N$ and to the number of steps used for updating the Lagrange multiplier as $S$.
>
> In this table $M$ represents the number of inner/adversarial optimizer steps (i.e. PGD steps in AdA or AT); $|x|$ is the dimensionality of the input; $|\theta|$ and $|\phi|$ are the number of parameters of the main classifier and of the auxiliary network respectively. The number of adversarial weight perturbations in AWP and SAM is denoted by $W$ (W is typically 1). For PAT, we also refer to the number of bisection iterations as $N$ and to the number of steps used for updating the Lagrange multiplier as $S$.
> We characterize the number of passes (forward and backward, jointly) through the main classifier network ($f_\theta$) and, for AdA and PAT, through an auxiliary neural network ($c_\phi$) separately. For methods using adversarial optimization we also describe the cost of the projection steps used for keeping perturbations within the feasible set. Taken together, these costs represent the computational complexity of performing one training step with each method.
>
> We have also added a new section in the paper (Appendix B.1) which includes an in-depth analysis of the computational and memory requirements of our method and a comparison with those of seven related works.

---

> ### Author Response · Authors · 2021-11-19
> **Rebuttal to Reviewer AUQw (part 4)**
>
> > In Appx E, what is the value of max threshold t? and how is it calculated?
>
>  The max threshold (t) is set manually. In initial experiments (using AdA - EDSR with AugMix) we sweeped over [0.3, 0.4, 0.7] for the threshold and selected the best performing setting on a validation set for each dataset: we set the threshold to 0.3 for CIFAR and 0.7 for ImageNet. We have updated the paper (Appendix E) to specify this.
>
> > In the sample guarding mechanism, how many of the augmented samples are guarded? are there any statistics? [...]
>
> Figure 2 can be used to estimate the proportion of augmented samples which are guarded. For an SSIM of 0.3, by inspection we can note that a very small proportion (i.e. less than 1%) of adversarial samples are guarded for either CAE (ν = 0.015) or EDSR (ν = 0.015). With an SSIM threshold of 0.7, approximately 43% of samples are guarded for CAE (ν = 0.015) and 0.62% for EDSR (ν = 0.015). As a result, we can observe that for our best variant on CIFAR-10 (AdA with EDSR), we only guard 0.62% of samples and as such we can state that non-guarded augmented samples are largely influential in this setting.
>
> > Do all methods (Ada, DeepAugment, AugMix) use the same training strategy (lr schedule, resizing on ImageNet, etc)?
>
> Yes, all methods (AdA, DeepAugment and AugMix) use exactly the same training strategy and the same codebase (as we re-implemented all baselines ourselves). We have clarified this in Appendix C.
>
> > In Appx C, it is mentioned that for ImageNet and CIFAR10 standard data augmentation has been used. Is this the basic setting? E.g, all methods used and evaluated in this paper, use the "standard" augmentations, in addition to the mentioned augmentations (e.g, DeemAugment, AugMix, Ada)?
>
> Yes, all methods use standard data augmentation as a pre-processing step. This is described in Appendix C in the "Combining data augmentation methods" paragraph. That is, for CIFAR-10 we use 4 pixel padding and random cropping, and for ImageNet we use a random left-right flip and a resized crop.
>
> > In Alg1, it seems there is no randomness choosing between training on real samples and augmented samples. Were models always trained on augmented samples (e.g, no original training was used)?
>
>  Yes, this is correct; AdA models are always trained on augmented samples only.
>
> > In Appx A: One of the benefits of Ada was mentioned to be its generality to other tasks and data modalities such as Audio and text. Though there is no provided result that can support this claim. I recommend to rephrase or strengthen the claim by additional results.
>
> Thank you for the observation. We have toned down our statement. Specifically, we now state "We believe the generic framework proposed in our paper, which (at a minimum) requires an auto-encoder and a scalar perturbation radius, is applicable to other domains beyond images. We leave further investigations to future work."
>
> > In section 4, discussions, where did 10^-5 ||\delta|| come from?
>
> This is L2 regularization with a fixed weight of 10^-5 which we use to penalize too aggressive weight perturbations. We have updated the paper (Section 4: Discussions) to clarify this.
>
> > are there any evaluation metrics on the generative models used? How good do they perform in the task they have been trained on?
>
> We use pre-trained weights for both EDSR and CAE and train a VQ-VAE and a U-Net ourselves (architecture and training details are provided in the original paper in App. C: "Corruption networks"). Metrics regarding the performance of the EDSR and CAE models can be found in their respective papers ([5], [6]).
>
> The U-Net model we train for CIFAR obtains 0.01508 L1 train loss, 0.000357 L2 train loss and 46.196 test PSNR; the one we train for ImageNet obtains 0.00706 L1 train loss, 0.000354 L2 train loss and 39.37 test PSNR.
>
> The VQ-VAE model we train for CIFAR obtains 0.08358 VQ-VAE loss (Eq. 3 in [7]) and 25.42 PSNR test loss; the one we train for ImageNet obtains 0.0834 VQ-VAE train loss, and 26.46 test PSNR.
>
> [5] Lim et al., Enhanced Deep Residual Networks for Single Image Super-Resolution, CVPR, 2017
>
> [6] Theis et al., Lossy Image Compression with Compressive Autoencoders, ICLR 2017
>
> [7] van den Oord et al., Neural Discrete Representation Learning, NeurIPS 2017

---

> ### Author Response · Authors · 2021-11-19
> **Rebuttal to Reviewer AUQw (part 5)**
>
> > Adversarial Mixing [4] is an augmentation method that similar to Ada uses a generative model, as well as an adversarial loss, to create augmented samples. In this regard, it is the closes method to Ada. I expect to see how it compares to Ada, at least on some of the experiments.
>
> While Adversarial Mixing [4] is similar algorithmically, it relies on knowing which part of the latents of the underlying generative model are disentangled from the classification label. This partitioning of the latents is unknown for the case of common corruptions and it is unclear that StyleGAN (which is the model used in [4]) can model such corruptions. As such, we decided to focus our comparisons with PAT rather than AdvMix where possible. Also note that AdA does not need a generative model (just an image-to-image model). The closest to [4] that looked into common corruptions without the requirement of knowing this disentanglement is [8]. In [8], the authors feed a subset of the common corruptions to train the underlying generative model. Despite this additional knowledge, they only achieve between 9.5 and 9.7 mCE. We have added a reference to [8] for completeness.
>
> [8] E. Wong and J. Z. Kolter,  Learning perturbation sets for robust machine learning, 2021.
>
>
>
> > Empirical results does not support the claims (considering single methods applied)
>
> We believe our updated paper and answers above support all claims. If not, we would appreciate additional comments that specify exactly which claims are not supported.
>
>
> > Is not compared to adversarial training methods on adversarial robustness AND perturbation robustness.
> > Some relevant approaches have been missed in comparisons, as discussed above.
>
> Thank you for pointing this out. As detailed in our answers, we have updated our paper (Appendix F) to compare against several additional adversarial training methods with results on both adversarial robustness and common image corruptions.

---

> > ### Comment · Reviewer_AUQw · 2021-11-22
> > **Response to authors**
> >
> > I thank the authors for their response.
> >
> > This statement, and others with a similar message, are incorrect (or misleading, to say the least):
> > "Our classifiers improve upon the state-of-the-art on common image corruption benchmarks conducted in expectation on CIFAR-10-C and improve worst-case performance against p-norm bounded perturbations on both CIFAR-10 and IMAGENET."
> >
> > When applied alone, AdA does not outperform AugMix in terms of perturbation robustness (as measured by mCE).
> >
> > In summary, there needs to be a discussion on the failures of AdA (e.g, when applied alone):
> > 1. compared against image corruption robustness methods (e.g, AugMix) Ada does not have a better performance (as measured by mCE) in terms of corruption robustness.
> >
> > 2. compared against adversarial training methods (e.g, AWP) Ada does not have a better performance (as measured by l_p accuracy) in terms of adversarial robustness.

---

> > > ### Author Response · Authors · 2021-11-23
> > > **Reply to AUQw**
> > >
> > > > This statement, and others with a similar message, are incorrect (or misleading, to say the least [...]
> > >
> > > We understand how this statement can appear misleading despite our best intentions. We have updated the statement to read: "Classifiers trained using our method in conjunction with prior methods (AugMix & DeepAugment) improve upon the state-of-the-art on common image corruption benchmarks conducted in expectation on CIFAR-10-C and also improve worst-case performance against Lp-norm bounded perturbations on both CIFAR-10 and ImageNet."
> > >
> > > To increase clarity, we have also updated the second sentence in Section 5 in the "Overview" paragraph to read: "On ImageNet (downsampled to 128×128) we demonstrate that our method can leverage 4 image-to-image models simultaneously to obtain the largest increases in mCE. Specifically, by combining AdA (All) with DeepAugment and AugMix we can obtain 62.90% mCE [...]".
> > >
> > > Similarly, we have updated our claim in the last sentence of the "Comparison to previous work on CIFAR-10" paragraph of Appendix F to read: "To the best of our knowledge, our best performing model (AdA (EDSR) coupled with AugMix and DeepAugment) is more robust to common image corruptions on CIFAR-10 than all previous methods, obtaining a new state-of-the-art mCE of 7.83%."
> > >
> > > Let us know if you believe other statements in our paper require further clarification.
> > >
> > >
> > > > there needs to be a discussion on the failures of AdA (e.g, when applied alone)
> > >
> > > Thank you for the suggestion. We have added a new section (Section 6) on limitations of our method (including on its performance when applied by itself) just before the conclusion section.
> > >
> > > > compared against adversarial training methods (e.g, AWP) Ada does not have a better performance (as measured by l_p accuracy) in terms of adversarial robustness
> > >
> > > This is expected, as mentioned above. The adversarially trained baselines (AT, TRADES and AWP) perform best on Lp-norm bounded perturbations, as they are designed to defend against precisely these types of attacks.
> > >
> > > We have included this in the new limitations section.
> > >
> > >
> > > > compared against image corruption robustness methods (e.g, AugMix) Ada does not have a better performance (as measured by mCE) in terms of corruption robustness.
> > >
> > > When AdA is used *alone* it does not result in increased corruption robustness -- except for one case: AdA alone performs better than AugMix on ImageNet; cf. rows 6 vs. 7 in Table 2.
> > >
> > > We have included this in the new limitations section.
> > >
> > > In the majority of cases, we can also see that AdA is complementary to AugMix, DeepAugment and both in terms of robustness to image corruptions. I.e. training using a specific AdA variant and AugMix and/or DeepAugment results in better mCE than training the same classifier _without_ using AdA (in all but one case, cf. rows 20 vs. 23 in Table 1).

---

> ### Comment · Reviewer_AUQw · 2021-11-29
> **After rebuttal**
>
>
> During the rebuttal, the authors responded to all my concerns, and strengthened their empirical evaluations with additional results. Also the claims of the paper are now aligned with the reported results, and the limitations of the proposed approach is clarified and sufficiently discussed. I find this paper a good reference for research on adversarial and perturbation robustness method that benchmarks a wide range of methods from both areas, with a method that although has its limitation, a strong baseline for the future work in this direction. I therefore increase my score.

---

### Official Review · Reviewer_fuXu · 2021-11-03

**Correctness:** 4
**Technical Novelty And Significance:** 3
**Empirical Novelty And Significance:** Not applicable
**Recommendation:** 6
**Confidence:** 5

**Main Review:**

*Strength*

In contrast to attacking the classifier, this approach makes a novel use of image-to-image translations network to generate corrupted samples for the classifier. This general proposed approach is amenable to learning a wide-set of image transformations, thus creating a potential for a big impact on the current state of the art in the field.

*Weakness*

I think that the proposed approach is significantly more computationally expensive than baselines (except DeepAugment) This is because the optimization process is solved over image-to-image translation networks. Unlike classification networks, these models largely operate on full-resolution images throughout the network, thus have much higher latency. Additionally the optimization process is solved individually for each sample. Can authors provide a comparison of the computation cost with baselines?

What is the impact of architecture-specific priors? There is a common trend, where U-Net and VQ-VAE models consistently perform worse than their counterparts (table 1, 2). In particular, clean accuracy itself degrades significantly when using them as a source of adversarial dataset augmentation. Is it because of their inability to reconstruct the input image in absence of perturbation or hard to achieve high fidelity in presence of perturbations. Can authors shed some light on this phenomenon?

Moving beyond image corruptions: Due to use of image-to-image translation network, current approach is mostly limited to image corruptions. For example, it is very hard for an image-to-image translation network to model affine transformations. Modelling such non-local transformations is an active area of research [1]. I encourage authors to discuss this limitation.

Another missing analysis in the paper is towards understanding how the proposed augmentations generalize to unseen corruptions at test time. I agree that the image-to-image translation network is expressive enough to model most image corruptions techniques, when specifically optimized for them (figure 3). However, we operate in an untargeted setting, i.e., not aware of the test-time corruption mechanism. In this untargeted setting, does the translation network using corruptions similar to the existing set of corruptions (such as blurring, weather changes)? This can be potentially measured with a classifier trained to recognize presence of specific corruptions in an image.

On a minor note, range of SSIM values in fig. 3 appears above 1 (which is not feasible since SSIM is bounded between [-1, 1]. Is it just a plotting issue with errorbars?


1. ​​Richardson, Eitan, and Yair Weiss. "The surprising effectiveness of linear unsupervised image-to-image translation." 2020 25th International Conference on Pattern Recognition (ICPR). IEEE, 2021.




**Summary Of The Paper:**

This paper makes use of image-to-image translation networks to generate adversarial data augmentation for input images. When combined with existing approaches, it achieves state-of-the-art robustness against unseen corruptions and worst-case perturbations.

**Summary Of The Review:**

This paper provides a general framework to defend against unseen image corruptions and adversarial perturbation. The general nature of the framework makes it feasible to use it with a wide range of corruptions.

---

> ### Author Response · Authors · 2021-11-19
> **Rebuttal to Reviewer fuXu (part 1)**
>
> We thank the reviewer for their thoughtful review and their appreciation of the work. We have split our reply into multiple comments.
>
> > "[...] more computationally expensive than baselines [...]"
>
> Our method (AdA) is indeed more computationally demanding than counterparts which utilize handcrafted heuristics. For example, AugMix specifies K image operations, which are applied stochastically at the input (i.e. over images). DeepAugment requires applying heuristic stochastic operations to the weights and activations of an image-to-image model - so this requires a forward pass through the image-to-image model (which can be computed offline). AdA must optimize over perturbations to the parameters of the image-to-image model; this requires several backpropagation steps and is similar to Lp-norm adversarial training (Madry et al., 2018) and related works.
> We summarize the computational requirements of our method (AdA), AugMix, DeepAugment, Perceptual Adversarial Training (PAT) (Laidlaw et al., 2021), Lp norm adversarial training (AT), TRADES (Zhang et al., 2019), Adversarial Weight Perturbations (AWP) (Wu et al., 2020) and Sharpness Aware Minimization (SAM) (Foret et al., 2021) in the table below:
>
> | Setup | Passes through classifier ($f_\theta$) | Passes through aux. net ($c_\phi$) | Adv. Projection |
> |:----------------|:-------------------------------------------|:---------------------------------------|:----------------------------------------------|
> | AdA (this work) | $\mathcal{O}(M)$ | $\mathcal{O}(M)$ | $\mathcal{O}(M \cdot \|\phi\| + \|x\|)$ |
> | PAT | $\mathcal{O}(M \cdot S)$ | $\mathcal{O}(M \cdot S)$ | $\mathcal{O}(N)$ |
> | AT | $\mathcal{O}(M)$ | - | $\mathcal{O}(M \cdot \|x\|)$ |
> | TRADES | $\mathcal{O}(M)$ | - | $\mathcal{O}(M \cdot \|x\|)$ |
> | AWP | $\mathcal{O}(M + W)$ | - | $\mathcal{O}(M\cdot\|x\| + W\cdot\|\theta\|)$ |
> | SAM | $\mathcal{O}(W)$ | - | - |
> | DeepAugment | $\mathcal{O}(1)$ | $\mathcal{O}(1)$ | - |
> | AugMix | $\mathcal{O}(1)$ | - | - |
>
> In this table $M$ represents the number of inner/adversarial optimizer steps (i.e. PGD steps in AdA or AT); $|x|$ is the dimensionality of the input; $|\theta|$ and $|\phi|$ are the number of parameters of the main classifier and of the auxiliary network respectively. The number of adversarial weight perturbations in AWP and SAM is denoted by $W$ (W is typically 1). For PAT, we also refer to the number of bisection iterations as $N$ and to the number of steps used for updating the Lagrange multiplier as $S$.
>
> We characterize the number of passes (forward and backward, jointly) through the main classifier network ($f_\theta$) and, for AdA and PAT, through an auxiliary neural network ($c_\phi$) separately. For methods using adversarial optimization we also describe the cost of the projection steps used for keeping perturbations within the feasible set. Taken together, these costs represent the computational complexity of performing one training step with each method.
>
> That being said, we highlight that *our method does not require the manual construction of heuristic transformations which can be time consuming*. Just to highlight the difficulty of building such heuristics, here is one of the 25 random weight transformations done by DeepAugment:
>
> ```
> i = np.random.choice(np.arange(1,13,4))
> z = torch.zeros_like(weights['body.'+str(i)+'.body.0.weight'])
> for j in range(z.size(0)):
>     shift_x, shift_y = np.random.randint(3, size=(2,))
>     z[:,j,shift_x,shift_y] = np.random.choice([1.,-1.])
> weights['body.'+str(i)+'.body.0.weight'] = conv2d(weights['body.'+str(i)+'.body.0.weight'], z, padding=1)
> ```
>
> As one can see, these are not only far from trivial but also do not generalize to other image-to-image models to the contrary of our method. We emphasize this in the introduction and have now added a new section in Appendix B.1 to analyze the computational and memory requirements of our method and compare it with those of seven related works.

---

> ### Author Response · Authors · 2021-11-19
> **Rebuttal to Reviewer fuXu (part 2)**
>
> > What is the impact of architecture-specific priors?  [...] Can authors shed some light on this phenomenon?
>
> There are significant differences between the image-to-image models and this also results in significant differences in downstream robustness (as shown in Tables 1 & 2).
> Throughout the paper, we experiment with four architectures. The VQ-VAE and CAE networks are trained to compress images and are lossy by definition (i.e. they are limited in their capacity to reconstruct their inputs). The EDSR and U-Net models are both trained to preserve, enhance or hallucinate fine-grained details (i.e. EDSR is trained on a super-resolution task and the U-Net is trained on an image completion task with randomly dropped pixels). Due to architectural and training objectives' similarities, we expect the resulting performance from VQ-VAE or CAE to be more similar to each other than the performance resulting from U-Net or EDSR -- and this is what we observe in Tables 1 and 2.
>
> In Figure 2 we analyze the distribution of SSIM distances between adversarial examples and their original counterparts for two image-to-image models. We observe that using the CAE lossy autoencoder results in images which are considerably more different than their original counterparts (wide distribution with low SSIM mean) than when using the EDSR image-to-image model which preserves fine-grained details (and results in a sharp distribution with a high SSIM mean). We expect similar relationships to hold for VQ-VAE (cf. CAE) and U-Net (cf. EDSR).
>
> Indeed, it is difficult to know which architecture will perform best. In some cases, this decision can be side-stepped by ensembling all available models: in Table 2 we show that, for ImageNet, ensembling all four models actually results in the best overall performance.
>
>
> >Moving beyond image corruptions: Due to use of image-to-image translation network, current approach is mostly limited to image corruptions. For example, it is very hard for an image-to-image translation network to model affine transformations. Modelling such non-local transformations is an active area of research [1]. I encourage authors to discuss this limitation.
>
> Thank you for pointing this out. Most image-to-image translation networks are indeed not well suited for capturing global image transformations such as affine transformations [1]. Any method which relies on image-to-image models (e.g., our method or DeepAugment) will be inheriting the limitations of the specific image-to-image models themselves.
>
> But, compared to DeepAugment, which relies on specific image-to-image models (as it uses handcrafted operations which only apply to these models), our method is more flexible as it can leverage **any**  image-to-image model.
>
> Regarding modeling affine transformations specifically: consider an image-to-image model which consists of a single spatial transformer layer [2] only. Applied at the input layer, the spatial transformer differentiably parameterizes a global image transformation, which includes affine transformations. Using our method with such an image-to-image model would directly allow our method to find adversarial examples by optimizing over the global image transformation.
>
> We have updated our paper (Appendix A) to include this discussion.
>
> [1] Richardson, Eitan, and Yair Weiss. "The surprising effectiveness of linear unsupervised image-to-image translation." 2020 25th International Conference on Pattern Recognition (ICPR). IEEE, 2021.
>
> [2] Max Jaderberg et al., "Spatial Transformer Networks". NeurIPS 2015.

---

> ### Author Response · Authors · 2021-11-19
> **Rebuttal to Reviewer fuXu (part 3)**
>
> > Another missing analysis in the paper is towards understanding how the proposed augmentations generalize to unseen corruptions at test time. I agree that the image-to-image translation network is expressive enough to model most image corruptions techniques, when specifically optimized for them (figure 3). However, we operate in an untargeted setting, i.e., not aware of the test-time corruption mechanism. In this untargeted setting, does the translation network using corruptions similar to the existing set of corruptions (such as blurring, weather changes)? This can be potentially measured with a classifier trained to recognize presence of specific corruptions in an image.
>
> First, we highlight that all corruptions (such as those from CIFAR-10-C or ImageNet-C) are unknown at training time. Second, since we find the worst-case corruptions at each training step (from the image-to-image models), we expect the resulting classifier to be robust to a large set of corruptions (which may or may not be included in the test-time corruptions) rather than the most common corruptions that can be generated by the image-to-image models (see Appendix F). Finally, we should not expect adversarial examples to be similar to test-time corruptions. In essence, these adversarial examples consist of the images that provide the most information to the classifier in order to improve its performance (over all corruptions that can be expressed by the image-to-image models).
>
> > On a minor note, range of SSIM values in fig. 3 appears above 1 (which is not feasible since SSIM is bounded between [-1, 1]. Is it just a plotting issue with errorbars?
>
> Yes, this was a plotting issue. We have now updated the figure so that each error bar represents the 95% confidence interval around the mean.

---

> ### Author Response · Authors · 2021-11-23
> **Addressed comments in-depth**
>
> We believe we have addressed the reviewer's comments in-depth. In light of this, we would be grateful if the reviewer will consider raising their recommendation score.

---

> ### Comment · Reviewer_fuXu · 2021-11-23
> **Further comments**
>
> I believe that the work now provides a clearer picture of limitations since authors have specifically discussed the limitation of proposed method in terms of computational cost. In general, finding worst-case perturbation is often computationally extensive since it involves an optimization process. It is remain to be seen in future work, how one can reduce the computational cost while reducing the cost (may be a single-shot generative method)
>
> In summary, this work generates worst-case perturbations on data manifold by exploiting the representations learned by most pixel-to-pixel translation network. Authors provide a rigorous set of results validating the effectiveness of this approach. However, it's unclear what type of transformations the network generated, which led to an improvement in final performance. I do want to point out that this is a limitation of not only this work but also previous works in the similar direction [1] and better tools needs to be developed to understand why a particular data augmentation strategy leads to a widespread impact on multiple test-time corruptions.
>
> I believe this work provides a novel and interesting approach towards solving the problem of distributional robustness. While it suffers some from some limitations, they provide avenues for future work. Overall I believe the the program will benefit from the inclusion of this paper and I recommend its acceptance.
>
> [1]. Laidlaw, Cassidy, Sahil Singla, and Soheil Feizi. "Perceptual Adversarial Robustness: Defense Against Unseen Threat Models." International Conference on Learning Representations. 2020.

---

### Official Review · Reviewer_w3th · 2021-11-04

**Correctness:** 4
**Technical Novelty And Significance:** 3
**Empirical Novelty And Significance:** 3
**Recommendation:** 8
**Confidence:** 3

**Main Review:**

- Strength:
    - Connecting common image corruption robustness and adversarial robustness is a good direction.
    - While models trained with AdversarialAugment is not specifically designed to defend against lp-norm bounded adversarial perturbations, their classifiers show some resilience. This is an important improvement over DeepAugment+AugMix, because these previous approaches didn't show robustness against adversarial perturbations. Previously, adversarially-trained networks were shown to be robust to some corruption types, but they fail at corruption types like Fog. Future work might be able to fix this gap by leveraging AdversarialAugment.

- Weaknesses:
    - The current formulation seems to be much more computationally demanding than their counterparts such as AugMix, DeepAugment etc. It would be more fair to discuss the limitation on this aspect and/or compare the baselines controlled for similar computational resources.

**Summary Of The Paper:**

The authors propose a new data augmentation technique, called AdversarialAugment, to increase robustness of image classification models.
The proposed method optimizes the parameters of image-to-image models to generate adversarially corrupted images, where they also show sufficient conditions for the consistency in the simple setting.
They empirically show that AdversarialAugment improves common corruption robustness on CIFAR-10-C as well as worst-case performance against lp-norm bounded perturbations on CIFAR-10 and ImageNet.

**Summary Of The Review:**

This paper contributes to the body of work that tries to tackle both types of robustness problems (common corruption and adversarial corruption) under a unifying view, which is an important step. While their method seems to be computationally demanding, it demonstrates good empirical performance along with theoretical justification.

---

> ### Author Response · Authors · 2021-11-19
> **Rebuttal to Reviewer w3th (part 1)**
>
> We thank the reviewer for their thoughtful review and their appreciation of the work. We have split our reply into multiple comments.
>
> > The current formulation seems to be much more computationally demanding than their counterparts such as AugMix, DeepAugment etc. It would be more fair to discuss the limitation on this aspect [...]
>
> Our method (AdA) is indeed more computationally demanding than counterparts which utilize handcrafted heuristics. For example, AugMix specifies K image operations, which are applied stochastically at the input (i.e. over images). DeepAugment requires applying heuristic stochastic operations to the weights and activations of an image-to-image model - so this requires a forward pass through the image-to-image model (which can be computed offline). AdA must optimize over perturbations to the parameters of the image-to-image model; this requires several backpropagation steps and is similar to Lp-norm adversarial training (Madry et al., 2018) and related works.
> We summarize the computational requirements of our method (AdA), AugMix, DeepAugment, Perceptual Adversarial Training (PAT) (Laidlaw et al., 2021), Lp norm adversarial training (AT), TRADES (Zhang et al., 2019), Adversarial Weight Perturbations (AWP) (Wu et al., 2020) and Sharpness Aware Minimization (SAM) (Foret et al., 2021) in the table below:
>
> | Setup | Passes through classifier ($f_\theta$) | Passes through aux. net ($c_\phi$) | Adv. Projection |
> |:----------------|:-------------------------------------------|:---------------------------------------|:----------------------------------------------|
> | AdA (this work) | $\mathcal{O}(M)$ | $\mathcal{O}(M)$ | $\mathcal{O}(M \cdot \|\phi\| + \|x\|)$ |
> | PAT | $\mathcal{O}(M \cdot S)$ | $\mathcal{O}(M \cdot S)$ | $\mathcal{O}(N)$ |
> | AT | $\mathcal{O}(M)$ | - | $\mathcal{O}(M \cdot \|x\|)$ |
> | TRADES | $\mathcal{O}(M)$ | - | $\mathcal{O}(M \cdot \|x\|)$ |
> | AWP | $\mathcal{O}(M + W)$ | - | $\mathcal{O}(M\cdot\|x\| + W\cdot\|\theta\|)$ |
> | SAM | $\mathcal{O}(W)$ | - | - |
> | DeepAugment | $\mathcal{O}(1)$ | $\mathcal{O}(1)$ | - |
> | AugMix | $\mathcal{O}(1)$ | - | - |
>
> In this table $M$ represents the number of inner/adversarial optimizer steps (i.e. PGD steps in AdA or AT); $|x|$ is the dimensionality of the input; $|\theta|$ and $|\phi|$ are the number of parameters of the main classifier and of the auxiliary network respectively. The number of adversarial weight perturbations in AWP and SAM is denoted by $W$ (W is typically 1). For PAT, we also refer to the number of bisection iterations as $N$ and to the number of steps used for updating the Lagrange multiplier as $S$.
>
> We characterize the number of passes (forward and backward, jointly) through the main classifier network ($f_\theta$) and, for AdA and PAT, through an auxiliary neural network ($c_\phi$) separately. For methods using adversarial optimization we also describe the cost of the projection steps used for keeping perturbations within the feasible set. Taken together, these costs represent the computational complexity of performing one training step with each method.
>
> We have added a new section in the paper (Appendix B.1) which includes an in-depth analysis of the computational and memory requirements of our method and a comparison with those of seven related works.

---

> ### Author Response · Authors · 2021-11-19
> **Rebuttal to Reviewer w3th (part 2)**
>
> > "[...] or compare the baselines controlled for similar computational resources."
>
> Our method has similar computational requirements to other methods relying on adversarial training (PAT, AT, TRADES, AWP, SAM). Results comparing our method to PAT appear in the original paper in Appendix F. We have also run new experiments to compare our method with AT, TRADES, AWP and SAM and include them below. We have updated our paper to include these new results in Appendix F.
>
> | Setup | mCE | Noise err. | Blur err. | Weather err. | Digital err. | Cifar-10 acc. | Cifar-10.1 acc. | $\ell_2$ acc. $\epsilon=0.5$ | $\ell_2$ acc. $\epsilon=1.0$ | $\ell_\infty$ acc. $\epsilon=\frac{1}{255}$ | $\ell_\infty$ acc. $\epsilon=\frac{2}{255}$ |
> |:-----------------------|:----------|:----------------|:---------------|:------------------|:------------------|:------------------|:--------------------|:--------------------------------|:--------------------------------|:------------------------------------------------|:-------------------------------------------------|
> | AT ($\ell_\infty$) | 23.64 | 20.56 | 20.29 | 25.90 | 27.04 | 86.08 | 74.20 | 51.66 | 16.64 | 82.32 | 78.20 |
> | AT ($\ell_2$) | 17.42 | **13.93** | 15.20 | 19.34 | 20.33 | 91.02 | 80.40 | 67.06 | 30.73 | **85.98** | 79.77 |
> | TRADES ($\ell_\infty$) | 24.72 | 21.31 | 21.49 | 27.26 | 27.99 | 84.79 | 71.00 | 54.55 | 19.51 | 81.33 | 77.59 |
> | TRADES ($\ell_2$) | 18.08 | 14.07 | 15.64 | 20.46 | 21.15 | 89.96 | 78.60 | 68.16 | 37.01 | 85.07 | 79.12 |
> | AWP ($\ell_\infty$) | 25.01 | 21.73 | 21.58 | 27.69 | 28.23 | 84.36 | 70.50 | 57.01 | 24.07 | 81.38 | 77.98 |
> | AWP ($\ell_2$) | 18.79 | 14.92 | 15.87 | 21.54 | 21.86 | 89.20 | 77.40 | **71.79** | **44.83** | 85.18 | **80.21** |
> | SAM | 24.59 | 49.44 | 24.74 | 12.80 | 17.59 | **96.17** | **90.35** | 0.01 | 0.00 | 48.10 | 6.86 |
> | AdA (EDSR) | **15.47** | 26.83 | **14.94** | **10.80** | **12.14** | 93.37 | 86.55 | 9.28 | 0.09 | 72.41 | 41.13 |
>
> In terms of robustness to common image corruptions, our method compares favorably to PAT and a suite of Lp-norm adversarial training methods, which all have comparable runtime requirements to AdA as they must also take gradient steps at runtime.
>
> On an equal footing of computational requirements with DeepAugment we can look at sampling only parameter perturbations (rather than optimizing for them) - these results are present in the original paper in Appendix F ("Effect of number of inner optimization steps on robustness to common image corruptions"); we include them below for completeness. We observe that we can already improve the mCE over nominally trained models by more than 11% by using just 2 steps (from 29.37% to 17.91%).
>
> | # PGD steps | Clean Err. | mCE | Noise: Gauss | Noise: Shot | Noise: Impulse | Blur: Defocus | Blur: Glass | Blur: Motion | Blur: Zoom | Weather: Snow | Weather: Frost | Weather: Fog | Weather: Bright | Digital: Contrast | Digital: Elastic | Digital: Pixel | Digital: JPEG |
> |:---------------------------|:----------------|:----------|:-----------------|:-----------------|:--------------------|:-------------------|:-----------------|:------------------|:----------------|:-------------------|:--------------------|:-------------------|:---------------------|:-----------------------|:----------------------|:--------------------|:-------------------|
> | 0 (i.e. only random init.) | 8.99 | 28.35 | 43.4 | 34.1 | 35.3 | 22.6 | 49.9 | 30.0 | 29.1 | 22.9 | 24.6 | 16.3 | 10.5 | 28.2 | 23.6 | 31.0 | 23.6 |
> | 2 | 7.87 | 17.91 | **21.5** | **18.3** | 28.5 | 18.8 | 27.5 | 17.7 | 23.7 | 16.0 | 15.4 | **11.3** | 8.5 | **12.6** | 18.6 | 14.0 | 16.4 |
> | 4 | 6.99 | **14.91** | 26.2 | 21.0 | **25.8** | 12.0 | 18.6 | **12.9** | 14.7 | 12.8 | 11.7 | 11.6 | 7.4 | 13.7 | 13.4 | 9.8 | 12.0 |
> | 6 | 6.83 | 15.94 | 30.2 | 23.7 | 29.8 | 12.1 | 19.2 | 13.7 | 15.4 | 13.0 | 12.2 | 12.8 | 7.2 | 15.1 | 13.3 | 9.5 | 11.9 |
> | 8 | 6.80 | 15.66 | 28.0 | 22.4 | 30.5 | 12.6 | **18.4** | 14.3 | 15.0 | **12.7** | 11.9 | 12.2 | 7.3 | 15.1 | 13.0 | 9.7 | 12.0 |
> | 10 | **6.63** | 15.47 | 27.3 | 21.5 | 31.7 | **11.8** | 20.0 | 13.9 | **14.1** | 12.9 | **11.5** | 11.8 | **7.0** | 14.9 | **12.7** | **9.4** | **11.5** |
>
> That being said, we highlight that *our method does not require the manual construction of heuristic transformations which can be time consuming*.

---

> ### Author Response · Authors · 2021-11-19
> **Rebuttal to Reviewer w3th (part 3)**
>
> > Correctness: 3: Some of the paper’s claims have minor issues. A few statements are not well-supported, or require small changes to be made correct.
>
> We believe that we have addressed the reviewer's comments in our updated paper (with extended computational and space complexity analysis and the inclusion of additional experimental results on Lp-norm adversarial training methods) and would kindly ask if they are willing to review the correctness score (as we believe that all claims in the paper are backed either theoretically or experimentally).

---

> > ### Author Response · Authors · 2021-11-23
> > **Addressed comments in-depth**
> >
> > We believe we have addressed the reviewer's comments in-depth. In light of this, we would be grateful if the reviewer will consider raising their recommendation score.

---

> > > ### Comment · Reviewer_w3th · 2021-11-30
> > > **Response**
> > >
> > > I thank for the detailed responses. The paper now clearly discusses the limitations and compares the proposed method with other baselines with similar computational requirements. Based on this, I revised the correctness score as well as the recommendation score.

---

### Decision · Program_Chairs · 2022-01-20

**Decision:**

Accept (Poster)

**Comment:**

The paper proposes an adversarial data augmentation technique searching for adversarial weight perturbations of a corruption network (e.g. a pretrained image-to-image model). The goal is to achieve common corruption robustness as well as a non-trivial level of adversarial robustness. The authors claim state-of-the-art-performance on CIFAR10-C.

Most reviewers had initial concerns which the authors could clarify in most cases. Finally, all reviewers argue for acceptance.

Strengths:
- no pre-defined corruption model necessary
- extensive experiments on CIFAR10 and ImageNet with SOTA results
- all reviewers agree that this paper would be valuable as a future reference

Weaknesses:
- the theoretical part is in my point of view rather misleading and should be completely rewritten as the authors lack here rigor concerning the setting they are working in (as also Reviewer 31sh criticizes). In particular the corruptions are not just a covariate shift (p(y|x) is invariant, only p(x) changes) as the corruptions mix the conditional distributions of different points. Thus under the given assumptions (which should be summarized at one point rather than being scattered over the text) the Bayes optimal classifer is invariant but not the Bayes optimal predictive probability distribution (which is clearly important for assessing uncertainty of the prediction).
However, when training with the cross-entropy loss we are estimating the predictive probability distribution and thus the given statement
about convergence of the risks makes no sense for me and the optimal parameters of the classifier need not be equal even if their classifications agree everywhere. Thus the required changes to fix this theoretical part go significantly beyond what the authors suggest in their rebuttal.
- the improvements over AugMix+DeepAugment (7.83 mCE vs 7.99 mCE) are negligible and most likely not statistically significant
While AdA has significantly higher adversarial robustness than AugMix+DeepAugment, it remains unclear if using AugMix+DeepAugment together with adversarial training for l_2 as done in
 [3] Kireev, Klim, Maksym Andriushchenko, and Nicolas Flammarion. "On the effectiveness of adversarial training against common corruptions." arXiv preprint arXiv:2103.02325 (2021).
could have led to a similar result

The paper provides an interesting approach to achieve robustness against common corruptions and all reviewers recommend acceptance. The theoretical part as written currently is misleading as discussed above - the authors have to make it completely rigorous (including proofs, formal statements/defininitions etc) or get rid of it.

Minor weak points:
- According to the leaderboard of RobustBench the SOTA for CIFAR10-C is  by now taken by the NeurIPS 2021 paper
Diffenderfer et al,  A Winning Hand: Compressing Deep Networks Can Improve Out-Of-Distribution Robustness
which achieves 96.56% standard accuracy and 92.78% mCE. This is 1.64% and 0.61% better than in the present paper and needs
to be discussed as prior work.
- for the adversarial robustness evaluation regarding "AutoAttack & MultiTargeted" you refer to Gowal et al (2020) but the robustness
evaluation has to be properly discussed in this paper